# Evaluating stratospheric ozone and water vapor changes in CMIP6 models from 1850-2100

James Keeble[1,2], Birgit Hassler[3], Antara Banerjee[4,5], Ramiro Checa-Garcia[6], Gabriel Chiodo[7,8], Sean Davis[4], Veronika Eyring[3,9], Paul T. Griffiths[1,2], Olaf Morgenstern[10], Peer Nowack[11,12], Guang Zeng[10], Jiankai Zhang[13], Greg Bodeker[14,15], Susannah Burrows[16], Philip Cameron-Smith[17], David Cugnet[18], Christopher Danek[19], Makoto Deushi[20], Larry W. Horowitz[21], Anne Kubin[22], Lijuan Li[23], Gerrit Lohmann[19], Martine Michou[24], Michael J. Mills[25], Pierre Nabat[24], Dirk Olivié[26], Sungsu Park[27], Øyvind Seland[26], Jens Stoll[22], Karl-Hermann Wieners[28], Tongwen Wu[29]

[1]Department of Chemistry, University of Cambridge, Cambridge, UK
[2]National Centre for Atmospheric Science (NCAS), University of Cambridge, Cambridge, UK
[3]Deutsches Zentrum für Luft- und Raumfahrt (DLR), Institut für Physik der Atmosphäre, Oberpfaffenhofen, Germany
[4]NOAA Earth System Research Laboratory Chemical Sciences Division, Boulder, CO USA
[5]Cooperative Institute for Research in Environmental Sciences (CIRES), University of Colorado Boulder, Boulder, CO, USA
[6]Laboratoire des sciences du climat et de l'environnement: Gif-sur-Yvette, Île-de-France, FR
[7]Department of Environmental Systems Science, Swiss Federal Institute of Technology, Zurich, Switzerland
[8]Department of Applied Physics and Applied Math, Columbia University, New York, NY, USA
[9]University of Bremen, Institute of Environmental Physics (IUP), Bremen, Germany.
[10]National Institute of Water and Atmospheric Research (NIWA), Wellington, New Zealand
[11]Grantham Institute, Department of Physics and the Data Science Institute, Imperial College London, London, UK.
[12]Climatic Research Unit, School of Environmental Sciences, University of East Anglia, Norwich, UK.
[13]Key Laboratory for Semi-Arid Climate Change of the Ministry of Education, College of Atmospheric Sciences, Lanzhou University, Lanzhou, 730000, Gansu, China
[14]Bodeker Scientific, 42 Russell Street, Alexandra, 9320, New Zealand
[15]School of Geography, Environment and Earth Sciences, Victoria University, New Zealand
[16]Atmospheric Sciences & Global Change Division, Pacific Northwest National Laboratory, Richland, WA, USA
[17]Atmosphere, Earth and Energy Division, Lawrence Livermore National Laboratory, Livermore, CA 94550, USA
[18]Laboratoire de Météorologie Dynamique, Institut Pierre-Simon Laplace, Sorbonne Université / CNRS / École Normale Supérieure – PSL Research University / École Polytechnique – IPP, Paris, France
[19]Alfred Wegener Institute, Helmholtz Centre for Polar and Marine Sciences, Bremerhaven, Germany
[20]Meteorological Research Institute (MRI), Tsukuba, Japan
[21]GFDL/NOAA, Princeton, NJ, USA.
[22]Leibniz Institute for Tropospheric Research, Leipzig, Germany
[23]State Key Laboratory of Numerical Modeling for Atmospheric Sciences and Geophysical Fluid Dynamics (LASG), Institute of Atmospheric Physics, Chinese Academy of Sciences, Beijing 100029, China
[24]CNRM, Université de Toulouse, Météo-France, CNRS, Toulouse, France
[25]Atmospheric Chemistry Observations and Modeling Laboratory, National Center for Atmospheric Research, Boulder, CO, USA
[26]Norwegian Meteorological Institute, Oslo, Norway
[27]Seoul National University, Seoul, South Korea
[28]Max Planck Institute for Meteorology, Hamburg, Germany
[29]Beijing Climate Center, China Meteorological Administration, Beijing, China

Correspondence to: James Keeble (james.keeble@atm.ch.cam.ac.uk)

**Abstract.** Stratospheric ozone and water vapor are key components of the Earth system, and past and future changes to both have important impacts on global and regional climate. Here we evaluate long-term changes in these species from the pre-industrial (1850) to the end of the 21$^{st}$ century in CMIP6 models under a range of future emissions scenarios. There is good agreement between the CMIP multi-model mean and observations for total column ozone (TCO), although there is substantial variation between the individual CMIP6 models. For the CMIP6 multi-model mean, global mean TCO has increased from ~300 DU in 1850 to ~305 DU in 1960, before rapidly declining in the 1970s and 1980s following the use and emission of halogenated ozone depleting substances (ODSs). TCO is projected to return to 1960's values by the middle of the 21$^{st}$ century under the SSP2-4.5, SSP3-7.0, SSP4-3.4, SSP4-6.0 and SSP5-8.5 scenarios, and under the SSP3-7.0 and SSP5-8.5 scenarios TCO values are projected to be ~10 DU higher than the 1960's values by 2100. However, under the SSP1-1.9 and SSP1-1.6 scenarios, TCO is not projected to return to the 1960's values despite reductions in halogenated ODSs due to decreases in tropospheric ozone mixing ratios. This global pattern is similar to regional patterns, except in the tropics where TCO under most scenarios is not projected to return to 1960's values, either through reductions in tropospheric ozone under SSP1-1.9 and SSP1-2.6, or through reductions in lower stratospheric ozone resulting from an acceleration of the Brewer-Dobson Circulation under other SSPs. In contrast to TCO, there is poorer agreement between the CMIP6 multi-model mean and observed lower stratospheric water vapour mixing ratios, with the CMIP6 multi-model mean underestimating observed water vapour mixing ratios by ~0.5 ppmv at 70hPa. CMIP6 multi-model mean stratospheric water vapor mixing ratios in the tropical lower stratosphere have increased by ~0.5 ppmv from the pre-industrial to the present day and are projected to increase further by the end of the 21$^{st}$ century. The largest increases (~2 ppmv) are simulated under the future scenarios with the highest assumed forcing pathway (e.g. SSP5-8.5). Tropical lower stratospheric water vapor, and to a lesser extent TCO, show large variations following explosive volcanic eruptions.

## 1 Introduction

Stratospheric ozone and water vapor are key components of the Earth system, and past changes in both have had important impacts on global and regional climate (e.g. Solomon et al., 2010; Dessler et al., 2013; Eyring et al., 2013; WMO 2018). Depletion of the ozone layer over the last few decades of the 20$^{th}$ century, driven by emissions of halogenated ozone depleting substances (ODSs), provides an excellent illustration of a forcing that has caused large dynamical and regional surface impacts, despite an overall small global radiative forcing (-0.05±0.10 Wm$^{-2}$ from 1750 to 2011; IPCC, 2013). The Antarctic ozone hole has resulted in lower springtime Antarctic lower stratospheric temperatures and has driven a strengthening of the westerly jet and a poleward expansion of the Hadley cell during the SH summer season (e.g. Thompson and Solomon, 2002; Gillett and Thompson, 2003; McLandress et al., 2010; Son et al., 2010; Polvani et al., 2011; Braesicke et al.; 2013, Keeble et al., 2014;

Morgenstern et al., 2018). In contrast, while no long-term trend in stratospheric water vapour has been established (Scherer et al., 2008; Hurst et al., 2011; Hegglin et al., 2014), large decadal variations have been suggested to affect surface temperatures (e.g. Solomon et al., 2010). Given these climate impacts, it is important to understand the drivers of stratospheric ozone and water vapor and to distinguish long-term trends from interannual and decadal variability.

The Intergovernmental Panel on Climate Change (IPCC) Fifth Assessment Report (AR5) highlights tropospheric ozone as the third most important anthropogenic greenhouse gas (GHG) with a global mean radiative forcing of $0.35\pm0.2$ $Wm^{-2}$, while stratospheric water vapor changes resulting from $CH_4$ oxidation exert a global mean radiative forcing of $0.07\pm0.05$ $Wm^{-2}$ (Hansen et al 2005, IPCC, 2007; IPCC, 2013). The primary contributor to the radiative forcing estimate for ozone is increased tropospheric ozone ($0.4\pm0.2$ $Wm^{-2}$), while recent depletion of stratospheric ozone due to the use and emission of halogenated ODSs, compounded with impacts on ozone of increasing $CO_2$, $CH_4$, and $N_2O$, has resulted in a weakly negative radiative forcing ($-0.05\pm0.1$ $Wm^{-2}$). Recently, Checa-Garcia at al. (2018a) estimated ozone radiative forcing using the ozone forcing dataset that was developed for the Coupled Model Intercomparison Project Phase 6 (CMIP6, Eyring et al., 2016), and calculated values of 0.28 $Wm^{-2}$, which, while consistent with the IPCC-AR5 estimate calculated using model simulations, represents an increase of ~80% compared to the CMIP5 ozone forcing dataset (Cionni et al 2011; Stevenson et al, 2013). The relative uncertainties in radiative forcing estimates for both stratospheric ozone and water vapor are large due to the challenges in constraining the concentrations of both during the pre-satellite era. As a result, the current radiative forcing estimates rely on ozone and water vapor fields derived from simulations performed by global climate models and Earth system models. However, models use different radiation schemes and model (in the case of models with interactive chemistry scheme) or prescribe (in the case of models without interactive chemistry schemes) ozone differently, further contributing to the uncertainty estimates.

Stratospheric ozone concentrations are determined by a balance between production and destruction of ozone through gas phase chemical reactions and transport (e.g. Brewer and Wilson, 1968). Gas phase ozone chemistry consists of sets of oxygen only photo-chemical reactions first described by Chapman (1930), alongside ozone destroying catalytic cycles involving chlorine, nitrogen, hydrogen and bromine radical species (e.g. Bates and Nicolet, 1950; Crutzen, 1970; Johnston, 1971; Molina and Rowland, 1974; Stolarski and Cicerone, 1974). Heterogeneous processes play a major role in determining ozone abundances in the polar lower stratosphere (e.g. Solomon, 1999) and following large volcanic eruptions (e.g. Solomon et al., 1996; Telford et al., 2009).

Changes in anthropogenic emissions of halogenated ODSs, $N_2O$, $CH_4$, $CO_2$ and other GHGs during the 21[st] century are expected to perturb these chemical cycles either directly through their role as source gases or by changing stratospheric temperatures and dynamics (Eyring et al., 2010; Keeble et al., 2017). Following the implementation of the Montreal Protocol and its subsequent Amendments, stratospheric concentrations of inorganic chlorine and bromine levelled off in the mid-1990s and are now in decline (Mäder et al., 2010; WMO, 2018), which has led to early signs of recovery of stratospheric ozone

(Keeble et al., 2018; Weber et al., 2018; WMO 2018) and the detection of statistically robust positive trends in September

Antarctic ozone (Solomon et al., 2016). Total column ozone in the mid- and high latitudes is projected to return to pre-1980 values during the coming decades (Eyring et al., 2013; Dhomse et al., 2018; WMO, 2018). Future emissions of $CH_4$ and $N_2O$, which are not regulated in the same way as halogenated ODSs, are associated with greater uncertainty and future concentrations of $HO_x$ (H, OH, $HO_2$) and $NO_x$ (NO, $NO_2$) radicals are highly sensitive to assumptions made about their future emissions. Additionally, increases in GHG concentrations are expected to lead to an acceleration of the Brewer–Dobson circulation (BDC;

Butchart et al., 2006, 2010; Shepherd and McLandress, 2011; Hardiman et al., 2014; Palmeiro et al., 2014), which may affect ozone concentrations directly through transport (e.g. Plumb, 1996; Avallone and Prather, 1996) and by influencing the chemical lifetimes of $Cl_y$, $NO_y$ and $HO_x$, source gases (e.g. Revell et al., 2012; Meul et al., 2014). However, recent research (Polvani et al., 2018, 2019) has shown, using model simulations, that stratospheric ozone depletion caused by increasing ODSs may have accounted for around half of the acceleration of the BDC in recent decades. As concentrations of ODSs decline,

stratospheric ozone recovery may offset, at least in part, future changes to the speed of the BDC resulting from GHG changes.

Stratospheric water vapor concentrations are determined predominantly through a combination of the dehydration air masses experience as they pass through the cold point tropical tropopause (Brewer, 1949; Fueglistaler et al., 2005) and *in-situ* production from $CH_4$ oxidation (Brasseur and Solomon, 1984; Jones et al., 1986; LeTexier et al., 1988). Direct injection by convective overshooting (Dessler et al., 2016) or following volcanic eruptions (Murcray et al., 1981; Sioris et al. 2016) are

120 also sources of stratospheric water vapor.

Observations of stratospheric water vapor show an increase during the late $20^{th}$ century (e.g. Rosenlof et al., 2001; Scherer et al., 2008; Hurst et al., 2011), followed by a sudden decrease of ~10% after 2000 (e.g. Solomon et al., 2010). Virtually all models project increases in stratospheric water vapor concentrations under increased $CO_2$ (e.g. Gettelman et al., 2010; Banerjee et al., 2019). Projected increases over the course of the $21^{st}$ century occur due to the predominant effect of increases in upper

tropospheric temperatures, offset in part by the effects of a strengthening BDC (Dessler et al., 2013; Smalley et al., 2017), with additional impacts from future $CH_4$ emissions (Eyring et al., 2010; Gettelman et al., 2010). Eyring et al. (2010) calculate a mean increase of 0.5-1 ppmv per century in stratospheric water vapor concentrations for models contributing to the Chemistry-Climate Model Validation (CCMVal) inter-comparison project, although agreement between models on the absolute increase is poor.

To advance our understanding of long-term changes to a number of components of the Earth system, including stratospheric ozone and water vapor, the CMIP Panel, operating under the auspices of the Working Group on Coupled Modelling (WGCM) of the World Climate Research Programme (WCRP), has defined a suite of climate model experiments, which together form CMIP6 (Eyring et al., 2016). Between the previous phase (CMIP5; Taylor et al., 2012) and CMIP6 there has been further development of existing models, new models have joined and a new set of future scenarios, the shared socioeconomic pathways

(SSPs; Riahi et al., 2017) that are used in climate projections by CMIP6 models as part of the Scenario Model Intercomparison

Project (ScenarioMIP; O'Neill et al., 2015), have been established. Earth system models have been further developed with improved physical parametrizations and some have added additional Earth system components (e.g. atmospheric chemistry, nitrogen cycle, ice sheets). As a result of this advancement in model complexity, the CMIP6 multi-model ensemble provides an opportunity to re-assess past and projected future stratospheric ozone and water vapor changes. In this study, we evaluate these changes against observations over the last three decades and examine long-term changes in these quantities from 1850 to 2100 under the SSP scenarios. Section 2 describes the simulations and models used in this study, with a focus on the treatment of stratospheric ozone and water vapor. Long-term changes in ozone and water vapor are evaluated in Sections 3 and 4, respectively, and implications are discussed in Section 5. Our results inform future studies that use CMIP6 simulations to investigate stratospheric composition changes and associated impacts.

## 2 Models and Simulations

This study evaluates long-term ozone and water vapor changes in 22 models which have performed the CMIP historical simulation and a subset of which have performed ScenarioMIP simulations. The treatment of stratospheric chemistry varies significantly across the models evaluated in this study. We evaluate all models which have produced ozone and water vapor output, regardless of the complexity of the stratospheric chemistry used, as these models may be used in other studies to diagnose the impacts of stratospheric composition changes on radiative forcing and/or regional climate change. In this Section, the models and simulations used in the subsequent analysis Sections are described, along with the observational datasets used for evaluation. Several of the figures were created with the Earth System Model Evaluation Tool (ESMValTool) version 2.0 (Eyring et al., 2019; Righi et al., 2019), a diagnostic and performance metric tool for enhanced and more comprehensive Earth system model evaluation in CMIP.

### 2.1 Models

At the time of the preparation of this manuscript, 22 models (AWI-ESM-1-1-LR, BCC-CSM2-MR, BCC-ESM1, CESM2, CESM2-FV2, CESM2-WACCM, CESM2-WACCM-FV2, CNRM-CM6-1, CNRM-ESM2-1, E3SM-1-0, E3SM-1-1, FGOALS-g3, GFDL-CM4, GFDL-ESM4, IPSL-CM6A-LR, MPI-ESM-1-2-HAM, MPI-ESM1-2-HR, MPI-ESM1-2-LR, MRI-ESM2-0, NorESM2-MM, SAM0-UNICON, and UKESM1-0-LL) have provided ozone mixing ratios and 18 models (AWI-ESM-1-1-LR, BCC-CSM2-MR, BCC-ESM1, CESM2, CESM2-FV2, CESM2-WACCM, CESM2-WACCM-FV2, CNRM-CM6-1, CNRM-ESM2-1, E3SM-1-1, GFDL-CM4, IPSL-CM6A-LR, MPI-ESM-1-2-HAM, MPI-ESM1-2-HR, MPI-ESM1-2-LR, MRI-ESM2-0, NorESM2-MM, and UKESM1-0-LL) have provided water vapor as diagnostics. Of the 22 models analysed in this study, six (CESM2-WACCM, CESM2-WACCM-FV2, CNRM-ESM2-1, GFDL-ESM4, MRI-ESM2-0, and UKESM1-0-LL) use interactive stratospheric chemistry schemes, while three (CNRM-CM6-1, E3SM-1-0, and E3SM-1-1)

use a simple chemistry scheme. The remaining 13 (AWI-ESM-1-1-LR, BCC-CSM2-MR, BCC-ESM1, CESM2, CESM2-FV2, FGOALS-g3, GFDL-CM4, IPSL-CM6A-LR, MPI-ESM-1-2-HAM, MPI-ESM1-2-HR, MPI-ESM1-2-LR, NorESM2-MM, and SAM0-UNICON) do not include an interactive chemistry scheme, and instead prescribe stratospheric ozone according to the CMIP6 ozone dataset (except in the case of CESM2, CESM2-FV2, and NorESM2 which prescribe ozone values from simulations performed with the CESM2-WACCM model). Relevant details of each model are provided below, and a summary is provided in Table 1.

The CMIP6 ozone dataset (Checa-Garcia, 2018b) is designed to be used by those models without interactive chemistry and was created using a different approach from the previous CMIP5 ozone database (Cionni et al., 2011). The CMIP5 dataset was based on stratospheric ozone values from a combination of model and observational datasets between the 1970s and 2011, and extended into the past and future based on assumptions of changes to stratospheric chlorine and the 11-year solar cycle. Tropospheric ozone values were based on a mean field of two models with interactive chemistry. In contrast, the CMIP6 ozone dataset was created using simulations from the CMAM and CESM-WACCM models, which both performed the REF-C2 simulation as part of the Chemistry-Climate Model Initiative (Eyring, et al., 2013; Morgenstern et al., 2017). As a result, the CMIP6 dataset provides a full three-dimensional field of ozone mixing ratios created using a single, consistent approach for both the stratosphere and troposphere, extending from preindustrial times to present day, and until the end of the 21$^{st}$ century following the different SSP scenarios (O'Neill et al., 2015). However, as the CMIP6 dataset uses values from model simulations, it has biases with respect to observations and uncertainties associated with the projections of stratospheric ozone beyond the period observations exist for, both from the pre-industrial to the start of the observational record, and from the present day to the end of the 21$^{st}$ century under the different SSP scenarios. More details on the CMIP6 ozone dataset can be found in Checa-Garcia (2018b)

**AWI-ESM-1-1-LR:** The Alfred Wegener Institute Earth System Model (AWI-ESM) is the global coupled atmosphere-land-ocean-sea ice model AWI Climate Model (AWI-CM; Semmler et al. 2020) extended by a dynamic land cover change model. Atmosphere and land are represented by a 1.85 x 1.85 degree horizontal resolution configuration of ECHAM6 (47 vertical levels up to 0.01 hPa ~ 80 km; Stevens et al., 2013) which includes a land component (JSBACH; Reick et al., 2013). The ocean is represented by the sea ice-ocean model FESOM1.4 (Wang et al. 2014) which runs on an irregular grid with a nominal resolution of 150 km (smallest grid size 25 km). Tropospheric and stratospheric ozone is prescribed from the CMIP6 dataset (Checa-Garcia et al., 2018a). GHG concentrations including $CO_2$, $CH_4$, $N_2O$ and CFCs are prescribed after Meinshausen et al. (2017). Methane oxidation and photolysis of water vapor are parameterized for the stratosphere and mesosphere (further information in Section 2.1.2 of Schmidt et al., 2013 and references therein).

**BCC-CSM2-MR:** The BCC-CSM2-MR model, developed by the Beijing Climate Center, is a coupled ocean–atmosphere model. Ozone in the stratosphere and troposphere is prescribed using monthly mean time-varying gridded data from the CMIP6 dataset. Other GHG concentrations including $CO_2$, $N_2O$, $CH_4$, CFC-11, CFC-12 are monthly zonal-mean values using the

CMIP6 datasets (Meinshausen et al., 2017, 2019). Stratospheric water vapor concentrations are prognostic values calculated in a similar way to those in the troposphere. A full description and evaluation of the BCC-CSM2-MR model is provided by Wu et al. (2019a).

**BCC-ESM1:** The BCC-ESM1 model, developed by the Beijing Climate Center, is a fully coupled global climate-chemistry-aerosol model. Tropospheric ozone is modelled interactively using the MOZART2 chemistry scheme, while stratospheric ozone is prescribed to the zonally averaged, monthly mean values from 1850 to 2014 derived from the CMIP6 data package in the top two model layers and relaxed towards the CMIP6 dataset between these layers and the tropopause. GHG concentrations including $CH_4$, $N_2O$, $CO_2$, CFC-11, and CFC-12 are prescribed using CMIP6 historical forcing data as
suggested in the AerChemMIP protocol (Collins et al., 2017). Stratospheric water vapor is a prognostic variable without any special treatment of $CH_4$ oxidation. A full description and evaluation of the BCC-ESM1 model is provided by Wu et al. (2019b).

**CESM2:** The Community Earth System Model version 2 (CESM2) is the latest generation of the coupled climate/Earth system models developed as a collaborative effort between scientists, software engineers, and students from the National Center for
Atmospheric Research (NCAR), universities, and other research institutions. CESM2(CAM6) uses the Community Atmosphere Model version 6 as its atmosphere component, which has 32 vertical levels from the surface to 3.6 hPa (about 40 km) and a horizontal resolution of 1.25° longitude by 0.95° latitude, and limited interactive chemistry for tropospheric aerosols. GHG concentrations including $CH_4$, $N_2O$, $CO_2$, CFC-11eq, and CFC-12 are prescribed using CMIP6 historical forcing data. CESM2 uses datasets derived from previous runs of CESM2-WACCM6, which includes interactive chemistry, for
tropospheric oxidants ($O_3$, OH, $NO_3$, and $HO_2$; 3D monthly means), stratospheric water vapor production from $CH_4$ oxidation (3D monthly means), stratospheric aerosol (zonal 5-day means), and $O_3$ for use in radiative transfer calculations (zonal 5-day means). A full description and evaluation of the CESM2 model is provided by Danabasoglu et al. (2020).

**CESM2-FV2:** The Community Earth System Model version 2 – finite volume 2° (CESM2-FV2) is based on CESM2, but with a reduced horizontal resolution for the atmosphere and land components of 2.5° longitude by 1.8° latitude. CESM2-FV2 uses
the same finite volume (FV) dynamical core as CESM2. Three tuning parameters were adjusted to maintain consistency with CESM2. To maintain the top-of-atmosphere energy balance, the clubb_gamma_coef parameter, described in Danabasoglu et al. (2020), was reduced from 0.308 to 0.280, and the autoconversion size threshold from cloud ice to snow (micro_mg_dcs) was decreased from 500 to 200 μm. To maintain sea salt aerosol burdens consistent with observations, the emission factor for sea salt was changed from 1.0 to 1.1.

**CESM2-WACCM:** The CESM2-WACCM model uses the Whole Atmosphere Community Climate Model version 6 (WACCM6) as its atmosphere component. WACCM6 has 70 vertical levels from the surface to $6 \times 10^{-6}$ hPa (about 140 km), a horizontal resolution of 1.25° longitude by 0.95° latitude. WACCM6 features a comprehensive chemistry mechanism with a

description of the troposphere, stratosphere, mesosphere, and lower thermosphere (TSMLT), including 231 species, 150 photolysis reactions, 403 gas-phase reactions, 13 tropospheric heterogeneous reactions, and 17 stratospheric heterogeneous reactions. The photolytic calculations are based on both inline chemical modules and a lookup table approach. The chemical species within the TSMLT mechanism include the extended $O_x$, $NO_x$, $HO_x$, $ClO_x$, and $BrO_x$ chemical families, $CH_4$ and its degradation products, $N_2O$ (major source of $NO_x$), $H_2O$ (major source of $HO_x$), plus various natural and anthropogenic precursors of the $ClO_x$ and $BrO_x$ families. The TSMLT mechanism also includes primary non-methane hydrocarbons and related oxygenated organic compounds, and two very short-lived halogens ($CHBr_3$ and $CH_2Br_2$) which add an additional $\sim$5 ppt of inorganic bromine to the stratosphere. WACCM6 features a new prognostic representation of stratospheric aerosols based on sulfur emissions from volcanoes and other sources, and a new detailed representation of secondary organic aerosols (SOAs) based on the volatility basis set approach from major anthropogenic and biogenic volatile organic compound precursors. A full description of WACCM6 is provided by Gettelman et al. (2019).

**CESM2-WACCM-FV2:** The CESM2-WACCM-FV2 model is based on CESM2-WACCM but with a reduced horizontal resolution for the atmosphere and land components of 2.5° longitude by 1.8° latitude. The three tuning parameters adjusted for CESM2-FV2 were set consistently for CESM2-WACCM-FV2. The dust emission factor was also changed from 0.7 to 0.26. In addition, several WACCM-specific adjustments were made for model consistency at FV2 resolution. The efficiency associated with convective gravity waves from the Beres scheme for deep convection (effgw_beres_dp) was changed from 0.5 to 0.1. The frontogenesis function critical threshold (frontgfc) was changed from 3.0 to 1.25. The background source strength used for waves from frontogenesis (taubgnd) was changed from 2.5 to 1.5. The multiplication factor applied to the lightning $NO_x$ production (lght_no_prd_factor) was changed from 1.5 to 1.0. Finally, the QBO is nudged in CESM2-WACCM-FV2, while it is self-generating in CESM2-WACCM.

**CNRM-CM6-1:** The CNRM-CM6-1 model, developed by the Centre National de Recherches Météorologiques, is a global climate model which uses a linearised scheme to model stratospheric ozone, in which ozone mixing ratios are treated as a prognostic variable with photochemical production and loss rates computed from its associated Earth System Model CNRM-ESM2-1. The model does not include interactive tropospheric ozone chemistry. Details of the linearization of the net photochemical production in the ozone continuity equation are provided by Michou et al. (2019). Tropospheric ozone mixing ratios are not calculated interactively, and are instead prescribed from the CMIP6 dataset. Methane oxidation is parameterized throughout the model domain by the introduction of a simple relaxation of the upper-stratospheric moisture source due to methane oxidation Untch and Simmons (1999). A sink representing photolysis in the mesosphere is also included. A full description and evaluation of the CNRM-CM6-1 model is provided by Voldoire et al. (2019).

**CNRM-ESM2-1:** The CNRM-ESM2-1 model, developed by the Centre National de Recherches Météorologiques, is a coupled Earth System model. The chemistry scheme of CNRM-ESM2-1 is an on-line scheme in which the chemistry routines are part of the physics of the atmospheric climate model and are called at each time-step (Michou et al., 2011). The scheme considers

168 chemical reactions, among which 39 are photolysis reactions and 9 represent the heterogeneous chemistry. This scheme is applied in the whole atmosphere above 560 hPa, but does not include tropospheric ozone non-methane hydrocarbon chemistry. The 3D concentrations of several trace gases interact with the atmospheric radiative code at each call of the radiation scheme. In addition to the non-orographic gravity wave drag parameterization, a sponge layer is also used in the upper levels to reduce spurious reflections of vertically propagating waves from the model top. This parameterization consists simply of a

linear relaxation of the wind towards zero. The linear relaxation is active above 3 Pa. A full description and evaluation of the CNRM-ESM2-1 model is provided by Seferian et al. (2019), while an evaluation of the ozone radiative forcing is detailed in Michou et al. (2019).

**E3SM-1-0 and E3SM-1-1:** The E3SM-1-0 and E3SM-1-1 models, developed by the U.S. Department of Energy, are coupled Earth System Models. They both use a simplified, linearized ozone photochemistry scheme to predict stratospheric ozone changes (Linoz v2; Hsu and Prather, 2009). Stratospheric water vapor does not include a source from methane oxidation. The

270 E3SM-1-1 model simulations differ from the E3SM-1-0 primarily by including active prediction of land and ocean biogeochemistry responses to historical increases in atmospheric $CO_2$. This is in contrast to the E3SM-1-0 simulations, which used a prescribed phenology for land vegetation and did not simulate ocean biogeochemistry. The ocean biogeochemistry has no physical feedbacks in E3SM-1-1, but the land biogeochemistry impacts surface water and energy fluxes, leading to minor

changes in the atmospheric circulation. A full description of the E3SM-1-0 model is provided by Golaz et al. (2019), while a complete description of E3SM-1-1 is provided in Burrows et al. (2020).

**FGOALS-g3:** The FGOALS-g3 model, developed by the Chinese Academy of Sciences, is a coupled ocean–atmosphere model. FGOALS-g3 does not include an interactive chemistry module, and ozone is prescribed in the stratosphere and troposphere following the recommendations by CMIP6. Stratospheric water vapor concentrations are prognostic values

calculated in a similar way to those in the troposphere. A full description and evaluation of the FGOALS-g3 model is provided by Li et al. (2019).

**GFDL-CM4:** The GFDL-CM4 model, developed by the National Oceanic and Atmospheric Administration's Geophysical Fluid Dynamics Laboratory, is a coupled ocean–atmosphere model. Ozone is prescribed using the recommended CMIP6 dataset throughout the troposphere and stratosphere, while stratospheric water vapor is interactive, but does not include a

285 source from methane oxidation. A full description and evaluation of the GFDL-CM4 model is provided by Held et al. (2019).

**GFDL-ESM4:** The GFDL-ESM4 model, developed by the National Oceanic and Atmospheric Administration's Geophysical Fluid Dynamics Laboratory, is a fully coupled chemistry-climate model. Stratospheric ozone is calculated using an interactive tropospheric and stratospheric gas-phase and aerosol chemistry scheme. The atmospheric component (AM4.1) includes 56 prognostic (transported) tracers and 36 diagnostic (non-transported) chemical tracers, with 43 photolysis reactions, 190 gas-

290 phase kinetic reactions, and 15 heterogeneous reactions. The tropospheric chemistry includes reactions for the $NO_x$-$HO_x$-$O_x$-

CO-CH$_4$ system and oxidation schemes for other non-methane volatile organic compounds. The stratospheric chemistry accounts for the major ozone loss cycles (O$_x$, HO$_x$, NO$_x$, ClO$_x$, and BrO$_x$) and heterogeneous reactions on liquid and solid stratospheric aerosols (Austin et al., 2013). Photolysis rates are calculated interactively using the FAST-JX version 7.1 code, accounting for the radiative effects of simulated aerosols and clouds. Details on the chemical mechanism will be included in Horowitz et al. (in prep). A full description and evaluation of the GFDL-ESM4 model is provided by Dunne et al. (2019).

**IPSL-CM6A-LR:** The IPSL-CM6A-LR model, developed by the Institut Pierre-Simon Laplace, is a coupled atmosphere-land-ocean-sea ice model. Stratospheric and tropospheric ozone is prescribed using the CMIP6 dataset but implemented so that profiles are stretched in a thin region (few kilometres only) around the tropopause, ensuring that the tropopause of the ozone climatology and that of the model match. Differences in tropopauses heights would lead to spurious ozone transport between the upper troposphere and lower stratosphere, a region where the corresponding non-physical radiative impact would be particularly high (e.g. Hardiman et al., 2019). Stratospheric methane oxidation is not included in the version of the model evaluated here. A full description and evaluation of the IPSL-CM6A-LR model is provided by Servonnat et al. (2020).

**MPI-ESM-1-2-HAM:** The MPI-ESM-1-2-HAM model, developed by the HAMMOZ consortium involving researchers from ETH Zurich (Switzerland), the Center for Climate Systems Modeling Zurich (Switzerland), the University of Oxford (United Kingdom), the Finish Meteorological Institute Kuopio (Finland), the Max-Planck-Institute for Meteorology Hamburg (Germany), Forschungszentrum Jülich (Germany), GEOMAR Helmholtz-Centre for Ocean Research Kiel (Germany), and the Leibniz Institute for Tropospheric Research (TROPOS) Leipzig (Germany) uses the Max-Planck-Institute Earth System Model version 1.2 (Mauritsen et al., 2019) as the basic ESM which is coupled to the aerosol microphysics package HAM including parameterizations for aerosol-cloud interactions. ECHAM6-HAM, the atmospheric part of the model has a spectral core and is run at a horizontal resolution of T63 (approx. 1.875° latitude x 1.875° longitude) and with 47 levels in the vertical domain up to 0.01 hPa (approx. 80 km). The abundance of ozone in the atmosphere is prescribed following the CMIP6 data set. Water vapor is a prognostic variable including a parameterization of methane oxidation and photolysis in the stratosphere and mesosphere. A full description of the model is provided by Tegen et al. (2019) and Neubauer et al. (2019).

**MPI-ESM1-2-HR and MPI-ESM1-2-LR:** The MPI Earth System Model (MPI-ESM) version 1.2, developed by the Max Planck Institute for Meteorology in Germany, is a global coupled atmosphere-land-ocean-sea ice-ocean biogeochemistry model. The atmosphere component, ECHAM6, uses spectral dynamics with a model top at 0.01 hPa. MPI-ESM is provided for two configurations (Müller et al., 2018; Mauritsen et al., 2019). In HR configuration, the atmospheric longitude/latitude resolution is 0.9375° x 0.9375° (T127) with 95 levels in the vertical, in LR it is 1.875° x 1.875° (T63) with 47 levels. For LR, the land surface uses dynamic land cover change instead of prescribed vegetation maps. For both configurations, ozone is prescribed on all levels following the CMIP6 recommendations. CH$_4$, N$_2$O, CO$_2$, CFC-11, and CFC-12 are prescribed using annual global means provided by CMIP6. Methane oxidation and photolysis of water vapor are parameterized for the stratosphere and mesosphere (Schmidt et al., 2013, cf. Section 2.1.2).

**MRI-ESM2-0:** The MRI-ESM2-0 model, developed by the Meteorological Research Institute, Japan Meteorological Agency, is a fully coupled global climate model which includes interactive chemistry. MRI-ESM2-0's chemistry component is the MRI-CCM2.1 module, which simulates the distribution and evolution of ozone and other trace gases in the troposphere and middle atmosphere. MRI-CCM2.1 is an updated version of MRI-CCM2 (Deushi and Shibata, 2011), which calculates a total of 90 chemical species and 259 chemical reactions. MRI-ESM2-0 simulates the stratospheric water vapor interactively with consideration for production of water vapor from $CH_4$ oxidation. A full description and evaluation of the MRI-ESM2-0 model is provided by Yukimoto et al. (2019).

**NorESM2-MM:** The second version of the Norwegian Earth System Model (NorESM2-MM) developed by the Norwegian Climate Center (NCC) is based on CESM2. However, NorESM2-MM uses a different ocean and ocean bio-geochemistry model, and the atmosphere component of NorESM2-MM, CAM-Nor, employs a different module for aerosol physics (including interactions with clouds and radiation) and includes improvements in the formulation of local dry and moist energy conservation, in the local and global angular momentum conservation, and in the computation for deep convection and air-sea fluxes. The surface components of NorESM2-MM have also minor changes in the albedo calculations and to land and sea-ice models. Similar to CESM2, NorESM2 prescribes ozone (zonally averaged fields, 5-day frequency) and stratospheric water vapor production from $CH_4$ oxidation (3D fields, monthly frequency) derived from runs of CESM2-WACCM6. A full description of the NorESM2-MM model is provided by Seland et al. (2020).

**SAM0-UNICON:** The SAM0-UNICON, developed by the Seoul National University, is a general circulation model based on the CESM1 model with a Unified Convection Scheme (Park 2014a, b) that replaces shallow and deep convection schemes in CESM1. Stratospheric and tropospheric ozone is prescribed as a monthly mean 3D field, taken from the CMIP6 ozone dataset, with a specified annual cycle. Stratospheric water vapor does not include a source from methane oxidation. A full description of the SAM0-UNICON model is provided by Park et al. (2019).

**UKESM1-0-LL:** The UKESM1-0-LL model, developed jointly by the United Kingdom's Met Office and Natural Environment Research Council, is a fully coupled Earth System Model. UKESM1-0-LL uses a combined troposphere-stratosphere chemistry scheme (Archibald et al., 2019), which includes 84 tracers, 199 bimolecular reactions, 25 uni- and termolecular reactions, 59 photolytic reactions, 5 heterogeneous reactions and 3 aqueous phase reactions for the sulfur cycle. As a result, stratospheric ozone and water vapor are fully interactive. A full description and evaluation of the UKESM1-0-LL model is provided by Sellar et al. (2019).

## 2.2 Simulations

To evaluate changes in stratospheric ozone and water vapor from 1850 to 2100, this study makes use of two types of simulations performed as part of the wider CMIP6 activity: the CMIP6 historical simulation (Eyring et al., 2016) and the ScenarioMIP future simulations (O'Neill et al., 2015).

The CMIP6 historical simulation runs from 1850 to 2014, in which the models are forced by common datasets based on observations which include historical changes in short-lived species (e.g. $NO_x$, CO, and VOCs) and long-lived GHGs and ODSs (e.g. $CO_2$, $CH_4$, $N_2O$, CFC-11, and CFC-12), global land use, solar forcing, stratospheric aerosols from volcanic eruptions and, for models without ozone chemistry, prescribed time varying ozone concentrations. These simulations are initialised from the pre-industrial control (piControl) simulation, a time-slice simulation run with perpetual 1850 pre-industrial conditions performed by each model.

The ScenarioMIP future simulations run from 2015 to 2100 and follow the newly developed SSPs, which provide future emissions and land use changes based on scenarios directly relevant to societal concerns regarding climate change impacts, adaptation and mitigation (Riahi et al., 2017). Broadly, the shared socioeconomic pathways follow 5 categories: sustainability (SSP1), middle of the road (SSP2), regional rivalry (SSP3) inequality (SSP4) and fossil-fuelled development (SSP5). Further, each scenario has an associated forcing pathway (i.e. the forcing reached by 2100 relative to the pre-industrial), and each specific scenario is referred to as SSPx-y, where x is the SSP and y is the radiative forcing pathway (the radiative forcing at the end of the century, in $Wm^{-2}$). For example, SSP3-7.0 follows SSP3 (regional rivalry), and has a 2100 global mean forcing of 7.0 $Wm^{-2}$ relative to the pre-industrial.

The SSP scenarios span a broad range of future emissions and land use changes, both of which have the potential to change total column ozone (TCO) through changes in both the troposphere and/or stratosphere, and stratospheric water vapor through changes in tropical tropopause layer (TTL) temperatures or $CH_4$. In general, low numbered SSPs (i.e. SSP1 and SSP2) assume lower abundances of long-lived GHGs ($CO_2$, $CH_4$, $N_2O$; Meinshausen et al., 2019) and lower emissions of ozone precursors (Hoesly et al., 2018). All SSPs follow the same emissions scenario for ozone depleting substances, based on continued compliance with the Montreal Protocol (Velders and Daniel, 2014), but the concentrations of ODSs vary slightly between scenarios due to changes in the lifetimes of each species associated with climate change (Meinshausen et al., 2019). It should be noted that recent studies have identified unreported emissions of CFC-11 (e.g. Montzka et al., 2018), and that the trajectory of ozone recovery is sensitive to the magnitude and duration of these emissions (e.g. Dhomse et al., 2019; Keeble et al., 2019), which are not included in the emissions assumptions of Velders and Daniel (2014). In this study, we use ozone and water vapor output for the SSP1-1.9, SSP1-2.6, SSP2-4.5, SSP3-7.0, SSP4-3.4, SSP4-6.0 and SSP5-8.5 scenarios. Details of which simulations were performed by each model are provided in Table 1.

Note that several models have performed a number of ensemble members for each of the simulations used in this study. In the analysis presented here, an ensemble mean is created for each model, and this ensemble mean is evaluated in this study and shown in each of the figures. The CMIP6 multi-model mean (MMM) is created by meaning across the individual CMIP6 model ensemble means, so that each model is weighted equally towards the CMIP6 MMM, preventing a model which has performed many ensemble members from dominating the MMM. The only area in which individual ensemble members are

considered separately is in the calculation of the TCO trends presented in Section 3.1.2, in which the individual ensemble members are used when calculating the statistical significance of the TCO trends.

## 2.3 Observation datasets

The evaluation of stratospheric ozone and water vapor makes use of two datasets: the NIWA-BS combined TCO database and SWOOSH zonal mean ozone and water vapor datasets.

Version 3.4 of the National Institute of Water and Atmospheric Research - Bodeker Scientific (NIWA-BS) combined TCO database takes daily gridded TCO fields from 17 different satellite-based instruments, bias corrects them against the global Dobson and Brewer spectrophotometer network, and merges them into a seamless homogeneous daily gridded (1.25° longitude x 1.0° latitude) TCO data record (see Bodeker et al., 2020). First, overpass data from the Total Ozone Mapping Spectrometer (TOMS) instruments flown onboard Nimbus-7, Meteor-3, Earth Probe, and Adeos, and from the Ozone Monitoring Instrument

(OMI) instrument on Aura, are bias corrected against the ground-based TCO measurements. Those five bias-corrected datasets then provide the basis for correcting the remaining datasets i.e. those from the Global Ozone Monitoring Experiment (GOME), GOME-2 and SCanning Imaging Absorption SpectroMeter for Atmospheric CHartographY (SCIAMACHY) instruments, the Solar Backscatter Ultraviolet Radiometer (SBUV) instrument flown on Nimbus-7, and the SBUV-2 instruments flown on NOAA-9, 11, 14, 16, 17, 18 and 19. The bias corrected measurements are then combined in a way that traces uncertainties

from the source data through to the final merged data product.

The Stratospheric Water and OzOne Satellite Homogenized (SWOOSH) dataset is a merged record of stratospheric ozone and water vapor measurements collected by a subset of limb sounding and solar occultation satellites spanning 1984 to the present (see Davis et al., 2016, for details). Specifically, SWOOSH comprises data from the Stratospheric Aerosol and Gas Experiment instruments (SAGE-II and SAGE-III/Meteor-3M), the Upper Atmosphere Research Satellite HAlogen Occultation Experiment

(UARS HALOE), the UARS Microwave Limb Sounder (MLS), and Aura MLS.

The source satellite measurements are homogenized by applying corrections that are calculated from data taken during time periods of instrument overlap. The primary SWOOSH product is a merged multi-instrument monthly-mean zonal-mean (10° latitudinal resolution) dataset on the pressure grid of the Aura MLS satellite (12 levels per decade). Because the merged product contains missing data, a merged and filled product is also provided for studies requiring a continuous dataset. These merged

and filled products for ozone and water vapor (combinedanomfillo3q and combineanomfillh2oq respectively) from SWOOSH version 2.6 are used in this study for comparison with CMIP6 model fields.

## 3 Ozone

### 3.1 Evaluation over recent decades

Before investigating long-term changes in stratospheric ozone, we evaluate each model's performance, and the performance
of the CMIP6 MMM, against observations. In the following Sections we evaluate the 2000-2014 climatological zonal mean distribution of ozone and the seasonal evolution of zonal mean total column ozone against observations, in the form of the combined zonal mean ozone dataset from SWOOSH and the TCO dataset from NIWA-BS.

### 3.1.1 2000-2014 climatological zonal mean and total column ozone

Latitude-height cross sections of zonal mean ozone volume mixing ratios for each CMIP6 model, the CMIP6 MMM, the
420 CMIP6 ozone dataset used by models prescribing ozone mixing ratios, and the SWOOSH dataset, averaged over the years 2000-2014, are shown in Figure 1. There is generally good agreement between the individual CMIP6 models and the SWOOSH dataset. All models broadly capture tropospheric and stratospheric ozone gradients, with a clear peak in ozone mixing ratios in the tropical stratosphere at around 10 hPa, the downwards bending of the contour lines towards high latitudes in the lower stratosphere (e.g. Plumb, 2002) and flat contour lines in the tropical upper stratosphere in the quasi-equilibrated photochemical
regime (e.g. Haigh and Pyle, 1982; Meul et al., 2014; Chiodo et al., 2018; Nowack et al., 2018).

Notable differences between the models occur in the uppermost stratosphere, and around the tropopause (Figure A1). In the upper stratosphere, the BCC-ESM1, CESM2, CESM2-FV2, FGOALS-g3, NorESM2-MM and SAM0-UNICON models all simulate much higher ozone mixing ratios than the CMIP6 MMM (see Figure A1). Additionally, the BCC-ESM1 and SAM0-UNICON models also have a different spatial structure in the distribution of ozone at these levels, with maxima in the mid-
430 latitudes at 1 hPa (see Figure 1). However, note that these models have lower model tops than the 1 hPa maximum altitude of the CMIP6 data request, and so these differences arise from interpolation to the pressure levels of the CMIP6 data request.

In the tropical tropopause region, the MRI-ESM2-0 and UKESM1-0-LL models significantly overestimate ozone mixing ratios, while the SAM0-UNICON model has much lower mixing ratios in this region with respect to the CMIP6 MMM. The tropical tropopause is a region in which chemistry-climate models have typically performed poorly, due to the fact that ozone
mixing ratios in this region are controlled by a combination of chemical production, vertical transport of ozone poor air from the troposphere and mixing of ozone rich stratospheric air. Gettelman et al. (2010) documented the seasonal cycle of ozone at 100 hPa from 18 models involved in the CCMVal-2 inter-comparison project and showed that while there is good agreement

between the MMM and the observations, there is a large spread in ozone mixing ratios between individual models, and many models do not accurately capture the observed seasonal cycle. For the CMIP6 models investigated here, there is also good agreement between the climatological (2000-2014), tropical (15°S-15°N) MMM ozone mixing ratios at 70 hPa and the SWOOSH dataset (Figure 2), both for the absolute ozone mixing ratios and the amplitude of the seasonal cycle. Many CMIP6 models accurately capture the seasonal cycle of ozone, with lower ozone mixing ratios simulated between February and April, and higher values in August and September. However, as with CCMVal-2 models, there is a large spread in modelled ozone mixing ratios in the tropical tropopause region. Both MRI-ESM2-0 and UKESM1-0-LL are high biased, while BCC-ESM1 and SAM0-UNICON are low biased compared to the observations and the CMIP6 MMM.

TCO climatologies (latitude vs month), averaged over the years 2000-2014, for the individual CMIP6 models, the CMIP6 MMM, the CMIP6 ozone dataset, and the NIWA-BS dataset are shown in Figure 3. Overall, the observed climatology patterns and annual cycle amplitudes, compared here against the NIWA-BS dataset, are well represented in the CMIP6 MMM and the individual models: lower values and smallest amplitude in the tropics that increase to the poles, with the highest TCO values around 60°S between August and November, and in the NH polar regions between January and May, and the smallest TCO values in the SH polar regions during the ozone hole period. However, despite this good qualitative agreement between the CMIP6 models and the NIWA-BS observational dataset, there is significant variation between individual CMIP6 models with respect to the CMIP6 MMM (Figure A2). CNRM-CM6-1 and CNRM-ESM2-1 underestimate TCO in the polar regions, while overestimate TCO in the tropics, while the MRI-ESM2-0 and UKESM1-0-LL overestimate TCO globally. Of particular note are the MRI-ESM2-0 and SAM0-UNICON models, which have large positive TCO anomalies with respect to the CMIP6 MMM at high southern latitudes in the spring, indicating they underestimate Antarctic polar ozone depletion.

Despite the differences between the individual CMIP6 models, there is generally good agreement between the zonal mean distribution of ozone in the CMIP6 MMM and the SWOOSH dataset throughout much of the stratosphere (Figure 4), with differences between 70 hPa and 3 hPa typically less than ±15%. Maximum ozone mixing ratios at ~10 hPa are slightly underestimated by the CMIP6 MMM, while ozone mixing ratios in the lower tropical stratosphere, and at ~1 hPa in the mid-latitudes, are overestimated (consistent with the analysis shown in Figure 2). The CMIP6 MMM also overestimates ozone mixing ratios at all latitudes in the upper troposphere between 200-100 hPa by ~20-40 %. Since upper tropospheric ozone is a particularly important climate forcing agent (Lacis et al., 1990; Stevenson et al., 2013; Young et al., 2013; Nowack et al., 2015; Banerjee et al., 2018), this has important implications for the ozone radiative forcing estimated from climate model simulations. However, it should be noted that the uncertainties in the SWOOSH dataset are likely to be relatively large in the upper troposphere.

The lower row of Figure 4 shows the TCO differences between the CMIP6 MMM and the NIWA-BS. In the tropics and the NH mid-latitudes the differences are smaller than ±10 DU (<5% of the climatological value in these regions). The differences get slightly larger in the NH polar regions, but are largest in the SH mid- and high latitudes where the MMM overestimates

the observed TCO by up to 25 DU throughout much of the year, except during late spring when the CMIP6 MMM underestimates TCO up to 35 DU. Compared to CMIP5, the differences in the NH mid-latitudes and polar regions seem reduced in the MMM (Eyring et al., 2013; Lauer et al., 2017), whereas the differences between the MMM and observations are similar in the SH mid-latitudes between CMIP5 and CMIP6.

Figure 5 shows Taylor diagrams (Taylor, 2001) of the CMIP6 models performance for annual and seasonal mean TCO between

475 60°S–60°N against the NIWA-BS TCO dataset for the period 2000–2014. Taylor diagrams provide a statistical summary of how similar spatial patterns are between model simulation and observational data and have been widely used to test various aspects of model performance. In the Taylor diagram, the 'correlation' represents the spatial correlation between model and NIWA-BS dataset. The standard deviation is calculated as follows:

$$standard\ deviation = \frac{\sum_i (m_i - model\ mean)2}{\sum_i (m_i - observation\ mean)2}$$

while the bias is defined as follows:

$$bias = \frac{(model\ mean - observation\ mean)}{observation\ mean} \times 100$$

On the annual scale, the 22 CMIP6 models evaluated here can generally reproduce the spatial pattern of NIWA-BS TCO, with correlation coefficients greater than 0.9 for all models. The standard deviations of all the models fall between 0.75 and 2.20. With the exception of CNRM-CM6-1, most CMIP6 models have a positive bias in deviation compared with the NIWA-BS

dataset. On the seasonal scale, the models perform better during DJF and MAM than during JJA and SON. The CNRM-CM6-1and CNRM-ESM2-1 models falls closest to the reference line and also have relatively small biases compared with the NIWA-BS dataset, while in contrast, the MRI-ESM2-0 model has a larger standard deviation than other models. Both the MRI-ESM2-0 and UKESM1-0-LL models have large biases compared to the NIWA-BS dataset.

### 3.1.2 Regional Total Column Ozone changes 1960-2014

The temporal evolution of TCO in observations and the individual CMIP6 models, for both the global and regional averages, is shown in Figure 6. In general, the MMM overestimates the observed TCO values by up to 6% (10-20 DU) globally (Figure

6a), in the NH and SH mid-latitudes, and in the tropics (Figure 6c-e), but the trend in these regions is well captured. The spread within the analysed CMIP6 models is large, though, with MRI-ESM2-0 and UKESM1-0-LL overestimating TCO by up to 40 DU in, for example, the SH mid-latitudes. Both of these models used an interactive chemistry scheme to calculate ozone abundances in the troposphere and stratosphere. Although these two models overestimate ozone, the other models that calculate ozone fields interactively (CESM2-WACCM, CESM2-WACCM-FV2, CNRM-ESM2-1, and GFDL-ESM4) slightly underestimate the observed TCO values, indicating that there is no clear distinction of models with and without interactive chemistry as there was in the CMIP5 models (Eyring et al., 2013). While most of the analysed CMIP6 models show distinct interannual variability, it is noteworthy that there is no interannual variability detectable for three models (CESM2, CESM2-FV2 and SAM0-UNICON) that used prescribed ozone fields for their historical simulations (see Section 2.1).

The CMIP6 MMM underestimates the observed decline in TCO for March in the NH polar regions during the ozone depletion period (1980-2000) but tracks the observations well after 2000 (Figure 6f). This is also mirrored in the trends calculated for these periods for observations and the individual CMIP6 models and MMM (Table 2). While the TCO October values in the SH polar regions are overestimated in the MMM compared to observations, the decrease in TCO between 1980 and 2000 is stronger in the models than in the observations (see Table 2). These characteristics are very similar to the trends reported in Eyring et al. (2013) for the CMIP5 model simulations.

Table 2 follows Eyring et al. (2013, their Table 2) in showing the observed and modelled trends in TCO over the period 1980-2000. Additionally, we also show trends over the period of 2000-2014. They are calculated for 22 models, of which six models have interactive stratospheric chemistry (CESM2-WACCM, CESM2-WACCM-FV2, CNRM-ESM2-1, GFDL-ESM4, MRI-ESM2-0, and UKESM1-0-LL). For these six models, over the period 1980-2000, annual mean global trends range from -0.19 (MRI-ESM2-0) to -1.04 (UKESM1-0-LL) DU/year. The average trend from these models (hereafter referred to as "INTERACTIVE") is -0.57 DU/year which is within the uncertainty range of observed trends (from -0.56 to -0.74 DU/year). In the tropics, all models show weak negative trends, and the mean of the 6-member INTERACTIVE models in the tropics (-0.17 DU/year) compares well with the observed trends. In the northern mid-latitudes, models considerably underestimate the observed negative trends with the exception of CNRM-ESM2-1 (-0.95 DU/year – within the range of observed trends) and UKESM1-0-LL (-1.2 DU/year – slightly stronger than the observed negative trends). Over southern mid-latitudes, modelled trends are generally closer to the observed trends, with the exceptions of MRI-ESM2-0 and SAM0-UNICON which have substantially weaker negative trends. Again, UKESM1-0-LL overestimates the observed negative trends. At NH high latitudes, most of the models substantially underestimate the observed negative trends there, with the exception of CNRM-ESM2-0 and UKESM1-0-LL, with the latter having again a more negative trend than observed. This indicates that most CMIP6 models, particularly those prescribing stratospheric ozone fields, underestimate Arctic springtime ozone depletion. In contrast, at SH high latitudes, all models calculate large (in absolute terms) negative trends, indicating that Antarctic ozone depletion is having

a pronounced impact on ozone in the CMIP6 models. Overall, the INTERACTIVE models have larger trends than those models without interactive chemistry.

Over the period 2000-2014, many models show non-significant (at the 95% confidence level) positive trends in TCO. However, nine models show significant, albeit weak, positive trends in global TCO, of which three are INTERACTIVE models (CESM2-WACCM, MRI-ESM2-0, and UKESM1-0-LL). The significant positive trends calculated in these models show the largest positive trends in both the NH and the SH high latitudes and moderate positive trends in mid-latitudes (Table 2). The INTERACTIVE models collectively show stronger positive trends in all regions, compared to the MMM. Significant and the strongest positive ozone trends in the SH high latitudes occur in MRI-ESM2-0, NorESM2-MM, and UKESM1-0-LL, whereas significant and the strongest positive trends in the NH high latitudes occur in CESM2-WACCM, NorESM2-MM, and UKESM1-0-LL. Significant but weaker positive trends also occur in SAM0-UNICON, CESM2 and CESM2-FV2 for October monthly mean trends at SH high latitudes, although the statistical significance of the trends in these three models is a consequence of the small interannual variability identified in Figure 6.

### 3.2 Long-term evolution of Total Column Ozone from 1850-2014

The regional evolution of zonal mean, annual mean TCO for the CMIP6 MMM from 1850 to 2100 is shown in Figure 7. For the near global mean (60°S-60°N), TCO increases slowly from ~298 DU in 1850 to ~304 DU in 1960, before rapidly declining through the 1980s and 1990s due to emissions of halogenated ODSs, reaching a minimum in the late 1990s. The increases in TCO between 1850 and 1960 are more prominent in the NH and tropics, while the decreases at the end of the 20th century are stronger in the SH. In the NH, TCO values increase by 10-15 DU between 1850 and 1960. This increase in TCO is larger than the TCO depletion that occurs from 1960 to 2000 in response to the emission of halogenated ODSs, resulting in higher NH mid-latitude TCO values in the late 1990s than in the pre-industrial. In contrast, SH TCO values remain relatively constant from 1850 to 1960, before rapidly declining throughout the 1970s and 1980s. The distinctive 11-year solar cycle in TCO is superimposed on these long-term trends. In addition, the eruption of Mt. Krakatoa in 1883 can be clearly seen as an increase in TCO of around 3-5 DU, resulting in the highest TCO values for ~100 years between 1850 and 1950.

There is poor agreement in the simulation of pre-industrial TCO across CMIP6 models, which vary between 275 and 340 DU (Figure A3). The UKESM1-0-LL and MRI-ESM2-0 models have particularly high TCO, while the GFDL-CM4 values are lowest. Surprisingly, there is a ~20 DU range in pre-industrial TCO values between those models prescribing the CMIP6 ozone dataset, suggesting that TCO is not conserved after model implementation of the CMIP6 ozone dataset. When TCO values from each CMIP6 model are normalised to the 1960 annual mean value (Figure A4), there is a smaller difference between the modelled pre-industrial TCO values, which cover ±5 DU around the MMM for the global mean (60°S-60°N). When the models are normalised to the 1960 annual mean, it is also clear that, compared to the CMIP6 MMM, the CNRM-CM6-1, CNRM-ESM2-1 and UKESM1-0-LL models have much stronger ozone declines during the period of halogenated ODS emissions (at

all latitude ranges for the UKESM1-0-LL model and in the NH for the CNRM-CM6-1 and CNRM-ESM2-1 models), while the MRI-ESM2-0 model has much weaker TCO declines during this time. It is also clear that the CESM2-WACCM-FV2 model has higher interannual variability in TCO values in the tropics and NH than other CMIP6 models.

Zonal mean, annual mean partial ozone columns for the full stratosphere, upper stratosphere and lower stratosphere, averaged over 90°S-90°N, for a subset of the models are shown in Figure 8. These partial column values indicate that stratospheric ozone, in both the lower and upper stratosphere, did not change significantly between 1850 and 1960, suggesting that the increases in TCO seen in Figure 7 arise from changes in tropospheric ozone. It is also clear from Figure 8 that much of the high TCO bias for the UKESM1-0-LL model (Figure A3) comes from elevated stratospheric ozone mixing ratios, rather than a large tropospheric ozone bias. Figure 8 also shows projections of the full stratospheric, upper stratospheric and lower stratospheric partial columns from 2015-2100 following the SSP3-7.0 scenario. For the three models which have projections of the stratospheric partial column from 2015-2100 (CESM2-WACCM, GFDL-ESM4 and UKESM1-0-LL), upper stratospheric column ozone is projected to rapidly return to the 1960 historic values (defined here as the 1955-1965 multi-annual mean), reaching these values between 2030-2050, and by the year 2100 upper stratospheric ozone partial column values are projected to be larger than at any period between 1850 to 2100. In contrast, the lower stratospheric column returns to the 1960 value later in the 21$^{st}$ century (between 2080-2090) for the CESM2-WACCM and GFDL-ESM4 models, and in the case of the UKESM1 model the lower stratospheric ozone partial column does not return to the 1960 historic value during the 21$^{st}$ Century. The combined effect of these changes is that while the CESM2-WACCM and GFDL-ESM4 models project the 2086-2100 multi-annual mean stratospheric ozone column to slightly exceed the 1960 historic value (by +6 DU in the case of CESM2-WACCM and +3 DU in the case of GFDL-ESM4), the UKESM1-0-LL model projections indicate that stratospheric column ozone will not return to the 1960 historic value by the end of the 21$^{st}$ century (the 2086-2100 multi-annual mean being 5 DU lower than the 1960 historic value for this model).

Climatological differences between the present day (2000-2014) and the preindustrial (1850-1864) zonal mean ozone mixing ratios and TCO values are shown in Figure 9. The expected general decrease in stratospheric ozone due to ODS-induced stratospheric ozone depletion (e.g. Iglesias-Suarez et al., 2016) as well as a general increase in ozone in the troposphere due to the emission of ozone precursors (e.g. Stevenson et al., 2013; Young et al., 2013, 2018) are clearly captured by the CMIP6 MMM. The historical decrease in stratospheric ozone is most pronounced in the SH polar vortex, with the maximum TCO decrease during SH spring due to the role of heterogeneous activation of chlorine reservoir species on polar stratospheric clouds (e.g. Solomon et al., 1999). In absolute values, the decreases in stratospheric ozone mixing ratios dominate over the larger fractional tropospheric ozone changes in terms of the integrated number of ozone molecules in a vertical column, leading to historically globally reduced TCO. However, there is a pronounced seasonal cycle in these changes and a clear difference between the hemispheres, with widespread ozone decreases throughout the year in the SH but small TCO increases at high latitudes in the NH during the summer and autumn.

## 3.3 Long-term evolution from 2015-2100

From 2015 onwards, models follow the assumptions made in the various SSP scenarios. Many models which performed the historical simulations also provided ozone data from SSP1-2.6, SSP2-4.5, SSP3-7.0 and SSP5-8.5, while a much smaller number provided data from SSP1-1.9, SSP4-3.4 and SSP4-6.0 (see Table 1 for an overview of which models performed which SSP scenarios). Due to the different numbers of models performing each scenario, the MMM for each SSP is normalised to the 2014 value to produce one smooth dataset and allow for comparison between the trajectories of TCO under each SSP scenario.

Zonal mean, annual mean CMIP6 MMM TCO, averaged over 60°S-60°N, is projected in the simulations evaluated here to follow three main trajectories from 2015 to 2100 (Figure 7). Under SSP2-4.5, SSP4-3.4 and SSP4-6.0 TCO values are projected to return to their 1960's values by the middle of the 21$^{st}$ century, while under the SSP3-7.0 and SSP5-8.5 scenarios TCO values are projected to significantly exceed the 1960's values throughout the latter half of the 21$^{st}$ century. Despite the assumption that halogenated ODSs will continue to decline due to the Montreal Protocol, TCO values are not projected to return to the 1960's values under the SSP1-1.9 and SSP1-2.6 scenarios.

As with 60°S-60°N, SSP pathways which assume higher radiative forcing result in higher TCO at the end of the century for most latitude ranges (Figure 7), although the timing of the return of TCO to 1960's values varies. Annual mean TCO values at high southern latitudes are only projected to return to the 1960's values in the SSP3-7.0 and SSP5-8.5 scenarios. Conversely, TCO is projected to return to, and in most cases exceed, the 1960 annual mean value in all SSPs in the high northern latitudes. In the mid-latitudes, projected TCO values follow a pattern similar to those seen in the near global mean, although in the NH TCO is projected to return to the 1960s value in the SSP1-2.6 scenario and exceed the 1960 value under most other scenarios. Interestingly, in the NH mid- and high latitudes the 1980 annual mean TCO is larger than that of the 1960 annual mean, and as a result NH TCO values are projected to return to the 1980 values after returning to 1960 values. TCO projections in the tropics are quite different to those at other latitudes, with return to 1960's values only projected to occur in SSP3-7.0 and SSP5-8.5, and under the SSP5-8.5 scenario TCO values are projected to decline again in the latter half of century (consistent with an acceleration of the BDC and resulting decreases to tropical lower stratospheric ozone, e.g. Meul et al., 2016; Keeble et al., 2017).

Climatological differences between the end of the century (2086-2100 average) and the present day (2000-2014 average) zonal mean ozone mixing ratios and TCO values are shown in Figure 10 for SSP1-2.6, SSP2-4.5, SSP3-7.0 and SSP5-8.5, calculated using 12 of the 22 CMIP6 models evaluated in this study (BCC-CSM2-MR, CESM2, CESM2-WACCM, CNRM-CM6-1, CNRM-ESM2-1, GFDL-ESM4, IPSL-CM6A-LR, MPI-ESM1-2-HR, MPI-ESM1-2-LR, MRI-ESM2-0, NorESM2-MM, and UKESM1-0-LL). These models were selected as they had performed all four of the SSP scenarios plotted, and so differences between the figures do not result from the inclusion of different models.

Under each of these SSP scenarios, ozone mixing ratios in the upper stratosphere and SH polar lower stratosphere are projected to increase, consistent with the decline in halogenated ODSs assumed in all scenarios. The magnitude of the upper stratospheric increases in ozone is larger for scenarios which assume larger increases in GHG loading of the atmosphere, due to the resulting $CO_2$-induced cooling of the stratosphere. However, significant differences between the scenarios are seen in the troposphere and tropical lower stratosphere. Under the SSP1-2.6 and SSP2-4.5 scenarios, tropospheric ozone mixing ratios are projected to decrease, consistent with the large reduction in the emission of ozone precursors assumed in these scenarios (Gidden et al., 2019). Under SSP1-2.6 the decreases in tropospheric ozone are particularly strong in the NH, while the increases in stratospheric ozone outside of the Antarctic polar lower stratosphere are smaller than in other scenarios (consistent with less $CO_2$ induced cooling), and together these factors explain why TCO does not return to 1960 values in SSP1 scenarios. Strong emissions mitigation scenarios which decrease tropospheric ozone mixing ratios and thereby help to mitigate climate change and air quality impacts, slow or prevent ozone recovery, as measured by the return of TCO return to historic values. This calls into question whether using TCO return dates as a metric for ozone recovery is entirely appropriate to evaluate the success of the Montreal Protocol, and if other metrics might not better reflect the recovery of stratospheric ozone driven by changes in stratospheric chlorine loading (as discussed by Eyring et al., 2013; WMO 2018).

In contrast, ozone mixing ratios are projected to increase throughout much of the troposphere and upper stratosphere in the SSP3-7.0 and SSP5-8.5 scenarios, explaining the projected super-recovery of TCO values in the mid- and high-latitudes under these scenarios by the end of the century. However, ozone mixing ratios are projected to be lower in the tropical lower stratosphere by the end of the century (Figure 10) due in part to the acceleration of the BDC (and the resulting decreases in tropical lower stratospheric ozone) and reduced production of ozone at these altitudes due to the thicker overhead column ozone (Eyring et al., 2013; Meul et al., 2016; Keeble et al., 2017). These lower stratospheric decreases offset the increases at higher altitudes, resulting in TCO values being lower at the end of the 21$^{st}$ century compared to the present day under most emissions scenarios, despite reductions in stratospheric halogens.

### 3.4 Comparison of models with interactive vs non-interactive ozone

A key question arising from the analysis above is whether those models using interactive chemistry schemes to model ozone differ in a consistent manner from those which prescribe ozone fields using the CMIP6 dataset. Of the 22 models evaluated in this study, only six (CESM2-WACCM, CESM2-WACCM-FV2, CNRM-ESM2-1, GFDL-ESM4, MRI-ESM2-0, and UKESM1-0-LL) use interactive stratospheric chemistry schemes while 10 (AWI-ESM-1-1-LR, BCC-CSM2-MR, BCC-ESM1, FGOALS-g3, GFDL-CM4, IPSL-CM6A-LR, MPI-ESM-1-2-HAM, MPI-ESM1-2-HR, MPI-ESM1-2-LR, and SAM0-UNICON) prescribe the CMIP6 ozone dataset. These two groups of models are used to define an 'interactive' and 'prescribed' multi-model mean. Three further models (CNRM-CM6-1, E3SM-1-0, and E3SM-1-1) use simplified ozone schemes, while

CESM2, CESM2-FV2, and NorESM2-MM prescribe ozone from previous CESM2-WACCM6 simulations. These six models are discounted from the comparison of models with interactive vs non-interactive ozone made here.

Several challenges exist when comparing CMIP6 models with interactive vs prescribed ozone. The first is the small number of models with interactive ozone. Another challenge is that while the 10 models in the prescribed model mean use the CMIP6 ozone dataset, how they implement these fields varies significantly from model to model (e.g. the vertical and horizontal regridding the fields undergo, whether the prescribed fields are adjusted to follow the model tropopause, whether the model uses the CMIP6 dataset in the stratosphere but model tropospheric ozone interactively, and how the models treat the upper boundary, where significant differences are seen in Figure A1). The result is that TCO is not conserved during this implementation process, and models ostensibly using the same prescribed ozone dataset differ by up to 20 DU throughout the historical simulation. Comparison is further complicated by the fact that the prescribed ozone fields are taken, in part, from a forerunner to the CESM2-WACCM model (in combination with fields from the CMAM model; Checa-Garcia, 2018b), and so there is overlap between one of the models with interactive chemistry and those using prescribed ozone.

Despite these caveats, some conclusions can be reached regarding the differences between models with interactive ozone and those that prescribe ozone. Models with interactive chemistry schemes, unsurprisingly, cover a much broader range of TCO values (~60 DU) from the pre-industrial to the present day than those which prescribe the CMIP6 ozone dataset. Despite this larger spread, the interactive multi-model mean agrees well (to within 4%) with the CMIP6 MMM throughout the historical period for each of the latitude regions examined here (see Figure A3), although TCO values are generally larger in the global (60°S-60°N) average by ~5 DU (~2%), owing predominantly to larger tropical TCO values (4-6 DU or ~2%) in the interactive models. Similarly, there is good agreement when climatological (2000-2014) zonal mean and total column ozone values are compared between the interactive and prescribed model means (Figure A5). For the zonal mean, throughout much of the stratosphere differences between these multi-model means are typically less than 5%, although larger differences are seen in the uppermost levels and at the tropical troposphere. Expressed as a total column difference, the interactive mean is ~10 DU higher than the prescribed mean in the tropics and midlatitudes.

However, the close agreement between the interactive and prescribed means masks significant differences between individual models with interactive ozone schemes and the CMIP6 MMM. The UKESM1-0-LL and MRI-ESM2-0 models both significantly overestimate TCO with respect to the CMIP6 MMM, while the CESM2-WACCM, CESM2-WACCM-FV2 and GFDL-ESM4 models all model lower TCO than the CMIP6 MMM (Figure A3). As a result, there is no consistent bias between the models with interactive and prescribed ozone fields, i.e., models with interactive chemistry schemes do not all have consistently higher TCO values than found in models prescribing stratospheric ozone, and instead fall either side of the prescribed multi-model mean.

Similarly, large differences to the CMIP6 MMM are modelled in the zonal mean distribution of ozone for each CMIP6 model with interactive chemistry, although no consistent bias exists across all 6 interactive models. For TCO trends, while the interactive and prescribed means agree within their associated uncertainties (Table 2), there are again large differences between the trends modelled by the interactive models. The UKESM1-0-LL and CNRM-ESM2-1 models have the strongest negative trends for the period 1980-2000 of the CMIP6 models evaluated in this study, while MRI- ESM2-0 has the weakest trend for this period.

As a result of this analysis, while it is possible to say that no consistent difference in the total ozone column abundance or trend exists between the interactive and prescribed means, this is likely a result of compensating biases from each individual model cancelling out. Certainly, some models with interactive stratospheric chemistry are clear outliers from the CMIP6 MMM and understanding the reasons for these differences is a challenge for the various modelling centres.

## 4 Stratospheric Water Vapor

As with ozone, before investigating long-term changes in stratospheric water vapor we evaluate each model's performance, and the performance of the CMIP6 MMM, against observations. In the following Sections we evaluate the 2000-2014 climatological zonal mean distribution of water vapor against the SWOOSH combined dataset and evaluate the source of stratospheric water vapor from $CH_4$ oxidation.

### 4.1 Evaluation of recent changes

Eighteen of the models used in this study provide stratospheric water vapor output from the historical simulations, with a smaller subset providing water vapor from the SSP scenarios (see Table 1). Zonal mean water vapor volume mixing ratios for each CMIP6 model, the CMIP6 MMM and the SWOOSH dataset, averaged over the years 2000-2014, are shown in Figure 11. There is relatively poor agreement between the individual CMIP6 models and the observations. The AWI-ESM-1-1-LR, CESM2, CESM2-FV2, CESM2-WACCM, CESM2-WACCM-FV2, MPI-ESM1-2-HR, MPI-ESM1-2-LR, and MRI-ESM2-0 models all capture the distribution of stratospheric water vapor mixing ratios reasonably well, with the largest percentage differences in the polar regions, most likely related to the formation and sedimentation of polar stratospheric cloud particles, and at the tropopause (see Figure A6). Several models (BCC-CSM2-MR, BCC-ESM1, E3SM-1-1, GFDL-CM4 and IPSL-CM6A-LR) do not accurately capture the increase in water vapor with altitude throughout the stratosphere, as these models do not include a representation of water vapor produced from $CH_4$ oxidation. In contrast, CNRM-CM6-1 and CNRM-ESM2-1 simulate very large changes in stratospheric water vapor between the tropical lower stratosphere and upper stratosphere, consistent with an overestimate in the water vapor production from $CH_4$ oxidation in the CNRM-ESM2-1 model (discussed below). Water vapor mixing ratios in the MPI-ESM-1-2-HAM and UKESM1-0-LL models are biased high throughout the

stratosphere compared to the SWOOSH dataset. Differences between the individual models and the CMIP6 MMM are shown in Figure A6.

Averaged across the CMIP6 models, the CMIP6 MMM exhibits the characteristic features associated with the spatial distribution of stratospheric water vapour (Figure 12). $H_2O$ mixing ratios in the tropical lower stratosphere are low and increase with increasing altitude due to $H_2O$ production from $CH_4$ oxidation, while a distinct region of low $H_2O$ mixing ratios are modelled in the high latitude lower stratosphere due to the removal of $H_2O$ through PSC sedimentation. However, $H_2O$ mixing ratios in the CMIP6 MMM are smaller at all points in the stratosphere than in the SWOOSH dataset, and this dry bias becomes more pronounced with increasing altitude. This is partly due to the inclusion of some models (BCC-CSM2-MR, BCC-ESM1, E3SM-1-1, GFDL-CM4 and IPSL-CM6A-LR) in the CMIP6 MMM which do not include $H_2O$ production from $CH_4$ oxidation. However, even when these models are explicitly excluded from the multi-model mean (lower panel, Figure 12), while there is better agreement with the SWOOSH dataset, modelled $H_2O$ mixing ratios are still lower than those in the SWOOSH dataset and the formation of stratospheric water vapour from $CH_4$ oxidation is underrepresented.

As with ozone mixing ratios, models have typically performed poorly in simulating water vapor mixing ratios in the tropical tropopause region. Gettelman et al. (2010) show the seasonal cycle of water vapor at 80 hPa from 16 models involved in the CCMVal-2 inter-comparison project, and while there is good agreement between the CMIP6 MMM evaluated here and the SWOOSH dataset, there is a large spread in model mixing ratios, and many models do not accurately capture the seasonal cycle. Climatological (2000f-2014) tropical stratospheric water vapor mixing ratios (average over 15°S to 15°N) at 70hPa, which lies just above the cold point entry into the stratosphere, are shown for the CMIP6 models in Figure 13. There is reasonable agreement between the seasonality of the CMIP6 MMM and that calculated for the SWOOSH combined dataset, although the CMIP6 MMM is between 0.5-1.0 ppmv lower than the observations throughout the annual cycle and the minima and maxima in the seasonal cycle both occur a few months earlier in the MMM than in the observations. However, individual models display a wide range of water vapor concentrations (between 1.5-6 ppmv). As seen in Figure 11, the UKESM1-0-LL model has high stratospheric water vapor mixing ratios compared to the SWOOSH dataset but captures the seasonal cycle well, while the IPSL-CM6A-LR, CNRM-CM6-1 and CNRM-ESM2-1 models all have much lower stratospheric water vapor mixing ratios and muted seasonal cycles.

The correlation between $H_2O$ and $CH_4$ mixing ratios can be used to infer the stratospheric water vapor source from $CH_4$ oxidation in each model. Based on observations and chemical understanding, 2 molecules of stratospheric water vapor will be produced for every molecule of $CH_4$ oxidised (LeTexier et al., 1988). Given this oxidation, typical water vapor mixing ratios of ~3.5 ppmv at the tropical tropopause and mean tropospheric mixing ratios of $CH_4$ ~1.75 ppmv, it is expected that throughout the tropical stratosphere the $H_2O$ mixing ratio will equal 7.0-2.0x$CH_4$ mixing ratio (SPARC, 2010). Observations made by ACE and MIPAS satellites support this expected gradient (e.g. Archibald et al., 2019).

Of the models evaluated here, output of both water vapor and CH$_4$ mixing ratios are available from six: BCC-CSM2-MR, BCC-ESM1, CESM2-WACCM, CNRM-ESM2-1, MRI-ESM2-0 and UKESM1-0-LL. Using data from these models, scatter plots of H$_2$O vs CH$_4$ have been plotted (Figure 14), which give an indication of the water vapour production from oxidation of CH$_4$ in these models. Even from this small sample, it is clear that there is a wide range in the complexity and accuracy of modelling H$_2$O formed from the oxidation of CH$_4$. Neither BCC-CSM2-MR nor BCC-ESM1 include stratospheric water vapor production from CH$_4$ oxidation, and so H$_2$O does not increase as CH$_4$ decreases (consistent with Figure 11). Other models capture the relationship, H$_2$O = 7.0-2.0xCH$_4$, to greater or lesser extents. UKESM1-0-LL and MRI-ESM2-0 slightly under produce H$_2$O from CH$_4$, while stratospheric water vapor increases by more than two molecules for every molecule of CH$_4$ oxidised in the CNRM-ESM2-1 model. These differences in the treatment of CH$_4$ oxidation have important consequences for estimates of methane's impact on the climate system and for future radiative forcing calculations, particularly under high CH$_4$ emissions scenarios (e.g. SSP3-7.0).

## 4.2 Long-term evolution from 1850-2014

The evolution of annual mean water vapor mixing ratios at 70 hPa, averaged from 15°S-15°N, in the CMIP6 MMM and individual CMIP6 models is shown in Figure 15. Water vapor mixing ratios in the CMIP6 MMM remain relatively constant at just below 3 ppmv from 1850 to ~1950, before slowly increasing throughout the latter half of the 20$^{th}$ century and first decades of the 21$^{st}$ century. Superimposed on these long-term trends are abrupt increases in 70 hPa water vapour mixing ratios following large magnitude volcanic eruptions, particularly the eruption of Krakatoa in 1883 and Pinatubo in 1991, which increase TTL temperatures, resulting in increased annual mean water vapor mixing ratios of up to 0.5 ppmv. There is poor agreement between the CMIP6 MMM and water vapour mixing ratios from the SWOOSH dataset, with the CMIP6 MMM 0.5-1.0 ppmv lower than observed values throughout the period of observations (consistent with Figure 13). There is also poor agreement between the long-term trend in the CMIP6 MMM and the SWOOSH dataset. While 70 hPa H$_2$O mixing ratios increase from 1984-2014 in the CMIP6 MMM, there is no clear increase over this period in the SWOOSH dataset, which is instead fairly constant and has much higher interannual variability, with the abrupt year 2000 decrease in water vapour mixing ratios a prominent feature in the SWOOSH timeseries.

There is broad disagreement between the individual CMIP6 models throughout the historical period, with simulated stratospheric water vapour mixing ratios varying between 1.5-5.5 ppmv in the pre-industrial period (Figure A7). The lower panels in Figure 15 show 70 hPa water vapour mixing ratios for the individual CMIP6 models and the SWOOSH dataset, and it is clear that while many models underestimate water vapour mixing ratios (e.g. BCC-ESM1, CNRM-CM6-1, CNRM-ESM2-1, E3SM-1-1, GFDL-CM4 and IPSL-CM6A-LR), other models better capture the observed H$_2$O mixing ratios. The UKESM1-0-LL model is alone in significantly overestimating H$_2$O mixing ratios during the period for which SWOOSH observations are available. Additionally, the individual models show very different sensitivities to volcanic eruptions, with larger increases in

water vapor mixing ratios following volcanic eruptions in the BCC-CSM2-MR, CNRM-CM6-1 and CNRM-ESM2-1, and MRI-ESM2-0 models, more muted responses in the BCC-ESM1, UKESM-0-LL, CESM2, CESM2-FV2, CESM2-WACCM, CESM2-WACCM-FV2, and NorESM2-MM models, and almost no response in the GFDL-CM4 and IPSL-CM6A-LR models.

To understand the long-term trends in stratospheric water vapor, it is instructive to analyse changes in temperature in the tropics at 100 hPa, which is close to the cold point and so controls the entry values of water vapor into the stratosphere. Long-term changes in CMIP6 MMM 100 hPa temperatures, averaged from 15°S-15°N, are shown in Figure 16. The rise in 70 hPa water vapor mixing ratios in the latter part of the 20$^{th}$ century, and following volcanic eruptions, can be attributed to the increase in temperature at the 100 hPa level. In the CMIP6 MMM, TTL temperatures have increased by ~1 K between 1850 and 2014, and can rise by 1-2 K following explosive volcanic eruptions.

Climatological annual mean, zonal mean $H_2O$ mixing ratio differences between the present day (2000-2014 averaged) and pre-industrial (1850-1864 averaged) periods are shown in Figure 17. Simulated stratospheric water vapor mixing ratios have increased between the pre-industrial and present-day periods throughout the stratosphere. In the lower stratosphere, this increase is ~0.2-0.4 ppmv, consistent with the increase in water vapour mixing ratios seen at 70 hPa in the tropics, and reflects the warming of the tropical tropopause cold point between the pre-industrial and present-day. However, the increase in stratospheric water vapor mixing ratios increases with altitude and is largest in the upper stratosphere (~0.8 ppmv), reflecting increased $CH_4$ mixing ratios and resulting increases in $H_2O$ mixing ratios formed from $CH_4$ oxidation.

## 4.3 Long-term evolution from 2015-2100

An increase in stratospheric water vapor concentrations under climate change is projected by virtually all climate models (Gettelman et al., 2010; Smalley et al., 2017; Banerjee et al., 2019). Eyring et al. (2010) calculate a mean increase of 0.5-1.0 ppmv per century in stratospheric water vapor concentrations for models involved in the CCMVal-2 inter-comparison project, although agreement between models on the absolute increase is poor. The increase is likely due to the prevailing effect of a warming troposphere over other driving factors (Dessler et al., 2013; Smalley et al., 2017), and represents a climate feedback, as the associated radiative effect of the increases are correlated with increasing surface temperatures (Banerjee et al., 2019). Here we find consistent results, with increasing stratospheric water vapor concentrations under each SSP scenario (Figure 15). The magnitude of the increases generally follows the radiative forcing across the scenarios (and thus the degree of tropospheric warming). Low forcing scenarios (SSP1-1.9 and SSP1-2.6) project increasing stratospheric water vapor until the middle of the century and then a stabilization to around 3.5 ppmv. Middle of the road scenarios (SSP2-4.5, SSP4-6.0 and SSP4-3.4) reach around 4 ppmv by 2100. High forcing scenarios (SSP3-7.0 and SSP5-8.5) show rapid increases in stratospheric water vapor throughout the century, reaching around 5 ppmv by 2100.

As with the historical changes explored in Section 4.2, projected changes in water vapor mixing ratios at 70 hPa are strongly correlated with simulated changes to 100 hPa temperatures (compare Figures 15 and 16). In general, the higher the assumed GHG emissions in the SSP scenario, the larger the projected 100 hPa temperatures by the end of the century. Under SSP1-1.9 and SSP1-2.6, 100 hPa temperatures are projected to remain relatively close to present day values but are projected to increase by ~4.5 K under the SSP5-8.5 scenario.

Climatological annual mean, zonal mean $H_2O$ mixing ratio differences between the end of the century (2086-2100 averaged) and present day (2000-2014 averaged) periods for SSP1-2.6, SSP2-4.5, SSP3-7.0 and SSP5-8.5 are shown in Figure 18. Under the SSP1-2.6 scenario, stratospheric water vapor mixing ratios are projected to remain close to present day values throughout the stratosphere. However, in all other scenarios shown in Figure 18, stratospheric water vapor is projected to increase due to the increases in projected 100 hPa temperatures and increased $CH_4$ mixing ratios (particularly under SSP3-7.0, the scenario which assumes the largest increases in $CH_4$ emissions, which shows larger stratospheric water vapor increases in the upper stratosphere due to increased water vapor production from $CH_4$ oxidation).

## 5 Discussion and conclusions

This study presents an evaluation of stratospheric ozone and water vapor changes from the pre-industrial to the end of the 21$^{st}$ century in simulations performed by CMIP6 models under a range of future SSP scenarios. In total, for the historical period 1850-2014 ozone data was available from 22 models, while water vapor data was available from 18, and a subset of these models had also performed simulations under several SSP scenarios.

For zonal mean stratospheric ozone mixing ratios there is good agreement between the CMIP6 MMM and observations from the SWOOSH combined dataset, with biases within ±10%, while for TCO there is good agreement between the CMIP6 MMM and the NIWA-BS dataset from 40°S-90°N, with biases within ±20 DU (<±10%). Largest percentage zonal mean ozone mixing ratios biases occur in the tropical upper stratosphere, while for TCO the largest biases occur between 90°S-40°S. However, despite the agreement between the CMIP6 MMM and the observations, there are significant differences between the individual CMIP6 models.

From 1850 to 1960, global TCO in the CMIP6 MMM increased from ~298 DU to ~304 DU, before rapidly declining during the 1970s, 1980s and 1990s with the onset of halogenated ODS emissions. TCO increases in the early part of the historical period were driven by increases in tropospheric ozone, particularly in the NH. Superimposed on the long-term trend is the 11-year solar cycle, which causes TCO averaged from 60°S-60°N to vary by around ±1 DU, while the 1883 eruption of Krakatoa caused TCO values to increase by around 3-5 DU and resulted in the highest TCO values modelled between 1850 and 1950.

However, there is poor agreement between the individual CMIP6 models for the absolute magnitude of TCO in the pre-industrial and throughout the historical period, with model TCO values spread across a range of ~60 DU.

Models which prescribe stratospheric ozone from the CMIP6 ozone dataset show surprisingly large variation in TCO, particularly in the pre-industrial period, at which time there is a ~20 DU range in pre-industrial TCO values between those models prescribing the CMIP6 ozone dataset. There are also large percentage differences between zonal mean ozone fields output by the individual models and the CMIP6 ozone dataset, likely connected to the interpolation some models employ to provide data on the pressure levels of the CMIP6 data request. Together, this evidence suggests that TCO is not conserved after model implementation of the CMIP6 ozone dataset, and instead small differences are introduced between the models. A future challenge for modelling centres is to prescribe ozone concentrations in such a way as to preserve local mixing ratios and the total column abundance.

No consistent difference is identified between models which prescribe stratospheric ozone using the CMIP6 ozone dataset and those which use interactive chemistry schemes. There is good agreement between the CMIP6 MMM and the mean of models using interactive chemistry schemes throughout the historical period at all latitude ranges. However, there are large differences in modelled TCO values between models with interactive chemistry, and the close agreement between the CMIP6 MMM and the mean of models with interactive chemistry is likely a result of compensating biases from each individual model cancelling out. Certainly, some models with interactive chemistry are clear outliers from the CMIP6 MMM, and understanding the reasons for these differences is a challenge for the various modelling centres.

For the future period, from 2015-2100, the higher the forcing pathway assumed by the various SSPs evaluated here, the higher the TCO at the end of the century. Annual mean TCO is projected to return to the 1960s values at most latitudes by the middle of the 21$^{st}$ century under the SSP2-4.5, SSP4-3.4 and SSP4-6.0 scenarios, and under the SSP3-7.0 and SSP5-8.5 scenarios significant increases above the 1960s value are simulated, driven in part by the decline in ODS concentrations, large increases in ozone mixing ratios in the upper stratosphere associated with $CO_2$ cooling and increases in tropospheric ozone mixing ratios. In contrast, TCO values are not projected to return to the 1960's values at most latitude ranges in the SSP1-1.9 and SSP1-2.6 scenarios, due, in part, to smaller ozone mixing ratio increases in the stratosphere, consistent with reduced $CO_2$ induced cooling, and strong decreases in tropospheric ozone mixing ratios throughout the troposphere, driven by reductions in the emission of ozone precursors. While decreases in tropospheric ozone prevent TCO from returning to 1960's values, the decrease is undoubtedly a positive result for air quality, and perhaps calls into question whether TCO values are an accurate measure of stratospheric ozone recovery, or if other metrics can more accurately reflect the profile changes expected for stratospheric ozone recovery without being influenced by tropospheric changes.

Stratospheric water vapor mixing ratios are poorly represented in many of the CMIP6 models investigated in this study. For the climatological 2000-2014 period, the models simulate lower water vapour mixing ratios than those seen in the SWOOSH

dataset, particularly in the upper stratosphere. This results from several of the models studied here not including any representation of water vapor formed from the oxidation of $CH_4$ in the stratosphere. However, even when only models including water vapour production from $CH_4$ oxidation are evaluated, CMIP6 models still underestimate the increase in water vapour mixing ratios observed with increasing altitude. The seasonal cycle and water vapor mixing ratios for individual CMIP6 models at 70 hPa in the tropics shows poor agreement with the SWOOSH dataset, and further highlights the difficulties climate models have had over several generations of model intercomparison projects in the tropical tropopause region. When averaged together, there is reasonable agreement between the seasonality of the CMIP6 MMM and that calculated for the SWOOSH combined dataset, although the CMIP6 MMM is between 0.5-1.0 ppmv lower than the observations throughout the annual cycle and the minima and maxima in the seasonal cycle both occur a few months earlier in the MMM than in the observations.

For the CMIP6 MMM, 70 hPa water vapor mixing ratios remain relatively constant from 1850 to 1950, before slowly increasing to 2014. Throughout the historic period, the largest variations in water vapor mixing ratios occur at the time of major volcanic eruptions. From 2014, tropical water vapor mixing ratios at 70 hPa are projected to increase under all SSP scenarios, with the magnitude of the increases generally following the radiative forcing across the scenarios. Under SSP1-1.9 and SSP1-2.6 water vapor mixing ratios are projected to increase from 3.2 ppmv to 3.5 ppmv by the middle of the 21$^{st}$ century before stabilising, while under SSP3-7.0 and SSP5-8.5 water vapor mixing ratios show rapid increases throughout the century, reaching around 5 ppmv by the 2100.

The data available from the CMIP6 models evaluated here do not allow for thorough investigation into the drivers of the changes identified here. It is hoped that new datasets generated by models performing AerChemMIP simulations will provide greater insight into the wider chemical changes occurring throughout the atmosphere, including changes to stratospheric catalytic loss cycles and water vapor produced through $CH_4$ oxidation.

## 6 Acknowledgements

We acknowledge the World Climate Research Programme, which, through its Working Group on Coupled Modelling, coordinated and promoted CMIP6. We thank the climate modelling groups for producing and making available their model output, the Earth System Grid Federation (ESGF) for archiving the data and providing access, and the multiple funding agencies who support CMIP6 and ESGF. JK and PTG thank NERC for financial support through NCAS (Funder reference: R8/H12/83/003). BH was supported by the European Union's Horizon 2020 Framework Programme for Research and Innovation "Coordinated Research in Earth Systems and Climate: Experiments, kNowledge, Dissemination and Outreach (CRESCENDO)" project under Grant Agreement No. 641816. GZ and OM were supported by the NZ Government's Strategic Science Investment Fund (SSIF) through the NIWA programme CACV. MD was supported by the Japan Society for the Promotion of Science (grant numbers: JP20K04070). G.C. was supported by the Swiss National Science Foundation within

the Ambizione program (grant reference PZ00P2_180043). The work of SB and PC was supported by the Energy Exascale Earth System Model (E3SM) project, funded by the U.S. Department of Energy, Office of Science, Office of Biological and Environmental Research. The work of PC was performed at LLNL under contract DE-AC52-07NA27344. We thank the GFDL model development team, who led the development of GFDL-CM4 and GFDL-ESM4, as well as the numerous scientists and technical staff at GFDL who contributed to the development of these models and conducted the CMIP6 simulations used here. We acknowledge Gokhan Danabasoglu for input on the CESM2 and CESM2(WACCM6) model descriptions. The E3SM-1-0 and E3SM-1-1 were obtained from the Energy Exascale Earth System Model project, sponsored by the U.S. Department of Energy, Office of Science, Office of Biological and Environmental Research.  The E3SM data were produced using resources of the National Energy Research Scientific Computing Center, a DOE Office of Science User Facility supported by the Office of Science of the U.S. Department of Energy under Contract No. DE-AC02-05CH11231. The corresponding recipe that can be used to reproduce several figures of this paper will be included in the ESMValTool v2.0. The ESMValTool v2.0 is released under the Apache License, Version 2.0. The latest release of ESMValTool v2.0 is publicly available on Zenodo at https://doi.org/10.5281/zenodo.3970975 (Andela et al., 2020). The ESMValTool and its workflow manager/preprocessor (ESMValCore) are developed on the GitHub repositories available at https://github.com/ESMValGroup (last access: 1 December 2020). We would like to thank Bodeker Scientific, funded by the New Zealand Deep South National Science Challenge, for providing the combined NIWA-BS total column ozone database.

Data availability: All model datasets used in this study are available through the Earth System Grid Federation (ESGF; https://esgf-index1.ceda.ac.uk/projects/cmip6-ceda/). Specific references for each model dataset can be found in Table 1, and the doi's for each dataset are included in the reference list. Version 3.4 of the National Institute of Water and Atmospheric Research - Bodeker Scientific (NIWA-BS) combined TCO database is available from http://www.bodekerscientific.com/data/total-column-ozone, while the Stratospheric Water and OzOne Satellite Homogenized (SWOOSH) dataset is available from https://csl.noaa.gov/groups/csl8/swoosh/.

Author contribution: JK and BH coordinated the research team. JK, BH, AB, RC-G, Gc, SD, VE, PTG, OM, PN, GZ and JZ processed and analysed the CMIP6 model and observational datasets. GB and SD provided the observational datasets. SB, PC-S, DC, CD, MD, LWH, AK, LL, GL, MM, MJM, PN, DO, SP, OS, JS, K-HW and TW led efforts from their respective research centres to perform the CMIP6 model simulations and provide the data. All authors contributed to the preparation of the manuscript.

Competing interests: The authors declare that they have no conflict of interest.

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

| Model | Resolution | Stratospheric Chemistry | Ozone | Water Vapor | Datasets |
|-------|-----------|------------------------|-------|-------------|----------|
| AWI-ESM-1-1-LR | 192 x 96 longitude/latitude; 47 levels; top level 80 km | Prescribed (CMIP6 dataset) | Historical (1) | Historical (1) | Danek et al. 2020 |
| BCC-CSM2-MR | 320 x 160 longitude/latitude; 46 levels; top level 1.46 hPa | Prescribed (CMIP6 dataset) | Historical (3) SSP1-2.6 (1) SSP2-4.5 (1) SSP3-7.0 (1) SSP5-8.5 (1) | Historical (3) SSP1-2.6 (1) SSP2-4.5 (1) SSP3-7.0 (1) SSP5-8.5 (1) | Wu et al., 2019 Xin et al., 2019 |
| BCC-ESM1 | 128 x 64 longitude/latitude; 26 levels; top level 2.19 hPa | Prescribed (CMIP6 dataset) | Historical (3) | Historical (3) | Zhang et al., 2018 |
| CESM2 | 288 x 192 longitude/latitude; 32 levels; top level 2.25 hPa | Prescribed (other) | Historical (11) SSP1-2.6 (1) SSP2-4.5 (1) SSP3-7.0 (2) SSP5-8.5 (2) | Historical (11) -- SSP2-4.5 (1) -- -- | Danabasoglu, 2019a Danabasoglu, 2019b |
| CESM2-FV2 | 144 x 96 longitude/latitude; 32 levels; top level 2.25 hPa | Prescribed (other) | Historical (1) | Historical (1) | Danabasoglu, 2019c |
| CESM2-WACCM | 144 x 96 longitude/latitude; 70 levels; top level 4.5e-06 hPa | Interactive chemistry | Historical (3) SSP1-2.6 (1) SSP2-4.5 (1) SSP3-7.0 (3) SSP5-8.5 (1) | Historical (3) -- SSP2-4.5 (1) -- -- | Danabasoglu, 2019d Danabasoglu, 2019e |
| CESM2-WACCM-FV2 | 144 x 96 longitude/latitude; 70 levels; top level 4.5e-06 hPa | Interactive chemistry | Historical (1) | Historical (1) | Danabasoglu, 2019f |
| CNRM-CM6-1 | T127; Gaussian Reduced with 24572 grid points in total distributed over 128 latitude circles (with 256 grid points per latitude circle between 30degN and 30degS reducing to 20 grid points per latitude circle at 88.9degN and 88.9degS); 91 levels; top level 78.4 km | Simplified online scheme | Historical (19) SSP1-2.6 (6) SSP2-4.5 (6) SSP3-7.0 (6) SSP5-8.5 (6) | Historical (19) SSP1-2.6 (6) SSP2-4.5 (6) SSP3-7.0 (6) SSP5-8.5 (6) | Voldoire, 2018 Voldoire, 2019 |

| CNRM-ESM2-1 | T127; Gaussian Reduced with 24572 grid points in total distributed over 128 latitude circles (with 256 grid points per latitude circle between 30degN and 30degS reducing to 20 grid points per latitude circle at 88.9degN and 88.9degS); 91 levels; top level 78.4 km | Interactive chemistry | Historical (5)<br>SSP1-1.9 (5)<br>SSP1-2.6 (5)<br>SSP2-4.5 (5)<br>SSP3-7.0 (5)<br>SSP4-3.4 (5)<br>SSP4-6.0 (5)<br>SSP5-8.5 (5) | Historical (5)<br>SSP1-1.9 (5)<br>--<br>SSP2-4.5 (5)<br>SSP3-7.0 (5)<br>SSP4-3.4 (5)<br>SSP4-6.0 (5)<br>SSP5-8.5 (5) | Seferian, 2018<br>Seferian, 2019 |
|---|---|---|---|---|---|
| E3SM-1-0 | Cubed sphere spectral-element grid; 5400 elements with p=3; 1 deg average grid spacing; 90 x 90 x 6 longitude/latitude/cubeface; 72 levels; top level 0.1 hPa | Simplified online scheme | Historical (5) | -- | Bader et al., 2019a |
| E3SM-1-1 | Cubed sphere spectral-element grid; 5400 elements with p=3; 1 deg average grid spacing; 90 x 90 x 6 longitude/latitude/cubeface; 72 levels; top level 0.1 hPa | Simplified online scheme | Historical (1) | Historical (1) | Bader et al., 2019b |
| FGOALS-g3 | 180 x 80 longitude/latitude; 26 levels; top level 2.19hPa | Prescribed (CMIP6 dataset) | Historical (3)<br>SSP1-2.6 (1)<br>SSP3-7.0 (1)<br>SSP5-8.5 (1) | -- | Li, 2019 |
| GFDL-CM4 | 360 x 180 longitude/latitude; 33 levels; top level 1 hPa | Prescribed (CMIP6 dataset) | Historical (1)<br>SSP2-4.5 (1)<br>SSP5-8.5 (1) | Historical (1)<br>SSP2-4.5 (1)<br>SSP5-8.5 (1) | Guo et al., 2018a<br>Guo et al., 2018b |
| GFDL-ESM4 | 360 x 180 longitude/latitude; 49 levels; top level 1 Pa | Interactive chemistry | Historical (1)<br>SSP1-1.9 (1)<br>SSP1-2.6 (1)<br>SSP2-4.5 (1)<br>SSP3-7.0 (1)<br>SSP5-8.5 (1) | Historical (1)<br>--<br>SSP1-2.6 (1)<br>--<br>SSP3-7.0 (1)<br>-- | Krasting et al., 2018<br>John et al., 2018 |
| IPSL-CM6A-LR | 144 x 143 longitude/latitude; 79 levels; top level 80 km | Prescribed (CMIP6 dataset) | Historical (20)<br>SSP1-1.9 (1)<br>SSP1-2.6 (3)<br>SSP2-4.5 (2)<br>SSP3-7.0 (10)<br>SSP4-3.4 (1)<br>--<br>SSP5-8.5 (1) | Historical (20)<br>SSP1-1.9 (1)<br>SSP1-2.6 (3)<br>SSP2-4.5 (2)<br>SSP3-7.0 (10)<br>SSP4-3.4 (1)<br>SSP4-6.0 (1)<br>SSP5-8.5 (1) | Boucher et al., 2018<br>Boucher et al., 2019 |

| | | | | | |
|---|---|---|---|---|---|
| MPI-ESM-1-2-HAM | 192 x 96 longitude/latitude; 47 levels; top level 0.01 hPa | Prescribed (CMIP6 dataset) | Historical (2) | Historical (2) | Neubauer et al., 2019 |
| MPI-ESM1-2-HR | 384 x 192 longitude/latitude; 95 levels; top level 0.01 hPa | Prescribed (CMIP6 dataset) | Historical (10) SSP1-2.6 (2) SSP2-4.5 (2) SSP3-7.0 (10) SSP5-8.5 (2) | Historical (10) -- SSP2-4.5 (2) -- -- | Jungclaus et al., 2019 |
| MPI-ESM1-2-LR | 192 x 96 longitude/latitude; 47 levels; top level 0.01 hPa | Prescribed (CMIP6 dataset) | Historical (10) SSP1-2.6 (3) SSP2-4.5 (3) SSP3-7.0 (3) SSP5-8.5 (3) | Historical (10) -- SSP2-4.5 (3) -- -- | Wieners et al., 2019 |
| MRI-ESM2-0 | 192 x 96 longitude/latitude; 80 levels; top level 0.01 hPa | Interactive chemistry | Historical (5) SSP1-1.9 (1) SSP1-2.6 (1) SSP2-4.5 (5) SSP3-7.0 (5) SSP4-3.4 (1) SSP4-6.0 (1) SSP5-8.5 (1) | Historical (5) SSP1-1.9 (1) SSP1-2.6 (1) SSP2-4.5 (5) SSP3-7.0 (5) SSP4-3.4 (1) SSP4-6.0 (1) SSP5-8.5 (1) | Yukimoto et al., 2019a Yukimoto et al., 2019b |
| NorESM2-MM | 288 x 192; 32 levels; top level 3 hPa | Prescribed (other) | Historical (1) SSP1-2.6 (1) SSP2-4.5 (1) SSP3-7.0 (1) SSP5-8.5 (1) | Historical (1) -- SSP2-4.5 (1) -- -- | Bentsen et al., 2019 |
| SAM0-UNICON | 288 x 192 longitude/latitude; 30 levels; top level ~2 hPa | Prescribed (CMIP6 dataset) | Historical (1) | -- | Park and Shin, 2019 |
| UKESM1-0-LL | 192 x 144 longitude/latitude; 85 levels; top level 85 km | Interactive chemistry | Historical (9) SSP1-1.9 (5) SSP1-2.6 (5) SSP2-4.5 (5) SSP3-7.0 (5) SSP4-3.4 (5) SSP5-8.5 (5) | Historical (9) SSP1-1.9 (5) SSP1-2.6 (5) SSP2-4.5 (5) SSP3-7.0 (5) SSP4-3.4 (5) SSP5-8.5 (5) | Tang et al., 2019 Good et al., 2019 |

**Table 1. Overview of models and data available at the time this manuscript was prepared, providing model name, horizontal and vertical resolution, the stratospheric chemistry scheme used, the simulations each model performed and the ESGF reference for the model datasets. For the stratospheric chemistry scheme, models use either interactive chemistry (denoting fully coupled, complex chemistry schemes), simplified online schemes (denoting simple, linear schemes), or prescribe stratospheric ozone fields. Most models prescribing stratospheric ozone use the CMIP6 dataset (Checa-Garcia, 2018b), except CESM2, CESM2-FV2, and NorESM2 which prescribe ozone values from simulations performed with the CESM2-WACCM model. Numbers in parentheses in the ozone and water vapour columns give the number of ensemble members that performed each of the listed simulations.**

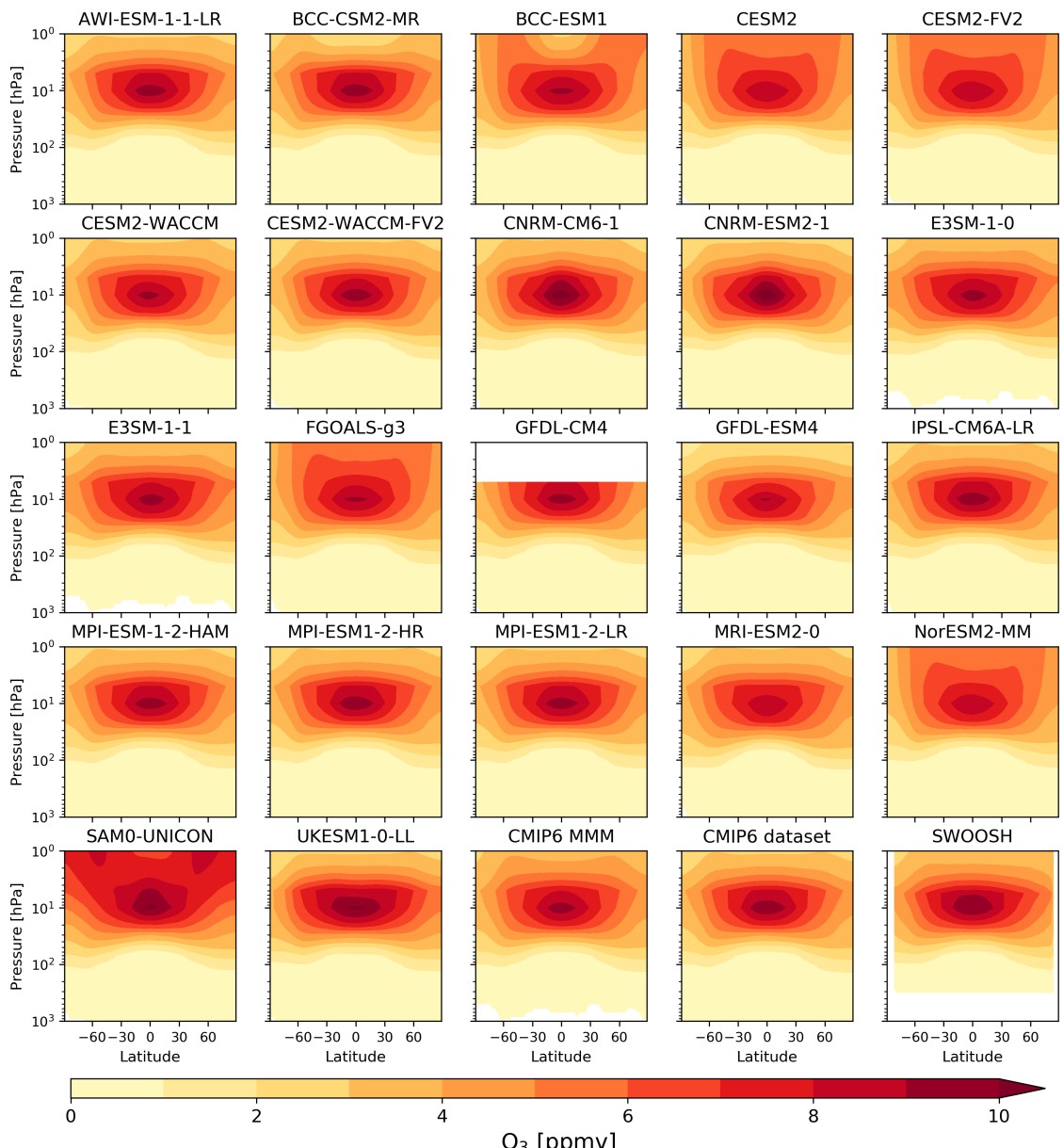

**Figure 1: Latitude vs altitude annual mean, zonal mean ozone mixing ratios (ppmv), averaged from 2000 to 2014, for each CMIP6 model, the CMIP6 multi-model mean (MMM), the CMIP6 ozone dataset used by models prescribing stratospheric ozone, and the SWOOSH combined dataset. GFDL-CM4 did not provide ozone output in the upper stratosphere, while the SWOOSH combined dataset only extends from ~300 hPa to 1 hPa.**

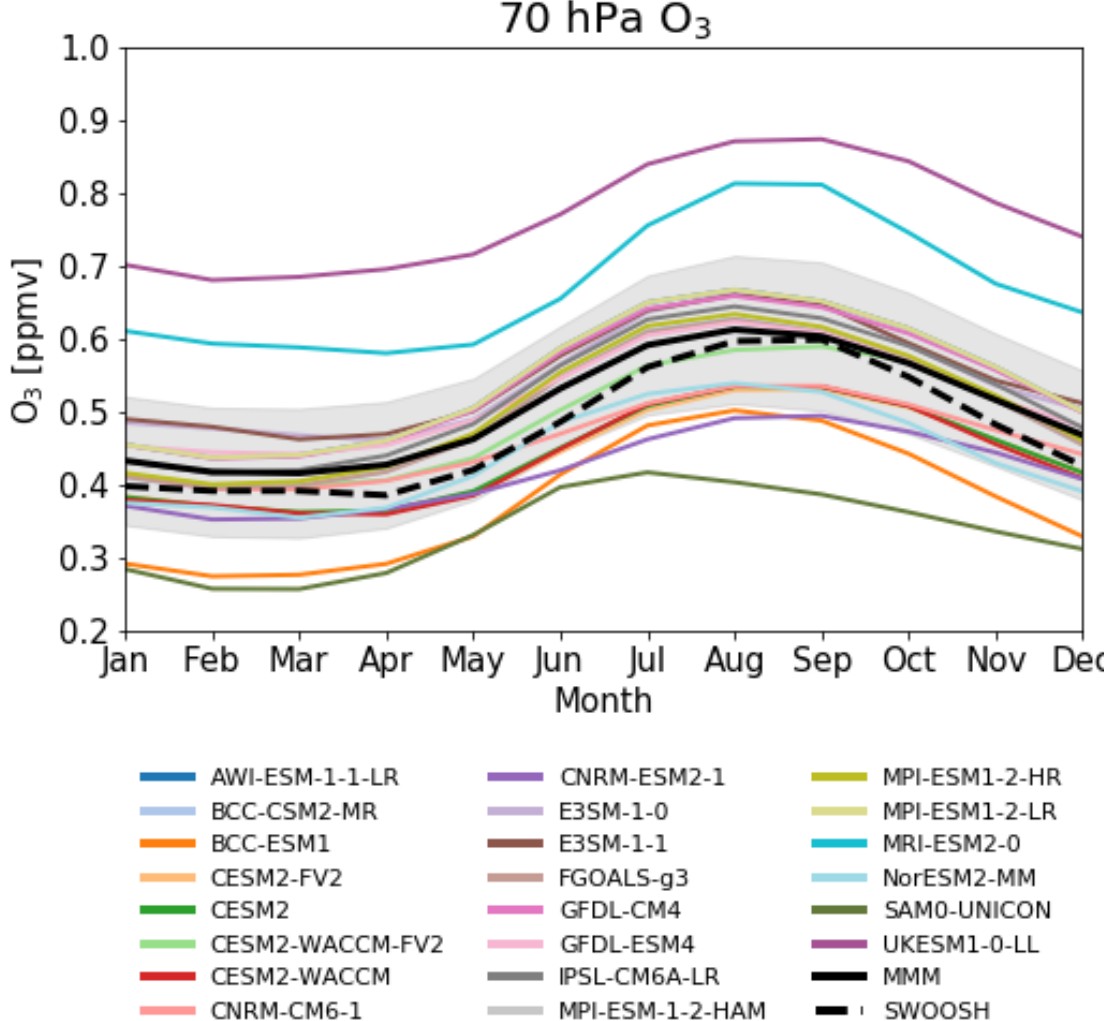

**Figure 2: Climatological (2000-2014) seasonal cycle of ozone (in ppmv) at 70 hPa (15°S-15°N average) for CMIP6 models, the CMIP6 multi-model mean (MMM; solid black line) and SWOOSH combined ozone dataset (dashed black line). The light grey envelope indicates the model spread about the MMM, calculated as the standard error of the mean.**

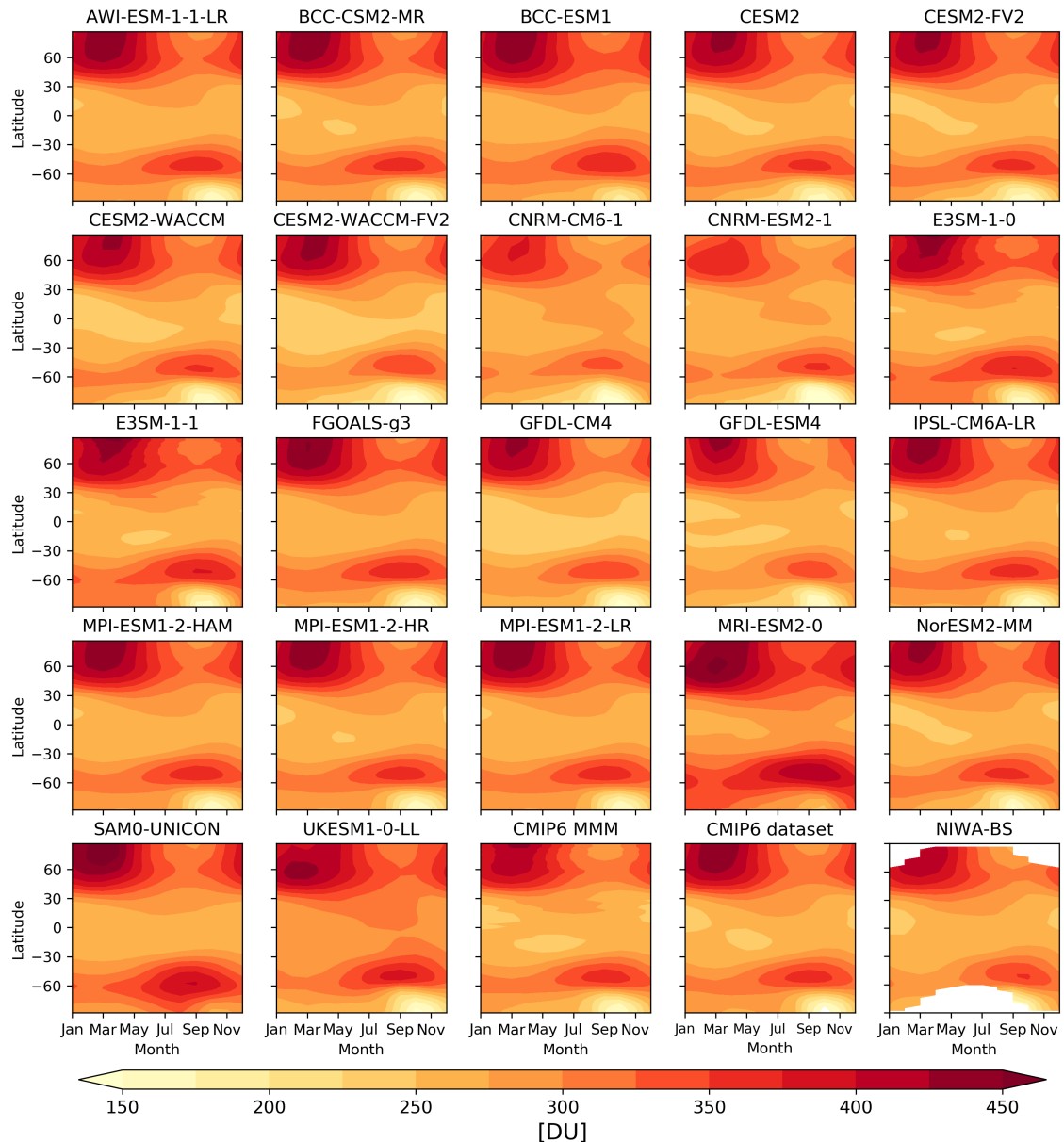

Figure 3: Month vs latitude climatological total column ozone (DU), averaged from 2000 to 2014, for each CMIP6 model, the CMIP6 multi-model mean (MMM), the CMIP6 ozone dataset used by models prescribing stratospheric ozone and the NIWA-BS dataset.

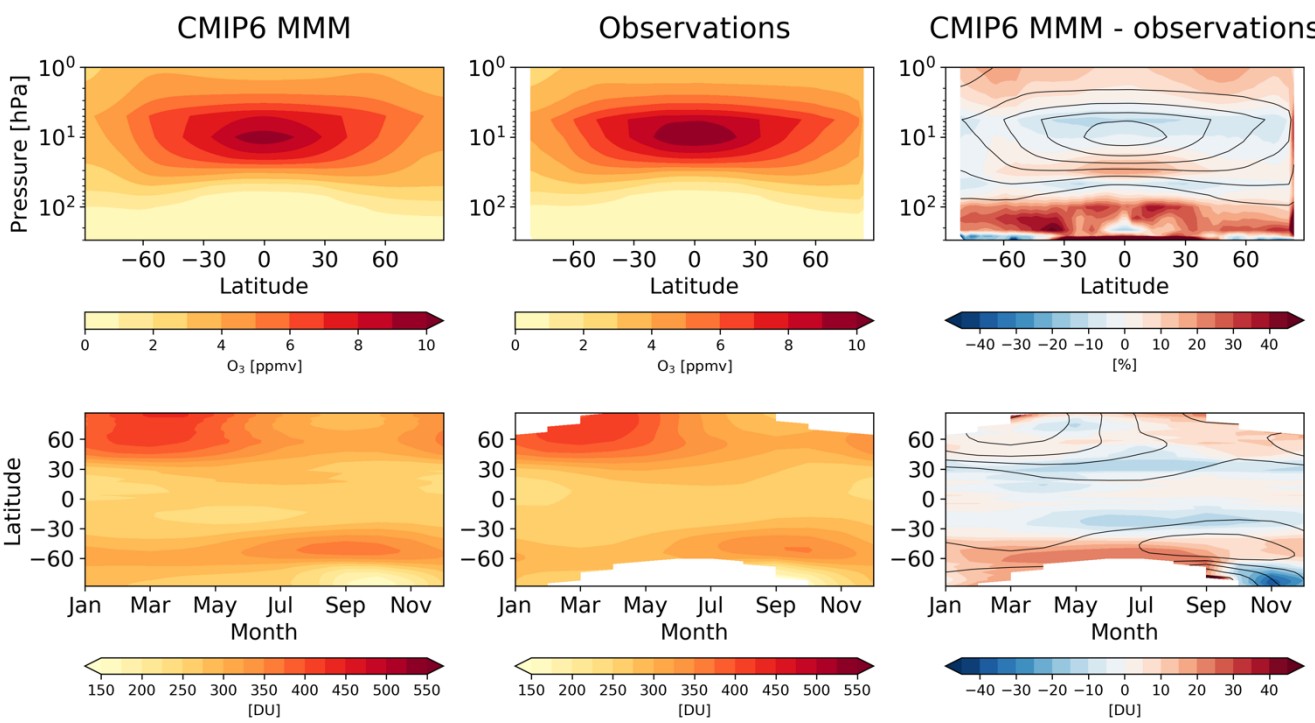

**Figure 4: Top row: 2000-2014 climatological zonal mean ozone for the CMIP6 multi-model mean (left), SWOOSH combined ozone dataset (centre) in ppmv, and corresponding differences (right) in %. Bottom row: 2000-2014 climatological total column ozone for the CMIP6 multi-model mean (left), NIWA-BS dataset (centre) and corresponding differences (right) in DU.**

1400

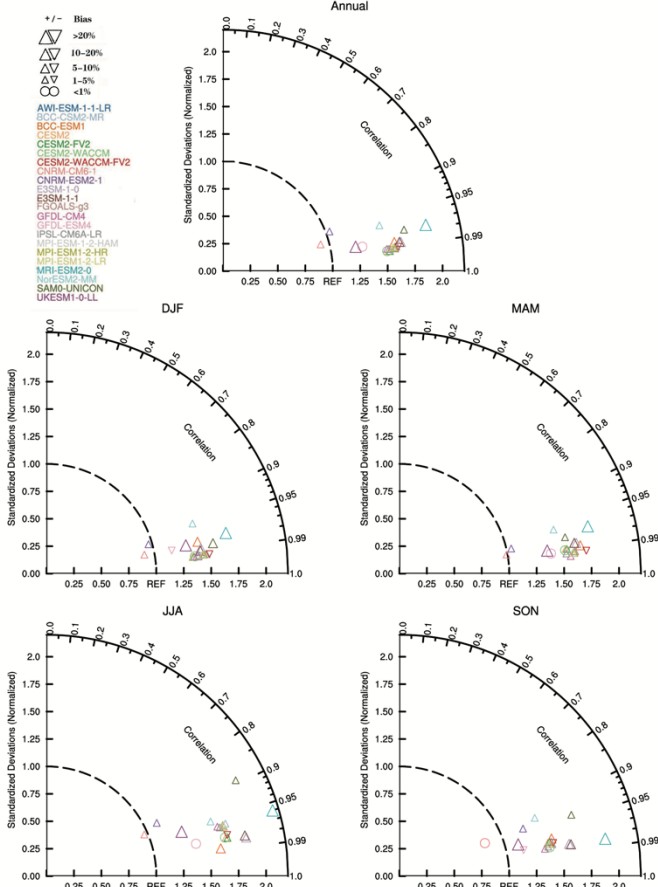

405 **Figure 5: Taylor diagrams for annual and seasonal (DJF, MAM, JJA and SON) mean total column ozone for the 22 CMIP6 models compared with the NIWA-BS dataset over 60°S–60°N for the period 2000–2014. On the Taylor diagrams, angular axes show spatial correlations between modelled and observed TCO; radial axes show spatial standard deviation (root-mean-square deviation), normalized against that of the observation; 'REF' represents the reference line; different symbols denote the percentage bias between observation and model. Each symbol represents an individual CMIP6 model.**

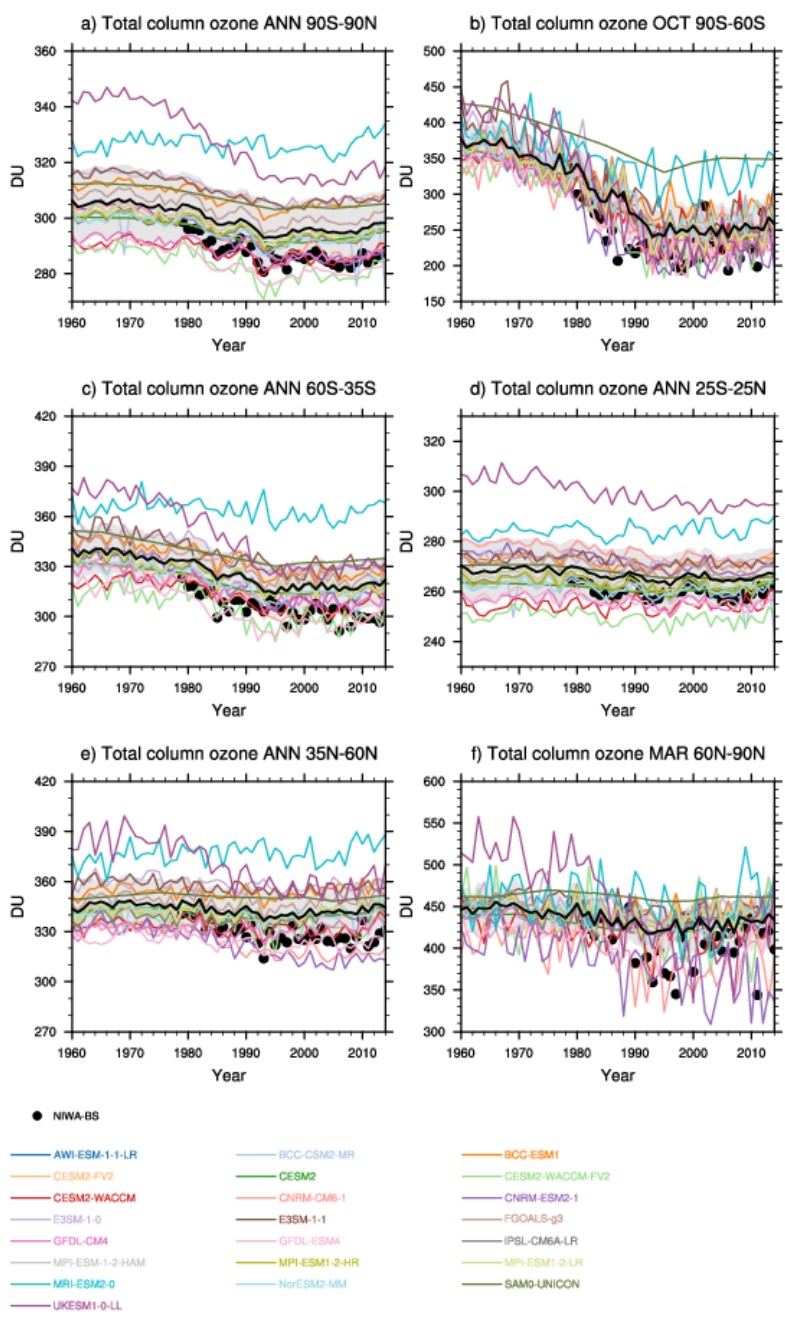

**Figure 6: Total column ozone for the individual CMIP6 models (coloured lines), the CMIP6 multi-model mean (black line) and NIWA-BS dataset (black circles), from 1960-2014, for a) annual mean values averaged from 90°S-90°N, b) October monthly mean values averaged from 90°S-60°S, c) annual mean values averaged from 60°S-35°S, d) annual mean values averaged from 25°S-25°N, e) annual mean values averaged from 35°N-60°N and f) March monthly mean values averaged from 60°N-90°N.**

| | 90S-90N<br>Annual mean | 25N-25S<br>Annual mean | 35N-60N<br>Annual mean | 35S-60S<br>Annual mean | 60N-90N<br>March mean | 60S-90S<br>Oct mean |
|---|---|---|---|---|---|---|
| Models | Modelled trends in total column ozone (TCO) between 1980 and 2000 (DU/year) | | | | | |
| **Ground-based** | **-0.56 ± 0.11** | **-0.08 ± 0.10** | **-0.84 ± 0.25** | **-0.90 ± 0.17** | **-3.18 ± 0.95** | **-5.25 ± 0.77** |
| **NOAA-SBUV** | **-0.74 ± 0.12** | **-0.16 ± 0.13** | **-1.12 ± 0.21** | **-1.21 ± 0.18** | **-3.30 ± 0.96** | **-3.99 ± 0.80** |
| **NASA TOMS-SBUV-OMI** | **-0.67 ± 0.13** | **-0.19 ± 0.12** | **-0.89 ± 0.23** | **-1.06 ± 0.17** | **-2.95 ± 0.91** | **-3.70 ± 0.82** |
| **NIWA-BS** | **-0.61 ± 0.12** | **-0.10 ± 0.11** | **-0.88 ± 0.23** | **-0.87 ± 0.16** | **-3.18 ± 0.92** | **-3.80 ± 0.78** |
| AWI-ESM-1-1-LR (1) | -0.44 ± 0.10 | -0.13 ± 0.10 | -0.28 ± 0.15 | -0.85 ± 0.18 | -0.78 ± 0.49 | -4.73 ± 0.95 |
| BCC-CSM2-MR (3) | -0.44 ± 0.10 | -0.13 ± 0.10 | -0.28 ± 0.15 | -0.87 ± 0.18 | -0.80 ± 0.51 | -4.87 ± 0.98 |
| BCC-ESM1 (3) | -0.40 ± 0.09 | -0.13 ± 0.08 | -0.21 ± 0.14 | -0.92 ± 0.18 | -0.59 ± 0.46 | -4.22 ± 0.80 |
| CESM2 (11) | -0.32 ± 0.04 | -0.08 ± 0.02 | -0.24 ± 0.04 | -0.59 ± 0.05 | -0.43 ± 0.08 | -3.98 ± 0.22 |
| CESM2-FV2 (1) | -0.32 ± 0.04 | -0.08 ± 0.02 | -0.24 ± 0.04 | -0.59 ± 0.05 | -0.41 ± 0.08 | -4.02 ± 0.23 |
| **CESM2-WACCM (3)** | -0.37 ± 0.11 | -0.04 ± 0.09 | -0.25 ± 0.11 | -0.78 ± 0.15 | -0.55 ± 0.56 | -5.36 ± 0.76 |
| **CESM2-WACCM-FV2 (1)** | -0.58 ± 0.15 | -0.17 ± 0.11 | -0.55 ± 0.32 | -1.04 ± 0.27 | -2.02 ± 1.23 | -4.51 ± 1.29 |
| CNRM-CM6-1 (19) | -0.77 ± 0.06 | -0.37 ± 0.05 | -0.78 ± 0.08 | -1.32 ± 0.11 | -1.74 ± 0.34 | -4.52 ± 0.58 |
| **CNRM-ESM2-1 (5)** | -0.77 ± 0.06 | -0.33 ± 0.05 | -0.95 ± 0.08 | -1.09 ± 0.10 | -2.81 ± 0.48 | -4.70 ± 0.50 |
| E3SM-1-0 (5) | -0.44 ± 0.03 | -0.13 ± 0.04 | -0.17 ± 0.05 | -0.90 ± 0.07 | -0.75 ± 0.23 | -5.16 ± 0.60 |
| E3SM-1-1 (1) | -0.44 ± 0.03 | -0.20 ± 0.09 | -0.19 ± 0.11 | -0.80 ± 0.15 | -0.15 ± 0.53 | -4.29 ± 1.00 |
| FGOALS-g3 (3) | -0.42 ± 0.10 | -0.18 ± 0.10 | -0.32 ± 0.15 | -0.89 ± 0.18 | -0.82 ± 0.47 | -4.66 ± 0.92 |
| GFDL-CM4 (1) | -0.41 ± 0.10 | -0.11 ± 0.09 | -0.25 ± 0.15 | -0.82 ± 0.18 | -0.73 ± 0.48 | -4.75 ± 0.95 |
| **GFDL-ESM4** (1) | -0.49 ± 0.10 | -0.17 ± 0.08 | -0.33 ± 0.13 | -0.99 ± 0.17 | -0.78 ± 0.54 | -4.87 ± 1.15 |
| IPSL-CM6A-LR (20) | -0.43 ± 0.09 | -0.14 ± 0.09 | -0.29 ± 0.15 | -0.81 ± 0.17 | -0.78 ± 0.47 | -4.73 ± 0.94 |
| MPI-ESM-1-2-HAM (2) | -0.44 ± 0.10 | -0.13 ± 0.10 | -0.28 ± 0.15 | -0.85 ± 0.18 | -0.78 ± 0.49 | -4.73 ± 0.95 |
| MPI-ESM1-2-HR (10) | -0.43 ± 0.10 | -0.13 ± 0.09 | -0.26 ± 0.15 | -0.83 ± 0.18 | -0.76 ± 0.49 | -4.75 ± 0.95 |
| MPI-ESM1-2-LR (10) | -0.44 ± 0.10 | -0.13 ± 0.10 | -0.28 ± 0.15 | -0.85 ± 0.18 | -0.78 ± 0.49 | -4.73 ± 0.95 |
| **MRI-ESM2-0** (5) | -0.19 ± 0.08 | 0.01 ± 0.10 | -0.16 ± 0.15 | -0.47 ± 0.12 | -1.10 ± 0.58 | -2.88 ± 0.35 |
| NorESM2-MM (1) | -0.42 ± 0.11 | -0.09 ± 0.09 | -0.28 ± 0.12 | -0.84 ± 0.16 | -0.52 ± 0.52 | -5.20 ± 0.86 |
| SAM0-UNICON (1) | -0.29 ± 0.03 | -0.19 ± 0.01 | -0.06 ± 0.01 | -0.49 ± 0.04 | -0.64 ± 0.04 | -2.56 ± 0.25 |
| **UKESM1-0-LL** (9) | -1.04 ± 0.07 | -0.33 ± 0.08 | -1.20 ± 0.10 | -1.66 ± 0.11 | -4.74 ± 0.46 | -6.96 ± 0.50 |
| **Mean (interactive)** | **-0.57 ± 0.09** | **-0.17 ± 0.09** | **-0.57 ± 0.15** | **-1.01 ± 0.15** | **-2.00 ± 0.64** | **-4.88 ± 0.76** |
| **Mean (all models)** | **-0.47 ± 0.08** | **-0.15 ± 0.08** | **-0.36 ± 0.12** | **-0.88 ± 0.14** | **-1.07 ± 0.46** | **-4.60 ± 0.76** |

| Models | Modelled trends in total column ozone (TCO) between 2000 and 2014 (DU/year) | | | | | |
|---|---|---|---|---|---|---|
| AWI-ESM-1-1-LR (1) | 0.11 ± 0.10 | 0.02 ± 0.15 | 0.39 ± 0.23 | -0.06 ± 0.26 | 0.01 ± 0.82 | 0.43 ± 1.12 |
| BCC-CSM2-MR (3) | 0.12 ± 0.10 | 0.02 ± 0.15 | 0.40 ± 0.23 | -0.04 ± 0.26 | -0.03 ± 0.86 | 0.45 ± 1.15 |
| BCC-ESM1 (3) | 0.19 ± 0.09 | 0.13 ± 0.10 | 0.45 ± 0.19 | 0.08 ± 0.25 | -0.28 ±0.64 | 0.52 ± 0.89 |
| CESM2 (11) | 0.34 ± 0.03 | 0.18 ± 0.02 | 0.41 ± 0.05 | 0.53± 0.06 | 1.01 ± 0.04 | 0.92 ± 0.06 |
| CESM2-FV2 (1) | 0.34 ± 0.03 | 0.18 ± 0.02 | 0.41 ± 0.05 | 0.52 ± 0.06 | 1.00 ± 0.04 | 0.92 ± 0.07 |
| **CESM2-WACCM** (3) | 0.31 ± 0.10 | 0.17 ± 0.14 | 0.37 ± 0.19 | 0.34 ± 0.19 | 1.54 ± 0.75 | 1.47 ± 0.83 |
| **CESM-WACCM-FV2 (1)** | 0.15 ± 0.18 | 0.08 ± 0.19 | 0.32 ± 0.40 | -0.01 ± 0.39 | -0.08 ± 2.49 | 0.10 ± 2.26 |
| CNRM-CM6-1 (19) | -0.13 ± 0.13 | -0.14 ± 0.09 | -0.04 ± 0.14 | -0.13 ± 0.23 | -0.04 ± 0.51 | -0.24 ± 1.34 |
| **CNRM-ESM2-1** (5) | 0.15 ± 0.10 | -0.05 ± 0.11 | 0.28 ± 0.16 | 0.27 ± 0.14 | 2.18 ± 1.39 | 1.13 ± 0.74 |
| E3SM-1-0 (5) | 0.16 ± 0.02 | 0.01 ± 0.03 | 0.29 ± 0.07 | 0.35 ± 0.06 | 0.34 ± 0.27 | 0.04 ± 0.87 |
| E3SM-1-1 (1) | 0.14 ± 0.08 | 0.02 ± 0.10 | 0.19 ± 0.19 | 0.18 ± 0.27 | 1.46 ± 0.77 | 0.87 ±1.85 |
| FGOALS-g3 (3) | 0.12 ± 0.10 | 0.04 ± 0.15 | 0.40 ± 0.23 | -0.02 ± 0.27 | 0.03 ± 0.79 | 0.42 ± 1.10 |
| GFDL-CM4 (1) | 0.10 ± 0.10 | 0.00 ± 0.14 | 0.38 ± 0.23 | -0.07 ± 0.26 | 0.00 ± 0.81 | 0.45 ± 1.11 |
| **GFDL-ESM4** (1) | 0.10 ± 0.09 | 0.13 ± 0.13 | -0.02 ± 0.24 | 0.19 ± 0.20 | -0.19 ± 0.86 | 1.07 ± 1.65 |
| IPSL-CM6A-LR (20) | 0.10 ± 0.10 | 0.01 ± 0.14 | 0.36 ± 0.22 | -0.05 ± 0.25 | 0.03 ± 0.78 | 0.46 ± 1.12 |
| MPI-ESM1-2-HAM (2) | 0.11 ± 0.10 | 0.02 ± 0.15 | 0.39 ± 0.23 | -0.06 ± 0.26 | 0.00 ± 0.82 | 0.43 ± 1.12 |
| MPI-ESM1-2-HR (10) | 0.11 ± 0.10 | 0.02 ± 0.15 | 0.38 ± 0.23 | -0.06 ± 0.26 | -0.01 ± 0.8 | 0.44 ± 1.12 |
| MPI-ESM1-2-LR (10) | 0.11 ± 0.10 | 0.02 ± 0.15 | 0.39 ± 0.23 | -0.06 ± 0.26 | 0.00 ± 0.82 | 0.43 ± 1.12 |
| **MRI-ESM2-0** (5) | 0.45 ± 0.13 | 0.10 ± 0.14 | 0.84 ± 0.32 | 0.60 ± 0.19 | 2.06 ± 1.36 | 2.35 ± 0.56 |
| NorESM2-MM (1) | 0.34 ± 0.10 | 0.19 ± 0.14 | 0.40 ± 0.19 | 0.37 ± 0.18 | 1.56 ± 0.64 | 1.50 ± 1.00 |
| SAM0-UNICON (1) | 0.08 ± 0.02 | 0.01 ± 0.02 | 0.12 ± 0.06 | 0.18 ± 0.01 | 0.24 ± 0.07 | 0.24 ± 0.11 |
| **UKESM1-0-LL** (9) | 0.26 ± 0.06 | 0.13 ± 0.04 | 0.34 ± 0.11 | 0.34 ± 0.09 | 1.49 ± 0.81 | 1.13 ± 0.32 |
| **Mean (interactive)** | **0.24 ± 0.11** | **0.09 ± 0.13** | **0.35 ± 0.24** | **0.29 ± 0.20** | **1.17 ± 1.28** | **1.21 ± 1.06** |
| **Mean (all models)** | **0.17 ± 0.09** | **0.06 ± 0.11** | **0.34 ± 0.19** | **0.15 ± 0.20** | **0.56 ± 0.78** | **0.71 ± 0.98** |

**Table 2. Linear trends and errors in area-weighted total column ozone (TCO) (DU/year) over the periods of 1980-2000 and 2000-2014. Observed trends over 1980-2000 are taken from those in Table 2 in Eyring et al. (2013). Models highlighted in bold have interactive stratospheric ozone chemistry and their means are shown in bold. Numbers in parentheses next to the models are the number of ensembles used for that model. The value following the ± symbol gives the statistical uncertainty of the trends at the 68% (1 sigma) confidence level.**

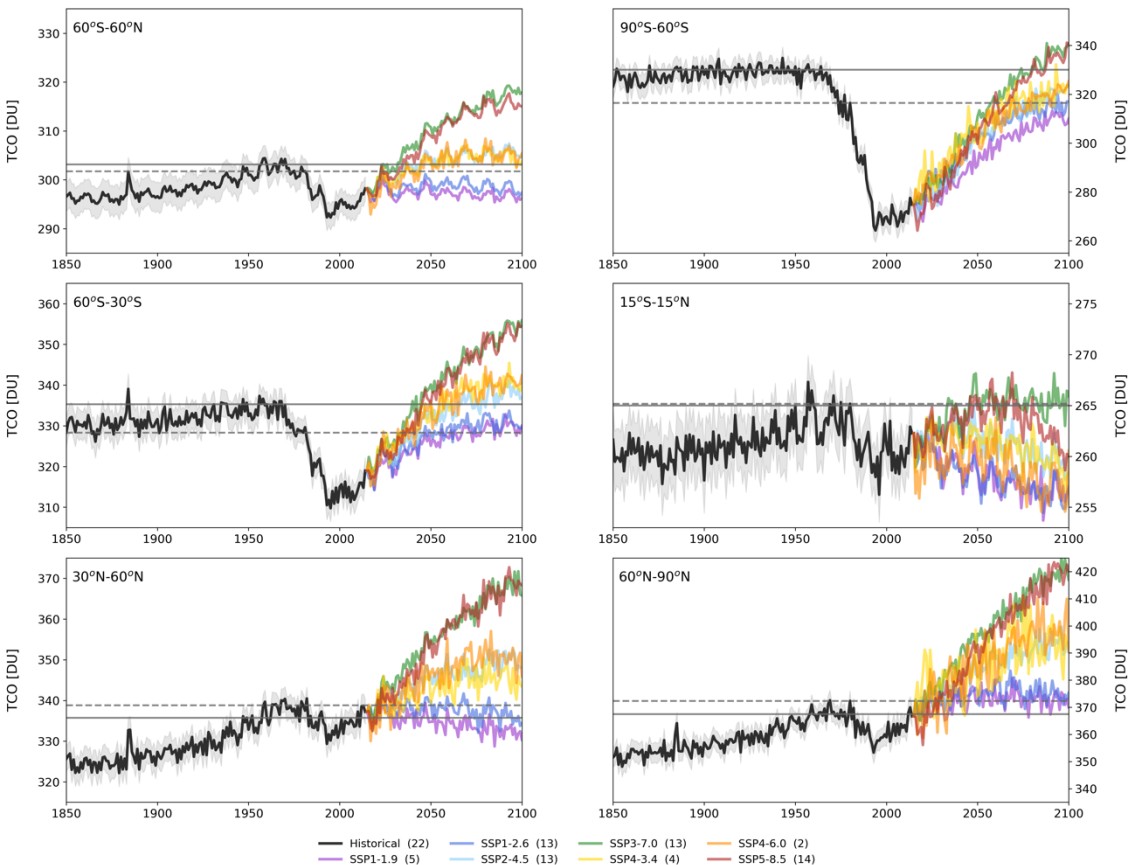

**Figure 7: Regional average, annual mean CMIP6 multi-model mean total column ozone for the historical simulation (black line), and SSP scenarios (coloured lines). The number of models performing each simulation is provide in parentheses in the legend. The light grey envelope indicates the model spread for the historical simulations (calculated as the standard error of the mean). Total column ozone values for the 1960 annual mean and 1980 annual mean are given by the solid and dashed horizontal grey lines respectively.**

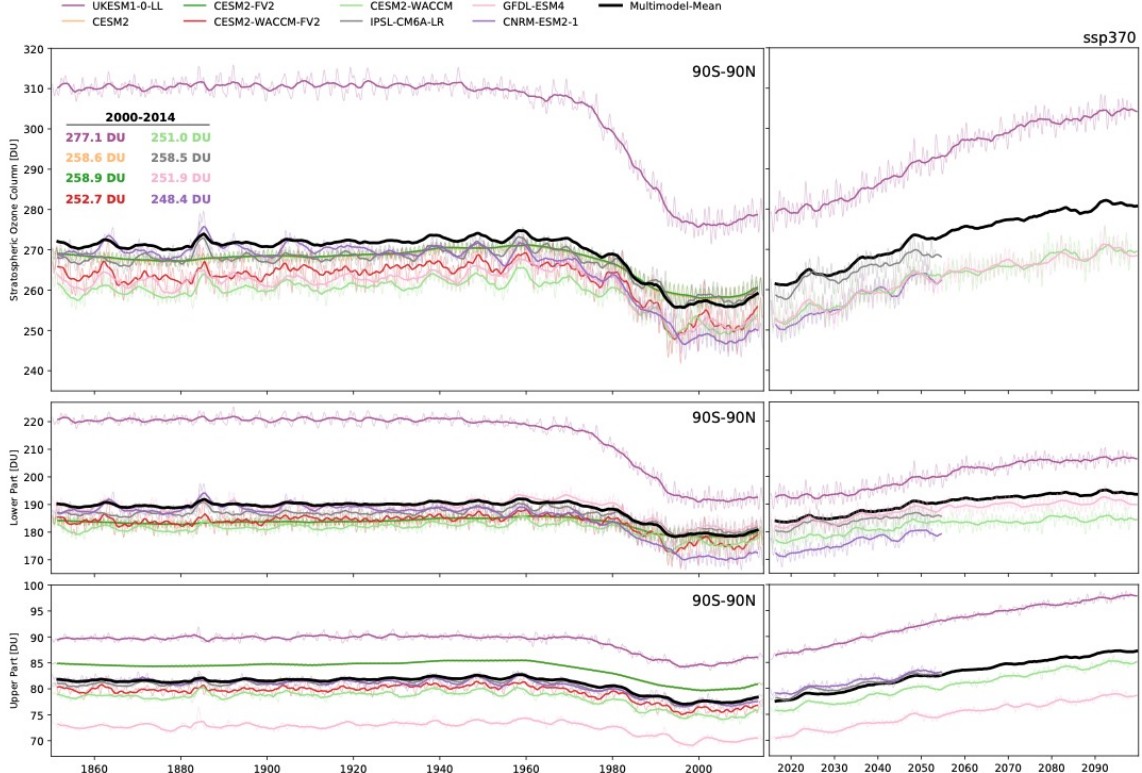

**Figure 8: Partial ozone columns, averaged from 90°S-90°N, from 1850-2014 following the historical simulation, and from 2015-2100 following the SSP3-7.0 scenario for a subset of the CMIP6 models evaluated in this study. Partial columns are calculated for the full stratosphere (tropopause to 1 hPa; upper panel), lower stratosphere (tropopause to 10 hPa; middle panel) and upper stratosphere (10 hPa to 1 hPa; lower panel). Values given in the top left panel give the climatological annual mean, global mean stratospheric column ozone values for the period 2000-2014 Note that for the multi-model mean (black line), not all models which have performed the historical simulation have also performed the SSP3-7.0, and so the multi-model mean has a discontinuity between the historical and future panels.**

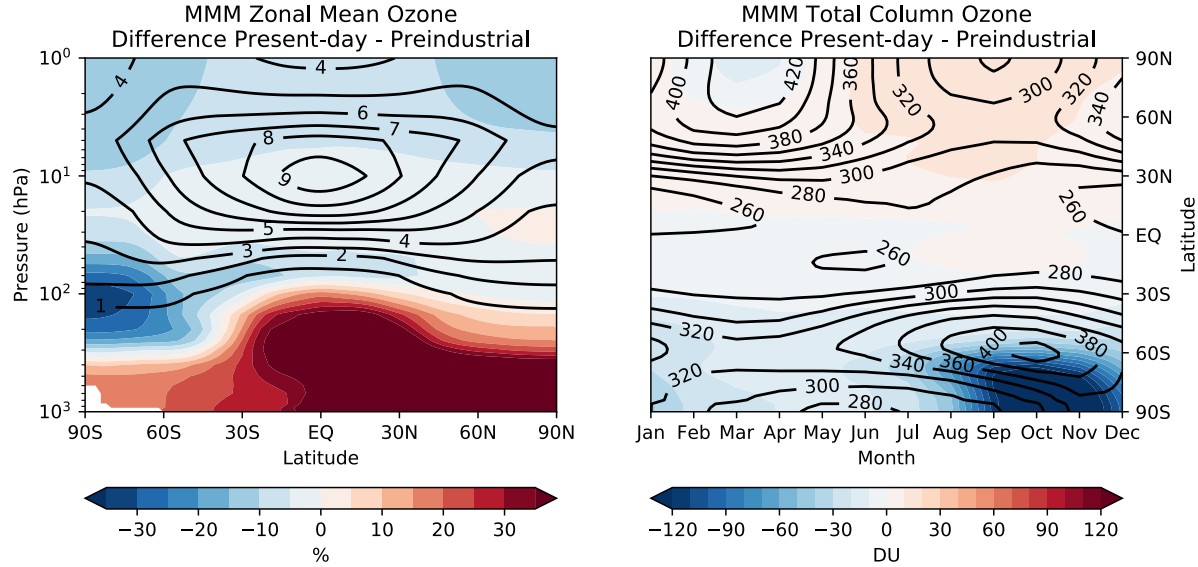

**Figure 9: CMIP6 multi-model mean (MMM) historical changes between the pre-industrial (1850-1864 averaged) and present day (2000-2014 averaged) modelled annual mean, zonal mean ozone mixing ratios in % (left), and seasonal total column ozone in DU (right), calculated for 21 of the 22 CMIP6 models evaluated in this study (GFDL-CM4 was excluded due to its low model top). The 2000-2014 averaged climatologies for both zonal mean ozone mixing ratios and total column ozone are shown in black contours.**

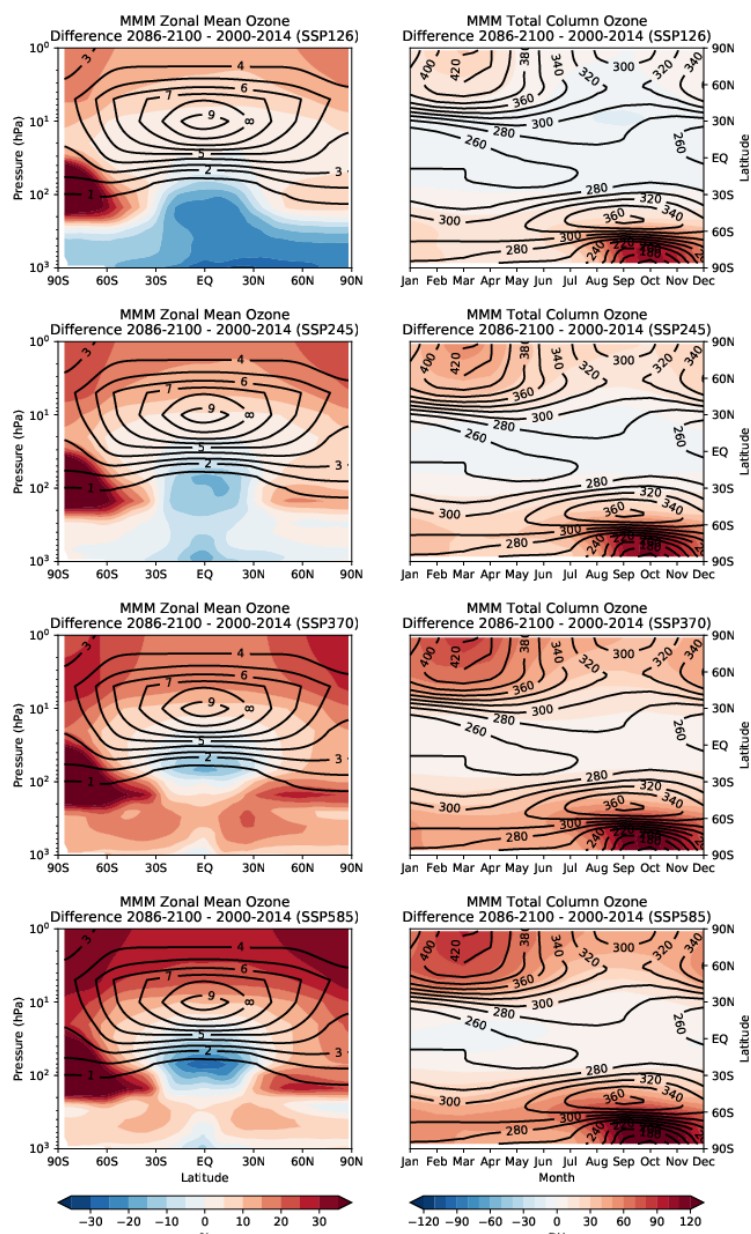

**Figure 10: Projected changes under the SSP1-2.6, SSP2-4.5, SSP3-7.0 and SSP5-8.5 scenarios between the present day (2000-2014 averaged) and end of century (2086-2100 averaged) modelled annual mean, zonal mean ozone mixing ratios in % (left), and seasonal total column ozone in DU (right), calculated using 12 of the 22 CMIP6 models evaluated in this study (BCC-CSM2-MR, CESM2, CESM2-WACCM, CNRM-CM6-1, CNRM-ESM2-1, GFDL-ESM4, IPSL-CM6A-LR, MPI-ESM1-2-HR, MPI-ESM1-2-LR, MRI-ESM2-0, NorESM2-MM, and UKESM1-0-LL). The 2000-2014 averaged climatologies for both zonal mean ozone mixing ratios and TCO are shown in black contours.**

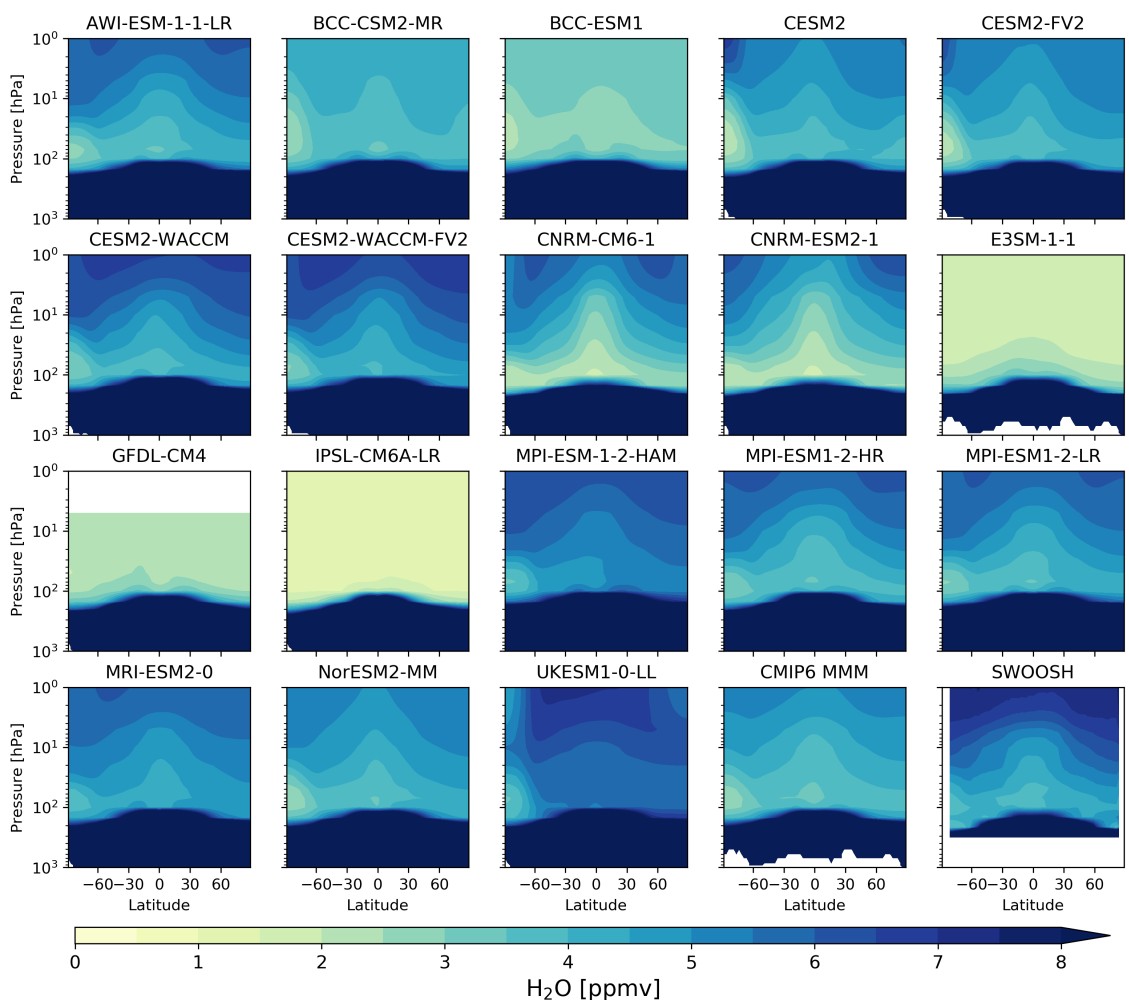

**Figure 11: Latitude vs altitude annual mean, zonal mean H$_2$O mixing ratios (ppmv), averaged from 2000 to 2014, for each CMIP6 model, the CMIP6 multi-model mean (MMM), and the SWOOSH combined dataset. GFDL-CM4 did not provide ozone output in the upper stratosphere, while the SWOOSH combined dataset only extends from ~300 hPa to 1 hPa.**

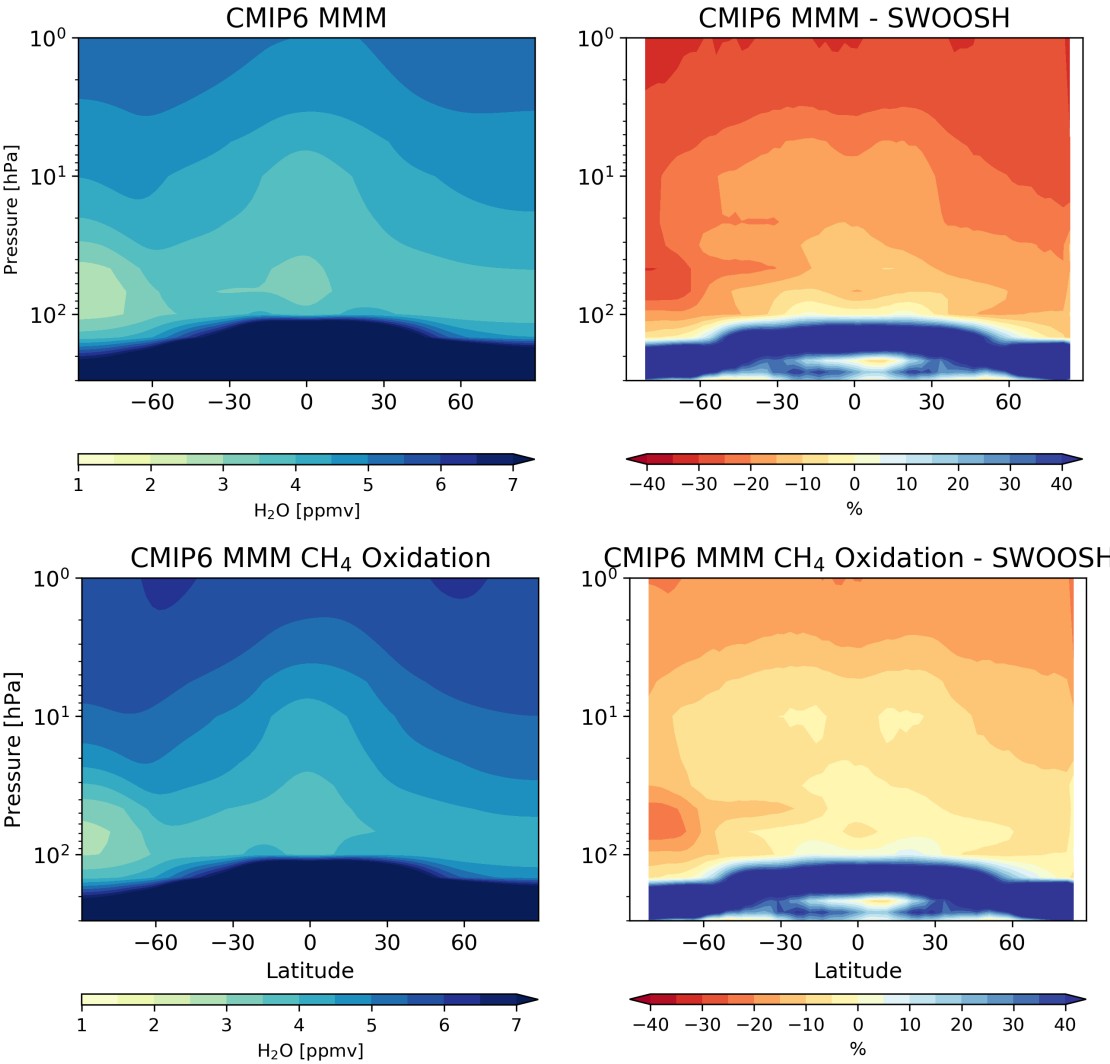

**Figure 12:** Top row: 2000-2014 climatological zonal mean $H_2O$ for the CMIP6 multi-model mean (MMM) in ppmv (left), and difference between the CMIP6 MMM and SWOOSH dataset in % (right). Bottom row: same as top row, but only including models which model $H_2O$ production from $CH_4$ oxidation. Note that for the differences, red colours indicate the model is drier (i.e. less $H_2O$ compared with the SWOOSH dataset).

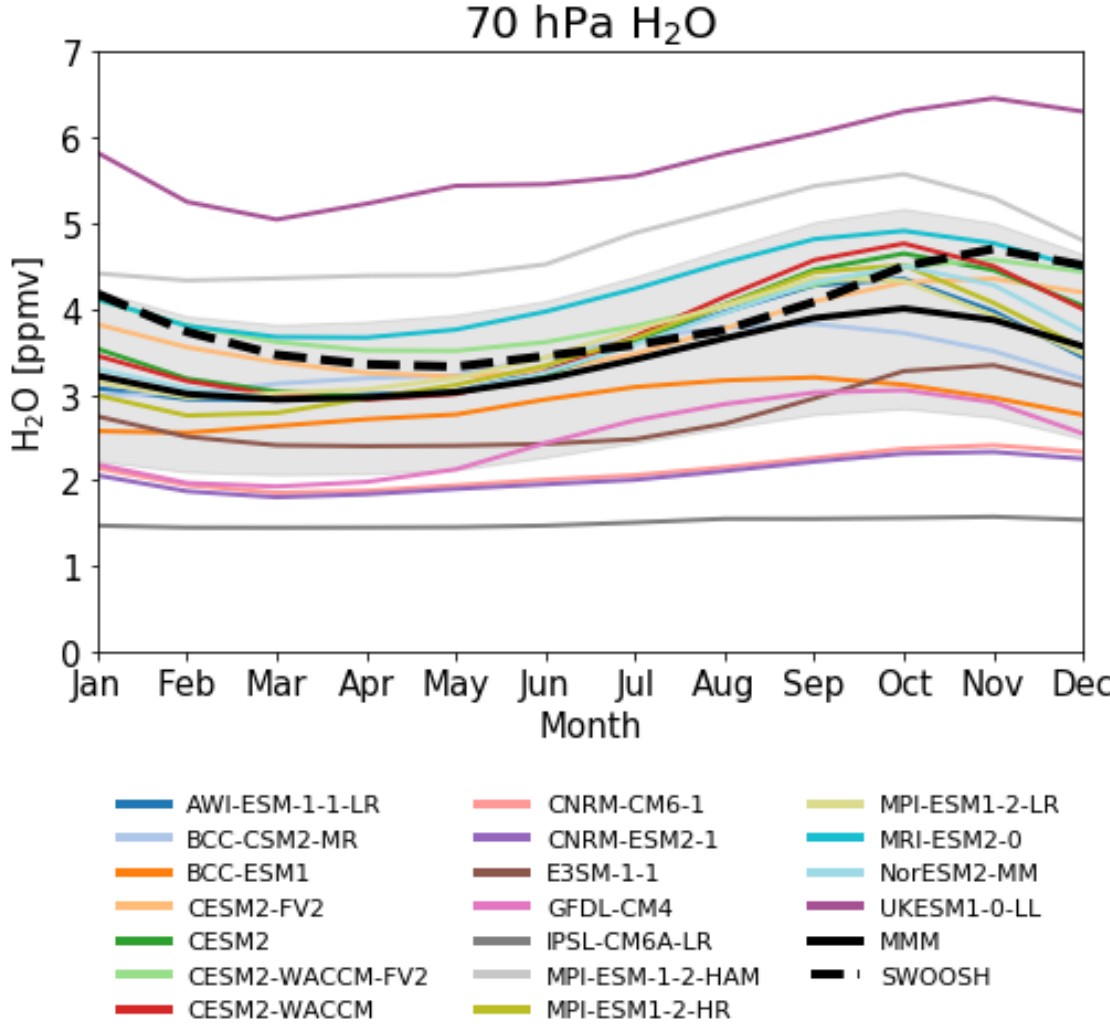

**Figure 13: Climatological (2000-2014) seasonal cycle of $H_2O$ mixing ratios (in ppmv) at 70 hPa for CMIP6 models, the CMIP6 multi-model mean (MMM; solid black line) and SWOOSH combined ozone dataset (dashed black line). The light grey envelope indicates the model spread (calculated as the standard error of the mean).**

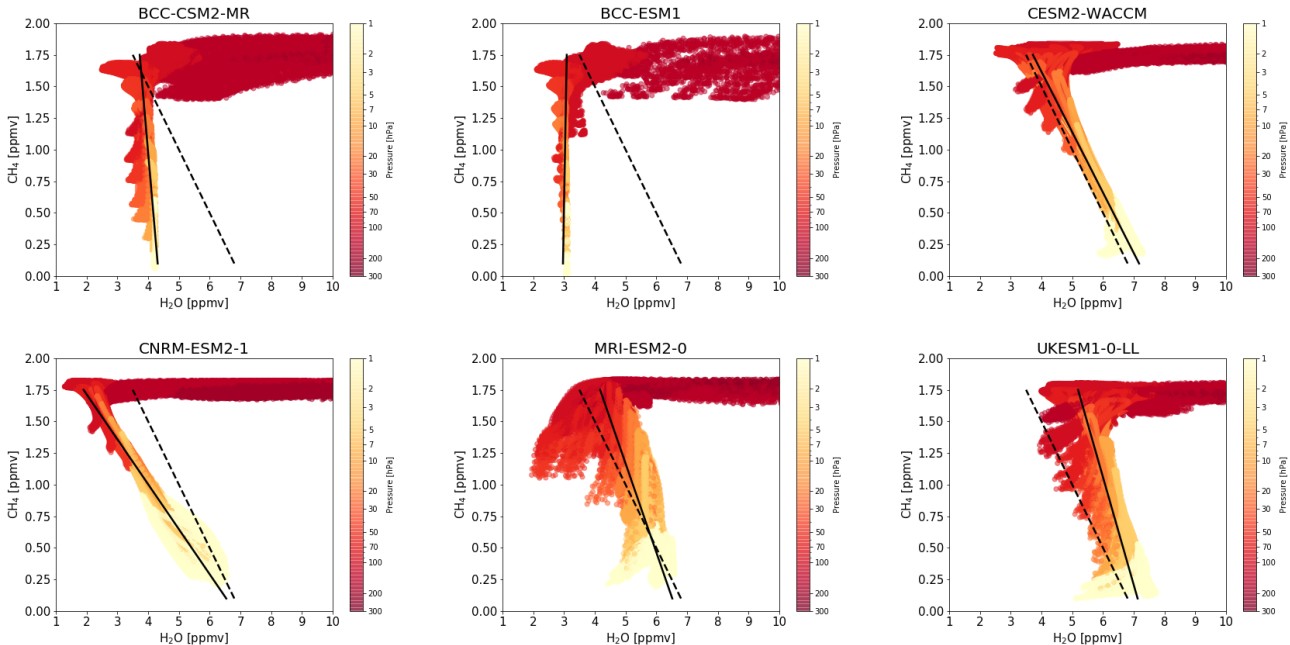

1470

**Figure 14: H₂O vs CH₄ scatter plots of the six CMIP6 models for which both H₂O and CH₄ mixing ratio are available from the historical simulation. The data shown here is monthly mean, zonal mean H₂O and CH₄ mixing ratios (in ppmv) for the years 2000-2014. The coloured shading of the points represents the altitude (in hPa). The black line gives gradient for all model points above 70 hPa, while the dashed black line gives SPARC estimate (H₂O = 7.0-2.0xCH₄)**

1475

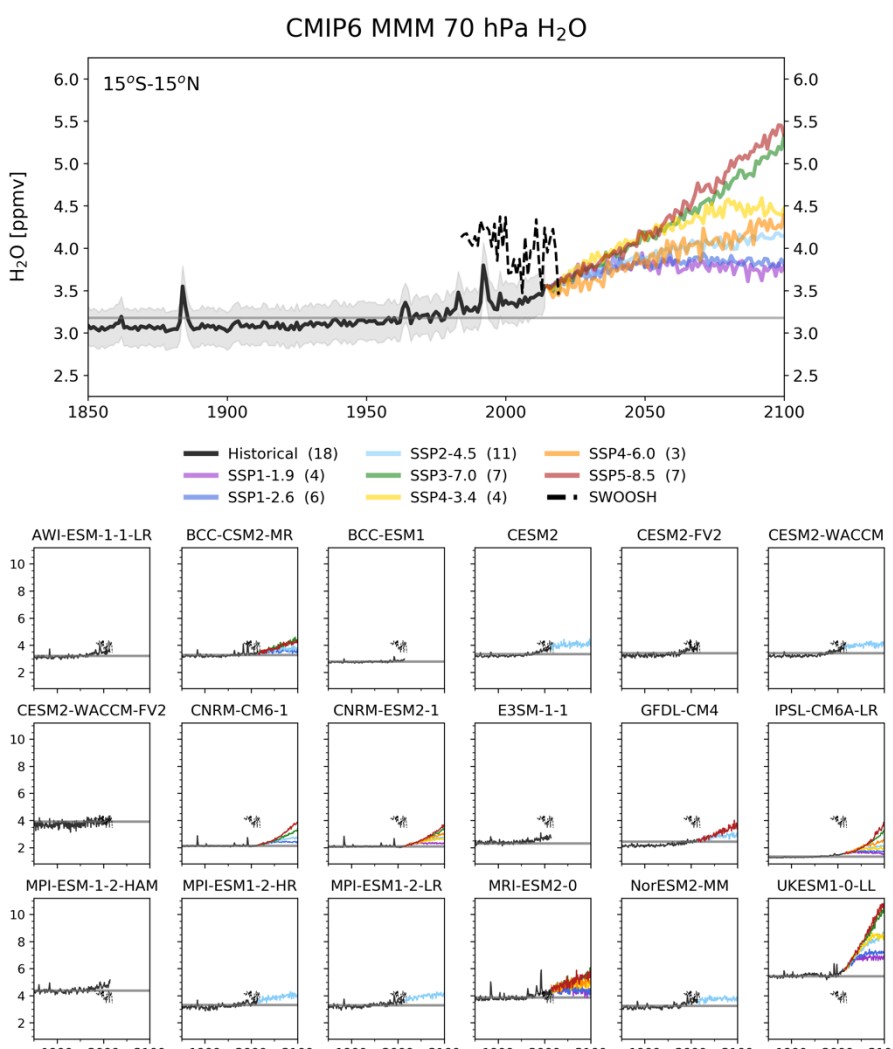

**Figure 15: Upper panel: CMIP6 multi-model mean H$_2$O mixing ratio (ppmv), averaged from 15°S-15°N at 70 hPa for the historical simulation (black line), and SSP scenarios (coloured lines). The number of models performing each simulation is provide in parentheses in the legend. The light grey envelope indicates the model spread for the historical simulations (calculated as the standard error). H$_2$O mixing ratios for the 1960 annual mean is given by the horizontal grey line. Observations from the SWOOSH combined dataset are shown in the dashed black line. Lower panels: As upper panel, but for each individual CMIP6 model.**

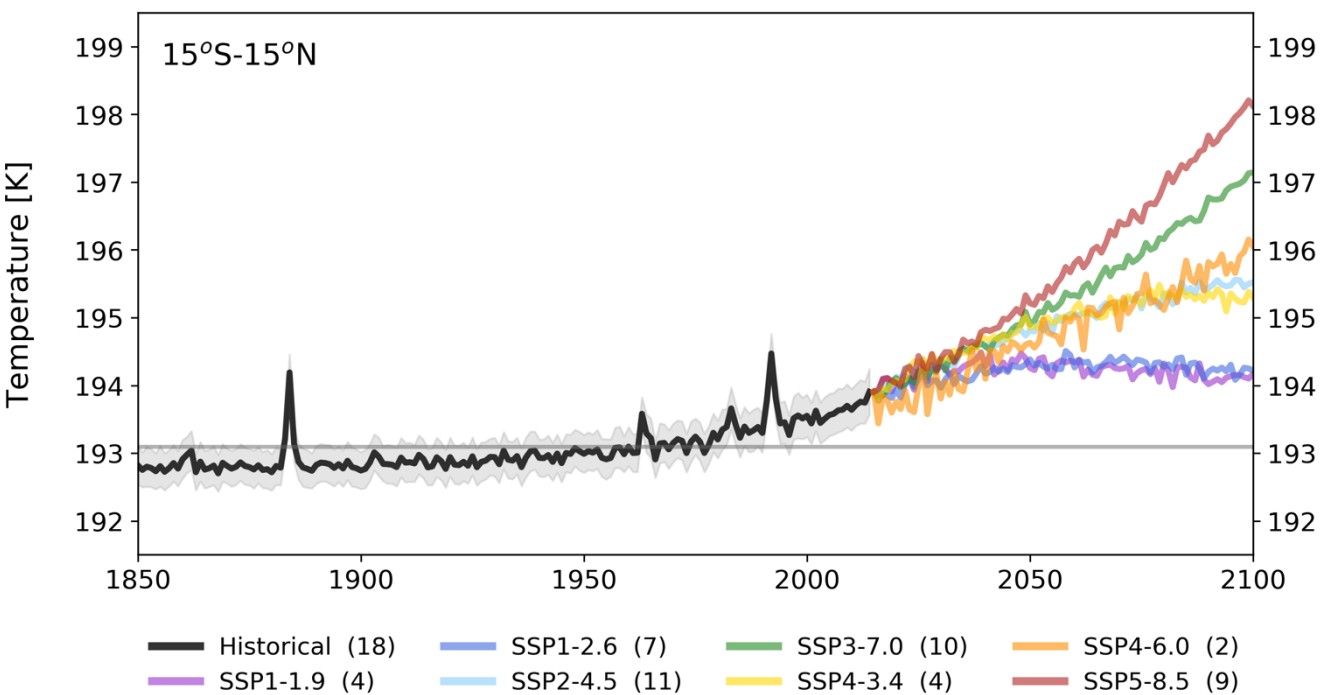

**Figure 16: CMIP6 multi-model mean 100 hPa Temperature (K), averaged from 15°S-15°N, for the historical simulation (black line), and SSP scenarios (coloured lines). The number of models performing each simulation is provide in parentheses in the legend. The light grey envelope indicates the model spread for the historical simulations (calculated as the standard error). Temperature for the 1960 annual mean is given by the horizontal grey line.**

490

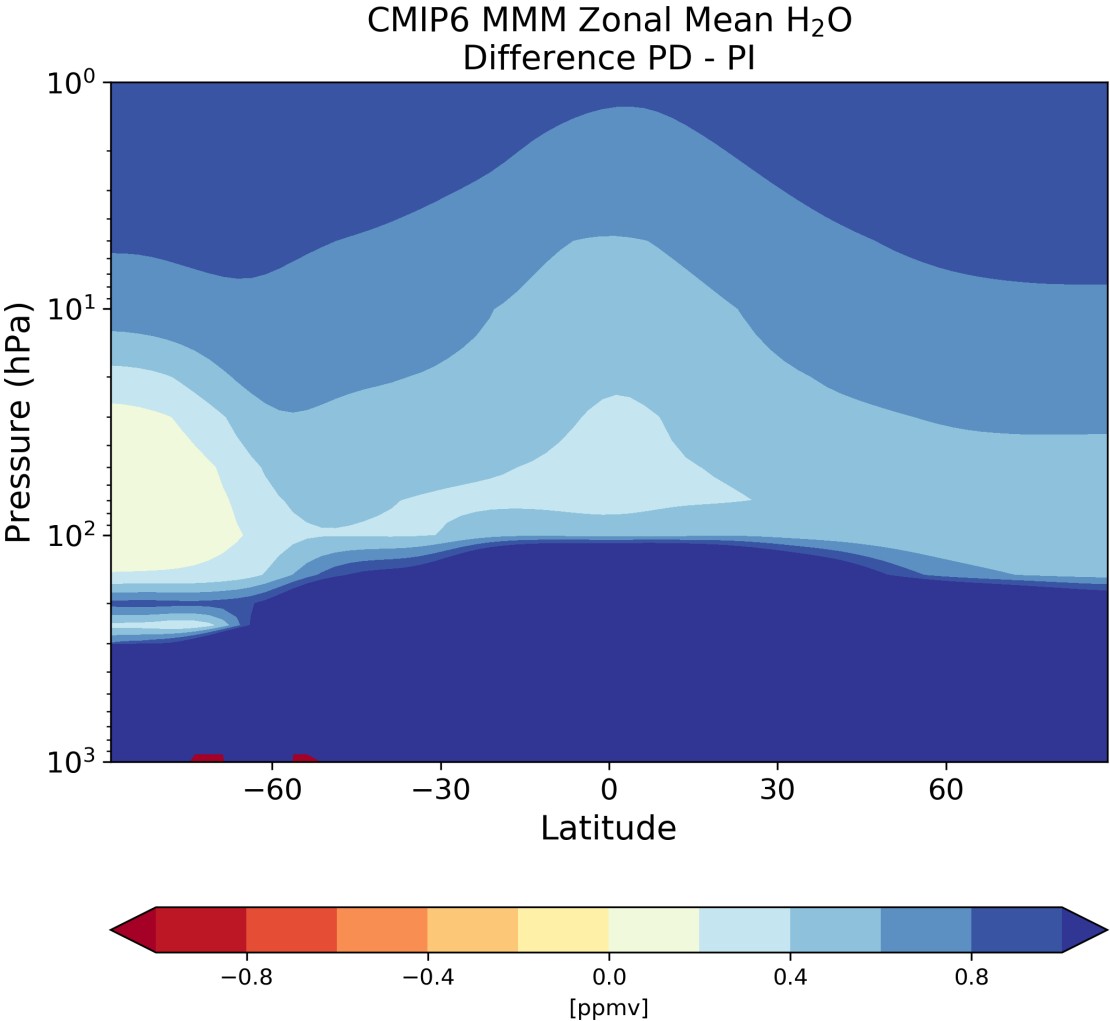

**Figure 17: Historical changes between the pre-industrial (1850-1864 averaged) and present day (2000-2014 averaged) modelled annual mean, zonal mean CMIP6 MMM $H_2O$ mixing ratios (ppmv), calculated using 17 of the 18 CMIP6 models evaluated in this study (GFDL-CM4 was excluded due to its low model top).**

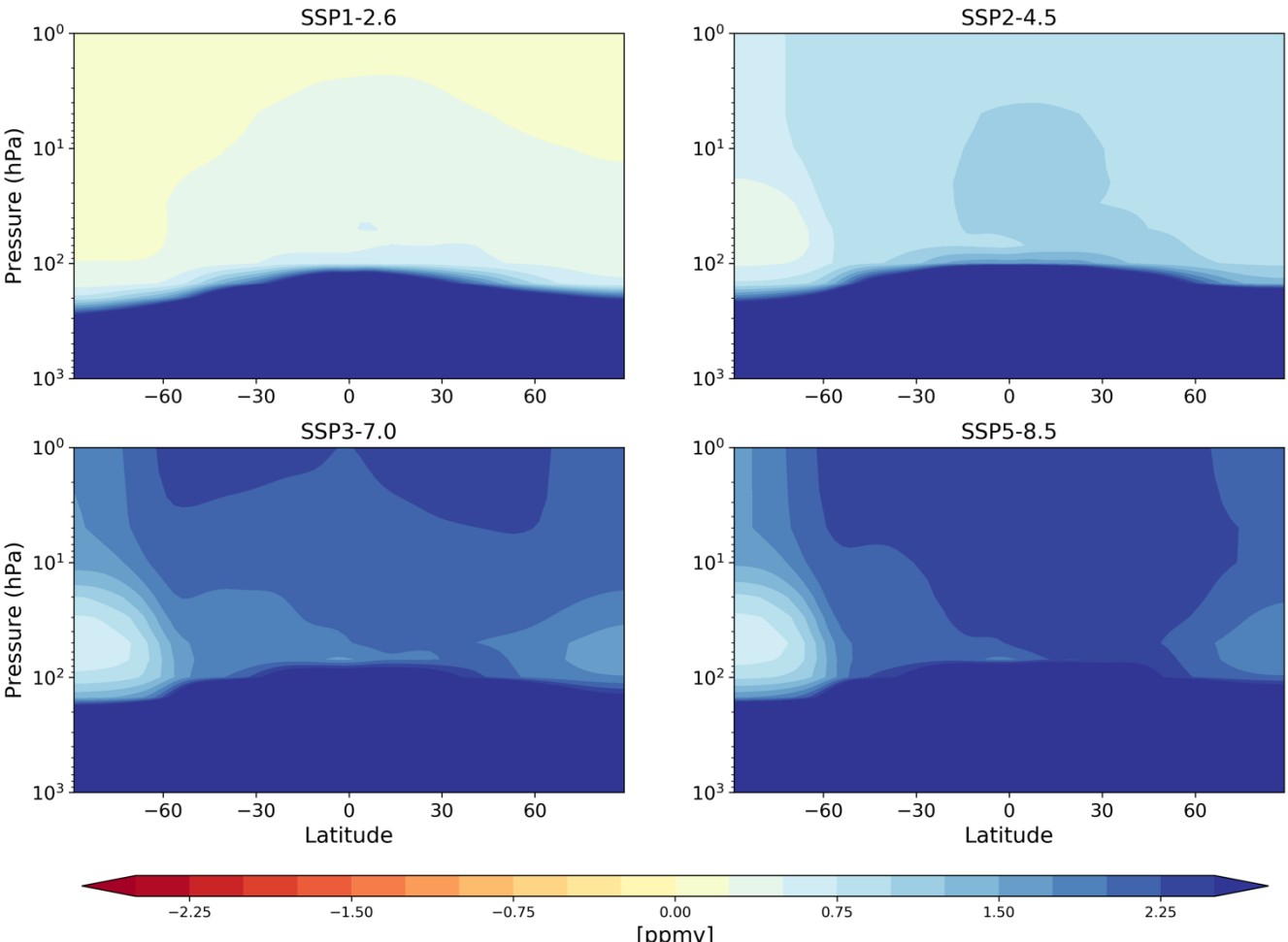

**Figure 18: Projected changes between the present day (2000-2014 averaged) and end of century (2086-2100 averaged) modelled annual mean, zonal mean CMIP6 MMM $H_2O$ volume mixing ratios (ppmv), calculated using 5 of the 18 CMIP6 models evaluated in this study (BCC-CSM2-MR, CNRM-CM6-1, IPSL-CM6A-LR, MRI-ESM2-0 and UKESM1-0-LL). These 5 models were chosen as they had each performed all of the SSPs shown.**

## Appendix

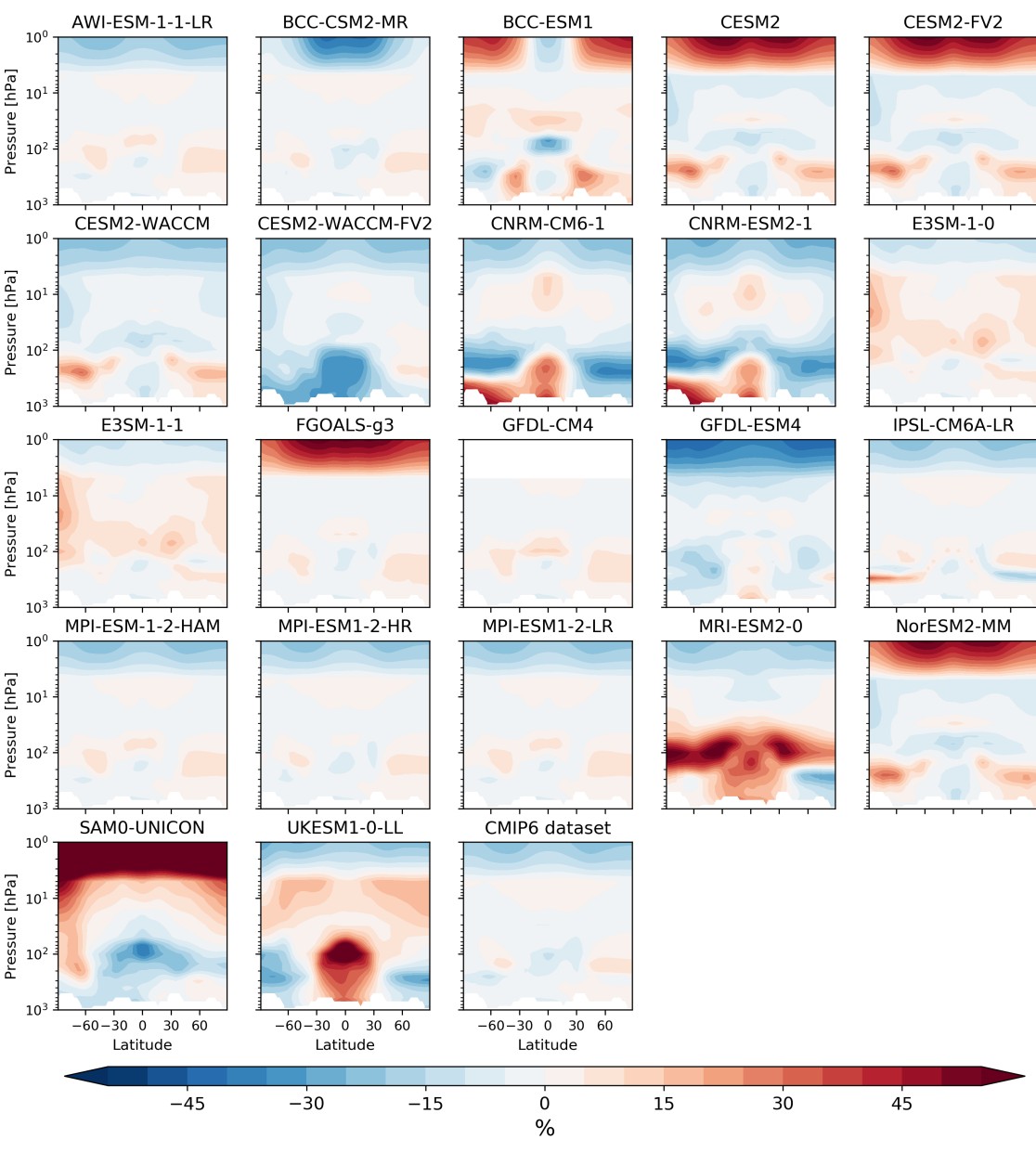

**Figure A1: Latitude vs altitude ozone anomalies (in %) for individual CMIP6 models, and the CMIP6 ozone dataset used by models prescribing stratospheric ozone, compared to the CMIP6 multi-model mean (MMM) averaged for the years 2000-2014. Differences calculated as model minus CMIP6 MMM.**

505

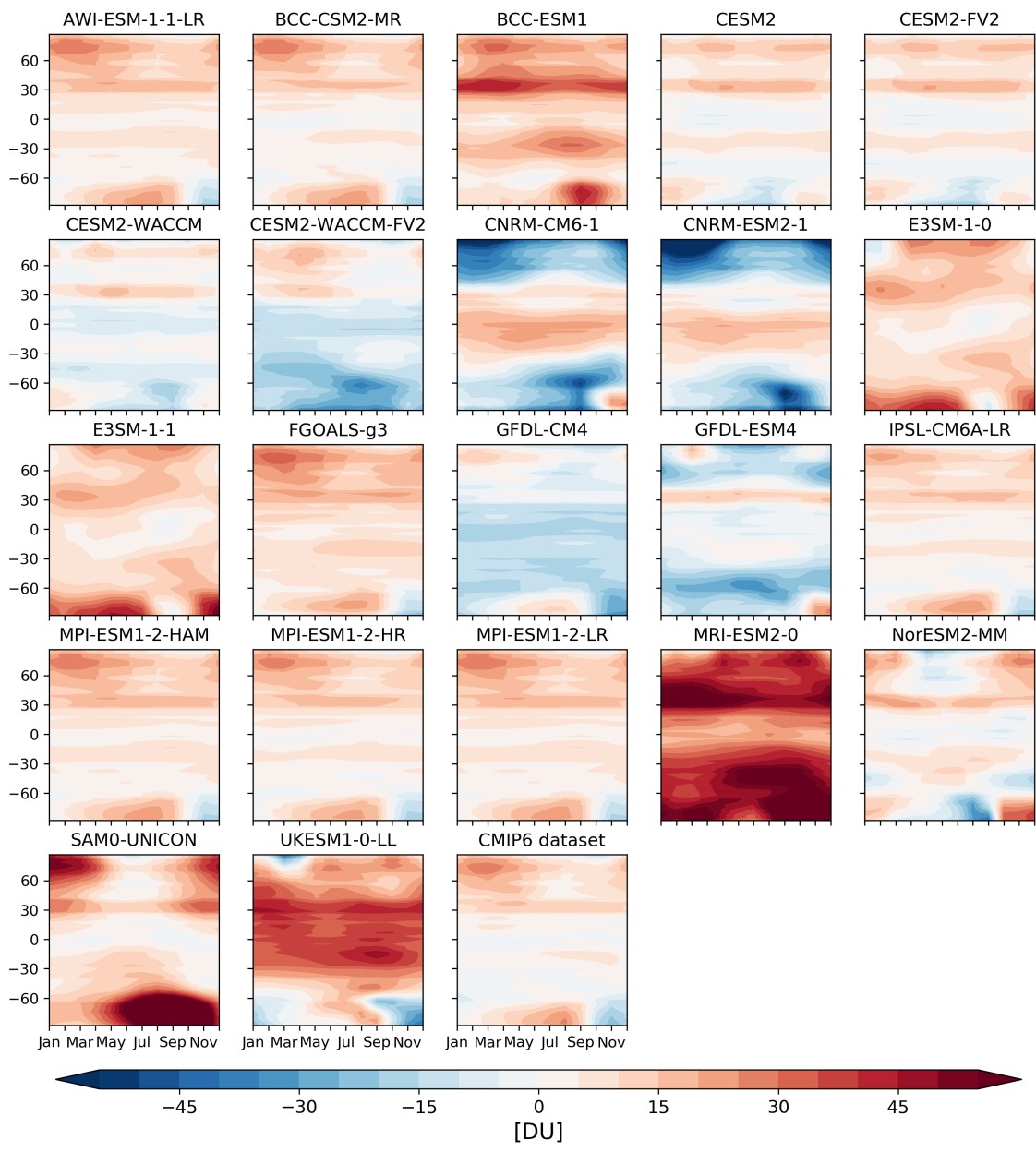

|510

**Figure A2: Climatological total column ozone anomalies (in DU) for individual CMIP6 models, and the CMIP6 ozone dataset used by models prescribing stratospheric ozone, compared to the CMIP6 multi-model mean (MMM) averaged for the years 2000-2014. Differences calculated as model minus CMIP6 MMM.**

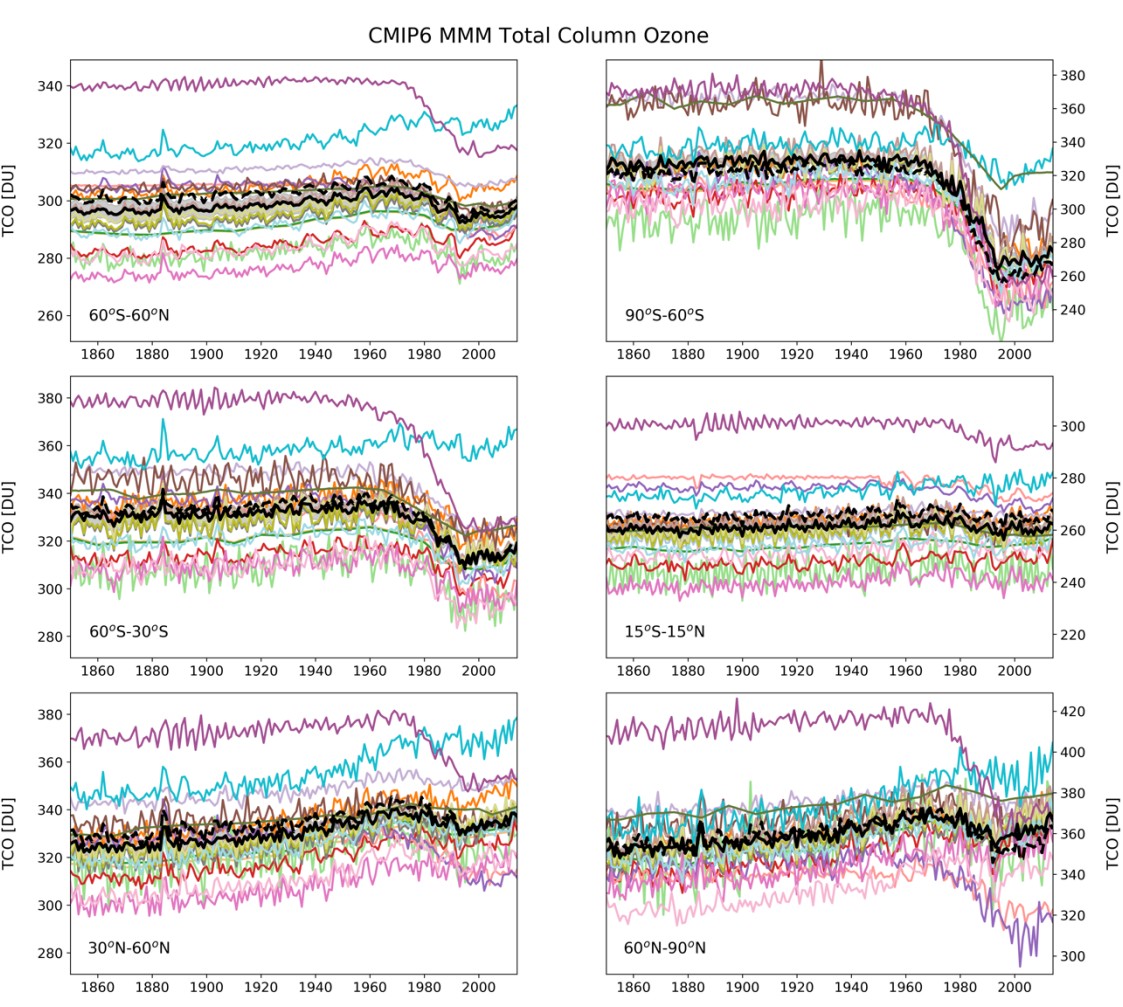

**Figure A3: Regional average total column ozone from each CMIP6 model (coloured lines) for the historical simulation, the CMIP6 multi-model mean (black line), and the multi-model spread (calculated as the standard error; grey shading). Also shown is the mean of models with interactive chemistry schemes (CESM2-WACCM, CESM2-WACCM-FV2, CNRM-ESM2-1, GFDL-ESM4, MRI-ESM2-0, and UKESM1-0-LL; dashed black line)**

### CMIP6 MMM Total Column Ozone

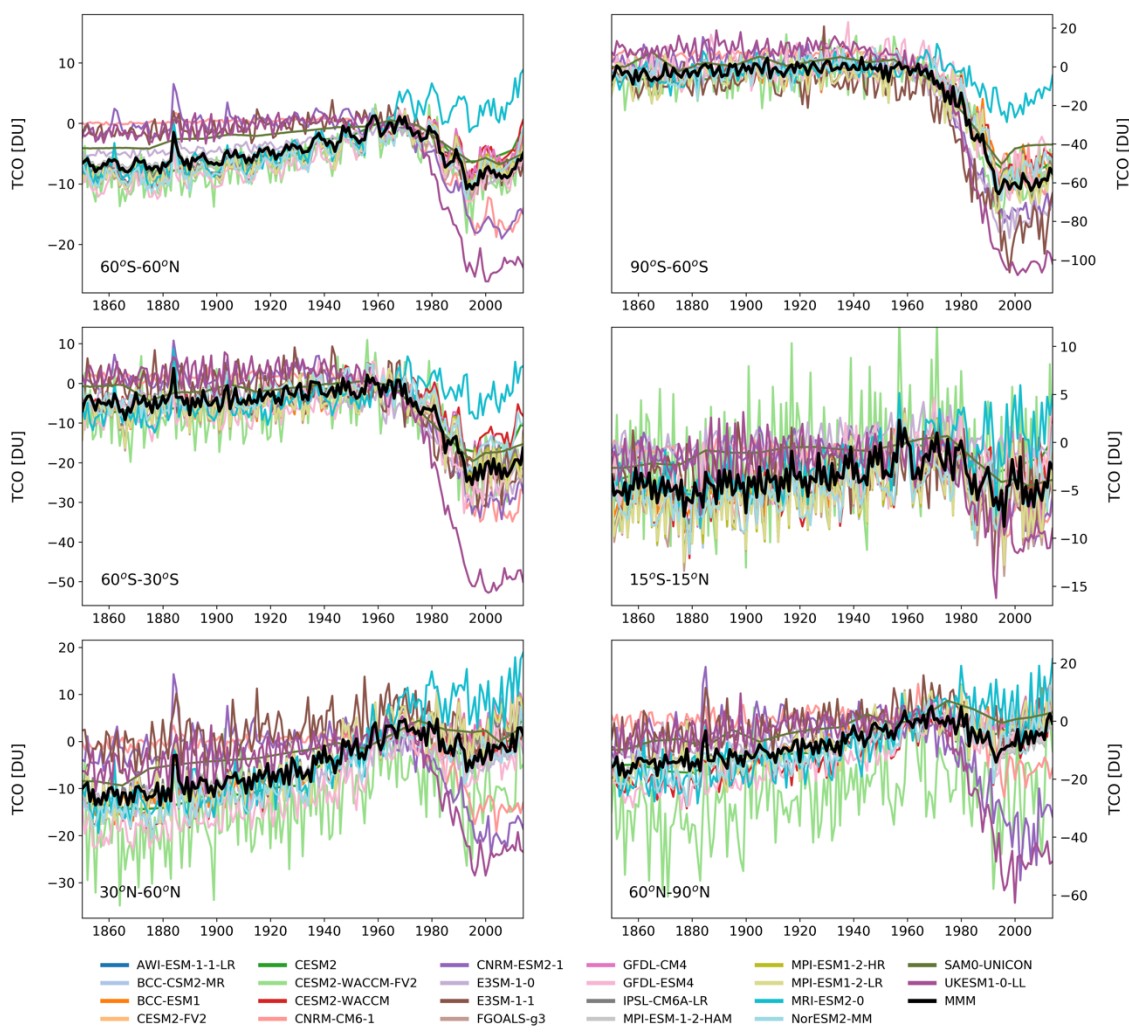

**Figure A4: As for A3, but each model is normalised to its 1960 value.**

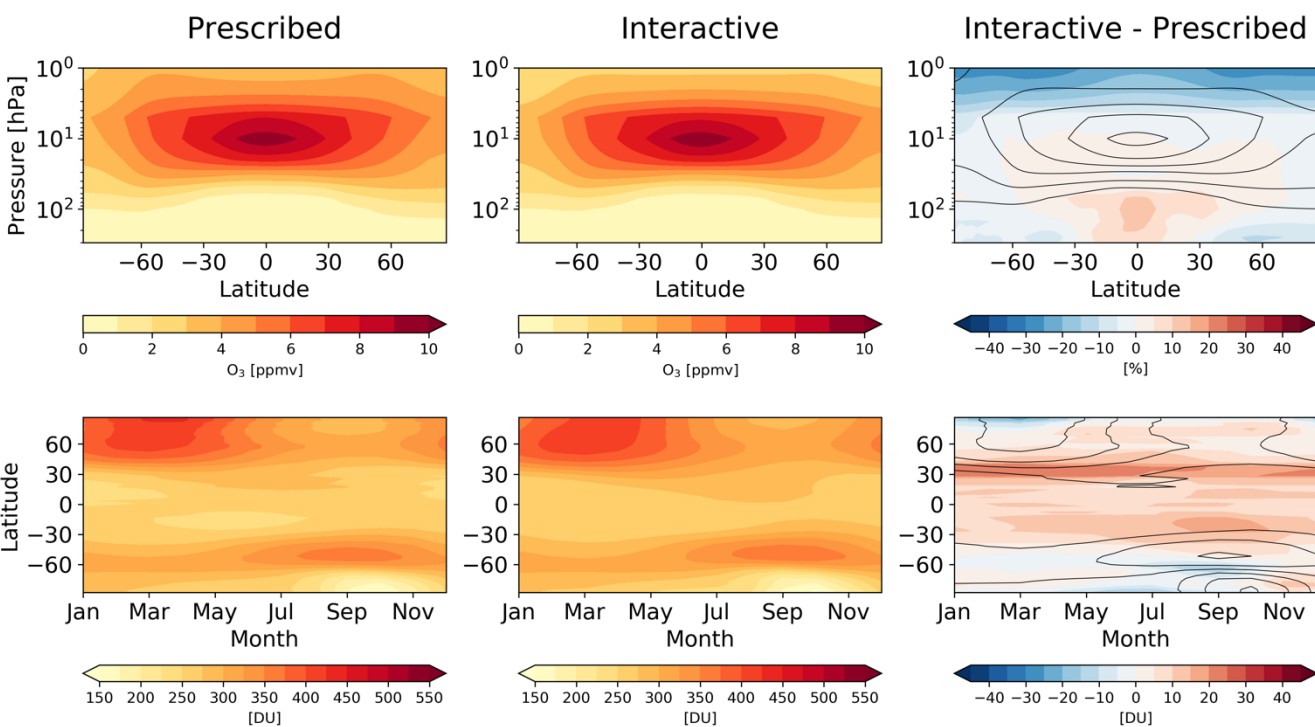

**Figure A5: Top row: 2000-2014 climatological zonal mean ozone for the CMIP6 multi-model mean (MMM) created when using only models prescribing the CMIP6 ozone dataset (Prescribed; left), the CMIP6 MMM created when using only models with interactive chemistry (Interactive; centre) in ppmv, and corresponding differences (right) in %. Bottom row: 2000-2014 climatological total column ozone for the Prescribed mean (left), Interactive mean (centre) and corresponding differences (right) in DU.**

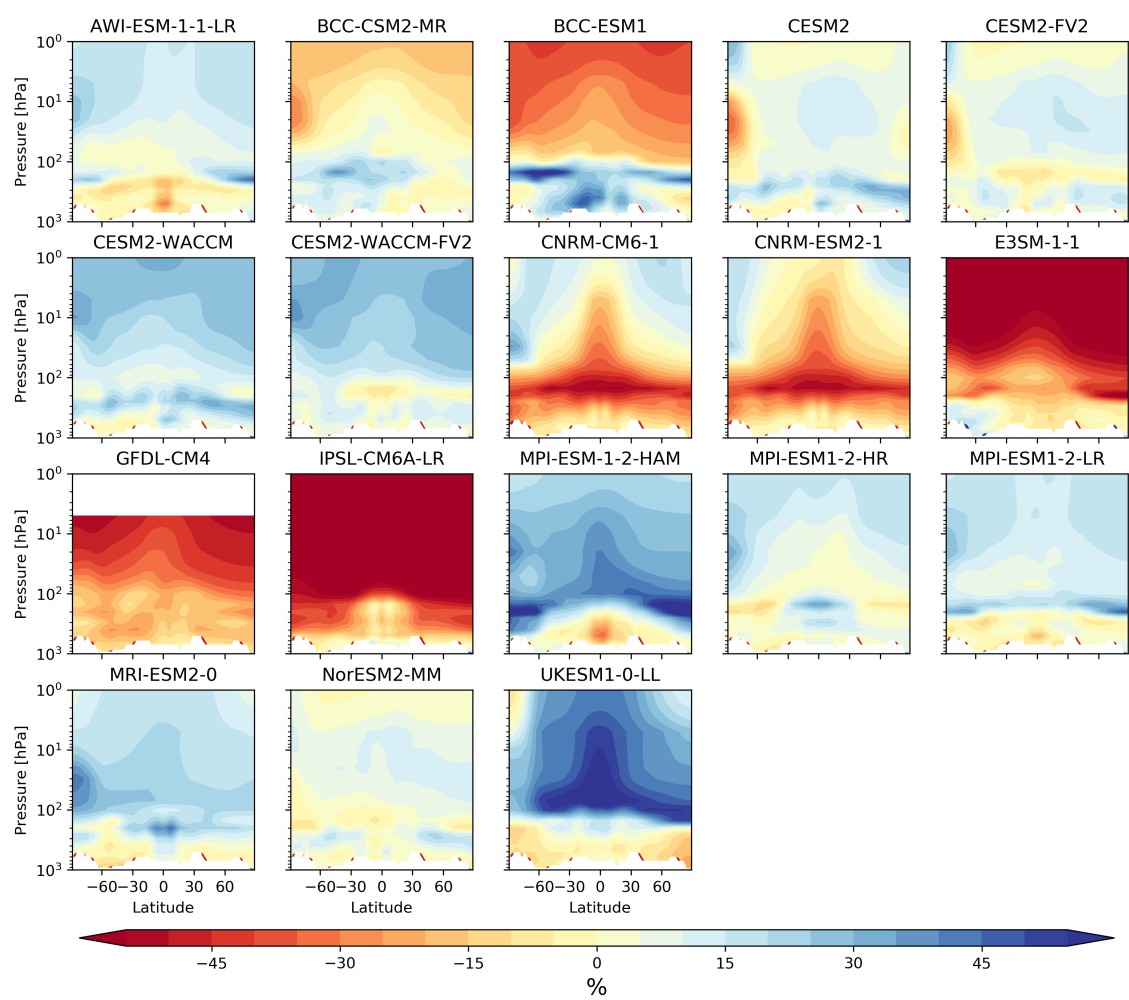

**A6: Latitude vs altitude H$_2$O anomalies (in %) for individual CMIP6 models compared to the CMIP6 multi-model mean (MMM) averaged for the years 2000-2014. Differences calculated as model minus CMIP6 MMM. Note that for the differences, red colours indicate the model is drier (i.e. less H$_2$O in the CMIP6 MMM compared with the observations).**

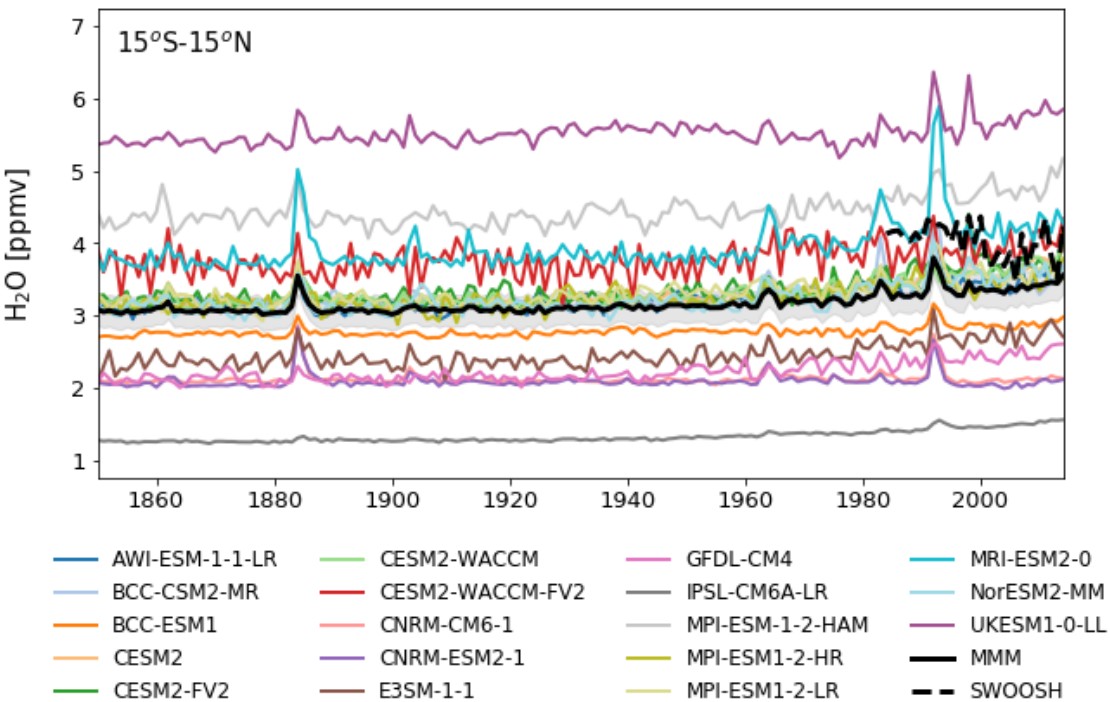

**Figure A7: 70 hPa H2O mixing ratio (ppmv) from each CMIP6 model (coloured lines) for the historical simulation, the CMIP6 multi-model mean (black line), and the multi-model spread (calculated as the standard error; grey shading).**