# Peer review of "Evaluating stratospheric ozone and water vapor changes in CMIP6 models from 1850-2100"

_Atmospheric Chemistry and Physics, 2019_

## Referee Comment (RC1) · Anonymous Referee #2 · 12 Mar 2020

The paper presents a comparison of ozone and H2O fields between different CMIP6 models. It also includes their evaluation against observations. It provides a lot of information that is necessary for the analysis of the climate simulations, notably the contributions of stratospheric ozone and H2O to radiative forcing in CMIP6 coupled climate simulations. I recommend publication after addressing several points. I think the difference between models that calculate interactively ozone and those that specify it should be even clearer in the analysis. The spread among models with specified ozone reflects how important the implementation process of a given ozone climatology into a model can be whereas the spread in the interactive models is of a different nature. I would suggest to add the ""Mean (specified)" in the Tables and add where possible "CMIP6 MMM (specified)" And 'CMIP6 MMM (interactive)' plots (e.g. Figure

4, 6). Here are below more minor comments.

341-343: "Notable differences between the models occur in the uppermost stratosphere, and around the tropopause (Figure A1). The BCC-ESM1, CESM2, FGOALS-g3 and SAM0-UNICON models all simulate much higher ozone mixing ratios in the upper stratosphere."

L75: "which, while consistent with the IPCC-AR5 estimate, represents an increase of ~80% compared to the CMIP5 ozone forcing dataset". Why would a difference of almost a factor 2 indicate consistency between these 2 estimates?

L77: "The relative uncertainties in radiative forcing estimates for both stratospheric ozone and water vapor are large due to the challenges in constraining the concentrations of both during the pre-satellite era". That's not the only problem. If 2 modelling groups are provided with the same preindustrial and present-day fields for ozone and H2O, the radiative forcings calculated by the 2 groups would differ substantially, depending on their radiative schemes, on how the forcing is implemented (notably with respect to the tropopause adjustment), etc...(see literature about model inter-comparisons including literature from some of the authors)

L80-86: I am not sure that this short history of stratospheric chemistry in the middle of the introduction is necessary.

L100:" However, recent research (Polvani et al., 2018, 2019) has shown that stratospheric ozone depletion caused by increasing ODSs has accounted for around half of the acceleration of the BDC in recent decades." It is an estimation based on model simulations. However, there is no clear agreement or at least quantitative agreement between BDC changes in model simulations and observations (BDC tracer proxies). The authors should be more cautious: recent model simulations indicate...ODS may have accounted.

L110: "followed by a sudden decrease of ~10% after 2000 (e.g. Solomon et al., 2010)."

Can models reproduce it? If not, again, one should be cautious with model results regarding H2O evolution.

L159-163: Some caveats should be added here. The fields are "unique, consistent" but it is just 2 models! In view of the inter-model differences and differences with observations (see Figures in the paper), there are biases in those recommended fields, large uncertainties, notably in the projections. Again more caution is required.

L446: "...CMIP6 models to simulation (simulating) pre-industrial TCO".

L446-447: Add a comment. The large differences for preindustrial are not totally surprising. Models are tuned to reproduce the observed ozone evolution (during the satellite era), not the preindustrial era when no observations are available.

L449: "Surprisingly, there is a ∼20 DU range in pre-industrial TCO values between those models prescribing the CMIP6 ozone dataset." Add: suggesting that the TCO is not conserved after model implementation.

L615: The conclusions do not differentiate between ozone interactive and specified models. The spread among models with specified ozone reflects how important the implementation process of a given ozone climatology into a model can be whereas the spread in the interactive models is of a different nature driven by differences between model schemes. The analysis provides important information on both approaches. I think the conclusion should also summarise the conclusions for each approach.

---

## Referee Comment (RC2) · Anonymous Referee #3 · 20 Mar 2020

General Comment

This paper evaluated CMIP6 models in terms of their ability to simulate past to future variation in the stratospheric ozone and water vapor. The authors examined how well the CMIP6 models represent stratospheric ozone profile and past change in total column ozone (TCO) by comparing with the observation in global and regional perspective. They showed the considerably diversified future projections of stratospheric ozone with different SSP scenarios. The stratospheric water vapor in CMIP6 models was revealed to have a couple of problems for adequately simulating its observed features partly because some models ignore methane oxidation in the stratosphere. I fully acknowledge the importance of this work and it will provide a useful information to the climate science community. This paper is rightfully within the scope of ACP, however,

[Figure]

I noticed several issues in this paper which cannot be passed over to be published. I suggested that the authors should consider the following comments: two major and several specific comments.

Major Comment 1:

The authors relatively well described the difference among CMIP6 models, pointing out some models showing over/under-estimation. However, they only provided a limited discussion and description about possible reasons for such a spread among models. As a result, the current manuscript ended up being a superficial model inter-comparison. I recommend the authors to spend more words to discuss why some models differ significantly from the other models. For example, could you discuss more about why UKESM1-0-LL model greatly overestimate the TCO, GFDL models underestimate, MRI-ESM2-0 showed quite a small temporal change in TCO from 1950 onward, and the like?

Major Comment 2:

There is several important information which were not properly provided in the manuscript.

[1] Description of CMIP6 models in chapter 2.1 should provide more unified information on each model. The current descriptions differed considerably between models. At least, 1) model resolution (horizontal and vertical), 2) treatment of ozone-related chemical process both in stratosphere and troposphere and 3) CH4 oxidation in the stratosphere should be provided for all models. Or it's better to include that information in Table 1.

[2] Many statistics used in the manuscript are not well defined. The authors should carefully describe the statistics in the manuscript and figure captions. Sometimes I could not understand what kind of temporal and/or special means were used in some figures, since the descriptions in the manuscript or figure captions are so rough and
blur. The lack of carefulness like this largely deteriorate readability and value of the manuscript. The author should carefully revise the manuscript and figure captions to provide sufficient description about the statistics used. I noticed several points in the following specific comments section.

Specific Comments:

- L80-81: Heterogeneous chemistry is also important.

- L99-100: Could you briefly describe how BDC control the oxidation of Cly, NOy and HOx species.

- L104: Why you didn't use the abbreviation "SWV" here? Please uniformly use the words which you dare to define throughout the manuscript.

- 2.1 Models:

[1] The models are described as "fully coupled", "online", or "interactive" chemistry. You should give precise description what these words mean. If there are no difference among them, you should use one specific word to describe it. I suppose all these words mean that calculated chemical species concentration is used in, so "coupled" or "interactive" with, the radiation calculation. Is it correct? Are there any model who calculate the chemical species "online" but they are not used in radiation calculation? Are there any models who only calculate stratospheric or tropospheric chemistry online but used prescribed concentration in the other sphere? Could you clearly describe these details of each model in this chapter or summarize in Table 1 ?(Please also see Major comment 2 [1])

[2] As to prescribed chemistry models, it should be clearly stated how these models treat the chemical species concentration in the model. I could not understand why these models output different ozone concentration as depicted in the Figures even if they prescribed the same CMIP6 dataset, although I know CESM2 used their original ozone data and so could output different ozone fields. This is one of the key points

for the readers to correctly understand this paper. Particularly, it should be described for each model whether the model prescribed concentrations entire model domain or only prescribed at the surface and allowed to calculate the atmospheric concentrations online.

- L152: Appendix shows the relevant "difference" among models but do not provide the "details" of each model.

- L273: How did each model force the historical changes in short-lived species (mainly air pollutants and its precursors) and long-lived GHG? Whether were they input as emission or surface concentration?

- L287: What are "low" SSPs ?

- L298-305: Any abbreviations should be spelled out at their first appearance, NIWA-BS, SWOOSH and satellite sensors names.

- L322: What is (10o) ?

- L343-345: CESM2 and FGOALS-g3 models showed larger overestimation than BCC-EMS1. Why did you particularly pick up these two (BCC-ESM1 and SAM0-UNICON) models here? Also, SAM0-UNICOM does not have peaks in the mid-latitude.

- Figure2: What is the shaded region?

- L358: It's hard to distinguish each model's line, so I'm not quite sure that I could tell BCC-ESM1 correctly, this model was not low-biased, but "negatively" high-biased. SAM0-UNICON model is also negatively high-biased.

- L362-363: Differences of the CMIP6 MMM from the observation described here in the lower stratosphere and over 1 hPa in the mid-latitudes are not mutual among all CMIP6 models. From Figure A1 it is clear that these differences are mainly owing to only a few models. So the author should describe here more carefully. Putting a figure of standard deviation among models together in Figure 4 might be an option.

- Figure3: It's better to use different color pallet to make it more easy to identify the difference among models.

- L380-381: Is the MMM TCO underestimation in SH polar region in polar winter real? The NIWA-BS data in this area in this season is mainly made by filling missing data as I correctly understand chapter 2.3 of the manuscript, so it might be artificial not real. Can you compare the MMM TCO with other data source, such as ground based TCO observation in SH polar region?

- Figure 5: Figures are a bit small and hard to recognize each symbol. Could you provide the detailed description how did you calculate statistics used in these Figures? It is not self-apparent what "spatial std dev" or "percentage bias" mean. There are several definitions to calculate those statistics. The descriptions can be in appendix or as a supplement material.

- L394-395: Why does a large "spatial" standard deviation for the SMA0-UNICON and MRI-ESM2-0 models indicate higher interannual, so "temporal", variability? (Please also consider my comment for Figure 5)

- L406-407: Why do only these two models show no interannual variability? How about other models who used prescribed ozone fields? (Please also see the comment for "2.1 Models" [2])

- Table2:

[1] The number of ensemble member for each model should be summarized in Table1. Moreover, it must be described somewhere in the manuscript how the ensemble member was treated in all the analysis for this paper. Did you use ensemble means for all the figure?

[2] What does "errors" exactly mean?

[3] Could you also provide the trend of observation (NIWA-BS) for 2000-2014?

- L411: What does "overall TCO decline" exactly mean here?

- L430: The modelled trends in TCO for 2000-2014 are small but not mostly non-significant.

- Figure7 (and for some other figures): Why did you use "standard error" not "standard deviation" for indicating the model spread? The standard deviation is appropriate for this purpose.

- L440: I could not see the TCO increase of "20-30DU" from 1850 to 1960 in NH in Figure 7. Could you revise the number?

- L440-441: English is too complicated for me to understand correctly what it means.

- L446: How did you evaluate the "ability" of models to simulate pre-industrial TCO? Since we don't have TCO observation in that era, we cannot ensure the model's ability through comparing model results with observation.

- L448-454: Could you make more discussion about the difference among the models in simulating the past TCO changes. Discussions on why the models prescribing CMIP6 ozone data showed such a large discrepancy and those on the overestimation of ozone decline by some models are desirable.

- Figure 8: Is the SSP370 scenario simulation result necessary for this figure. This part was never referred in the manuscript.

- L458: Typo. "TCO seen in Figure 8" -> Figure 7

- L460: Where did you describe about a large tropospheric ozone bias of UKESM1-0-LL in the manuscript? Which figure show that?

- L490: Is this for SSP1-1.9 scenario not for SSP1-2.6?

- L515: Could you add the changes in the BDC simulated in the CMIP6 models to Figure 10?

- L529-530: Figure A4 should be cited here if you want to refer to the percentage difference, since Figure 11 cannot show it.

- L535-536: How is the temperature at the tropical tropopause in the CNRM models? Is there any low temperature bias there which can cause the dry bias in the stratosphere in those models?

- L538" As for ?

- L555: What does "CH4" exactly means in this equation? Concentration? Mixing ratio? What is its unit?

- Figure 14: There are no reference to Figure 14 in the manuscript. The figure capture does not include the description of color bar. What does each point in the figure represent? Are they annual mean? Horrible lack of information for this figure.

- Figure 15: Why you did not comment anything on the comparison with the observation. The CMIP6 models apparently underestimate the observation and the modelled increasing trend in the stratospheric water vapor can not be seen in the observation. You should discuss about those comparison in the manuscript.

- L580-582: How is the temperature change at the tropopause not at the 100 hPa? Is the increase at 100 hPa temperature and the increase in water vapor quantitatively consistent with each other?

- L588-590: Could you show separately the relative contribution of 100 hPa temperature rise and CH4 concentration increase for the stratospheric water vapor increase in the CMIP6 models? Both for historical simulation and future projections.

---

## Referee Comment (RC3) · Anonymous Referee #1 · 24 Mar 2020

The manuscript presents an analysis of the evolution of ozone and stratospheric water vapour from the pre-industrial to the present-day (2000 – 2014) and out to 2100 from a number of coupled chemistry climate models that were submitted to CMIP6. In addition, the present-day distribution and seasonal cycle of ozone and water vapour from these models are compared to a number of observational datasets. While the factors controlling the projected evolution of ozone and water vapour seen in these simulations are well known, the presentation of CMIP6-era chemistry climate model simulations is an important update of the literature. In particular, the future projections for the new set of CMIP6 scenarios (SSPs) is welcome.

The paper is well written and the presentation of the results is clear so I do not have any significant concerns about the content. I will point out that one difficult aspect of the

overall presentation is the mixing of models of varying complexity in their representation of atmospheric chemistry. There are five models that include what could be considered a fully prognostic representation of chemistry, with others using specified ozone or linearized chemistry. To complicate the situation further, a couple of the models with specified ozone are closely related to models with prognostic chemistry, having derived ozone from one of the five models with prognostic chemistry in different ways. It makes interpretation of what exactly the multi-model mean represents a bit difficult to fathom. The authors have been generally clear in the description of the models and it is easy to figure out which models contain a full representation of chemistry and which do not. But I would suggest a few minor modifications to help the reader better understand the composition of the multi model ensemble and where the models with prognostic chemistry are significantly different than the other models.

For one, there are a number of models that specify stratospheric ozone using the CMIP6 dataset and yet, see lines 341 – 343 and lines 448-449, some of the models that use the specified CMIP6 ozone show significant differences with each other. Is it possible to include the CMIP6 ozone dataset in a few of the figures comparing the different models? Having the zonal average ozone from CMIP6 dataset shown in Figure 1 and the difference to the multi-model mean in Figure A1, would be very helpful. Perhaps Figure 5 as well?

Secondly, starting at line 538 there is discussion of water vapour in the models, including the behaviour of the CMIP6 multi-model mean that is shown in Figure 12. Given that a number of models do not include a chemical source of water and the fundamentally different behaviour that omitting methane oxidation produces, as seen in Figure 11, I would suggest defining the MMM in Figure 12 as only including those models that include the chemical source of stratospheric water. You really are mixing apples and oranges when you take all ten models that provided water vapour outputs and included them in the MMM. This is much less of a problem for the remaining plots that focus on lower stratospheric water (70 hPa), but for the zonal cross-sections I would suggest

either a separate plot of the mean of only those models that include CH4 oxidation or redefining the MMM plot to only include those same models. I would also suggest a similar segregation of models for Figure 17 since the zonal cross-section of PI to PD changes in water vapour will be fundamentally different depending on whether or not the models account for a chemical source of water vapour. Is this approach already used in Figure 18, where only five of the 10 models are used to construct the multi-model means of water vapour shown there?

My other comments are minor and are itemized below.

Lines 205-206: 'form' should be 'from' in 'instead prescribed form the CMIP6 dataset.'

Line 259: The text states that the SAM0-UNICON model uses specified ozone but doesn't say what the source of the data is: 'Stratospheric and tropospheric ozone is prescribed as a monthly mean 3D field with a specified annual cycle.' Is it CMIP6?

Lines 341 – 343: A couple of models are found to have large differences in ozone in the upper stratosphere relative to the multi-model mean, as shown in Figure A1. In particular BCC-ESM1 and FGOALS-g3 are singled out for having much higher concentrations of ozone in the upper stratosphere, but both of these models base their ozone on the CMIP6-specified ozone dataset in some manner. Does this indicate problems with the CMIP6 ozone dataset or problems in how the models used the data?

Lines 632 – 633: 'However, there is poor agreement between the individual CMIP6 models in the pre-industrial and throughout the historical period, with model TCO values spread across a range of ~60 DU.' To make this clearer I would suggest adding a few words along the lines of 'However, there is poor agreement between the individual CMIP6 models for the absolute magnitude of TCO in the...'

Line 1268: The caption on Figure 8 should state that the average is over 90S – 90N.

---

## Author Comment (AC1) · 30 Nov 2020

We thank the reviewer for their time reviewing the submitted manuscript and for their insightful comments. Please find our responses to each comment below - original reviewer comments in bold, author responses beneath. In addition to addressing the comments from the reviewer, it has also been possible to include in the analysis presented in the revised manuscript 8 additional CMIP6 models that have made available diagnostics since the original submission. Ozone and water vapour data is now available from the AWI-ESM-1-1-LR, CESM2-FV2, CESM2-WACCM-FV2, E3SM-1-1, MPI-ESM-1-2-HAM, MPI-ESM1-2-HR, MPI-ESM1-2-LR and NorESM2-MM models. We have added co-authors to the manuscript from research groups that have prepared and made available the data from these additional models. The inclusion of these models does not change the conclusions of the papers, nor does it significantly change the CMIP6 multi-model mean for the historical period or projections under different SSPs. However, including these models in the revised manuscript gives a more complete evaluation of available CMIP6 models.

**Reviewer 1:**
**The manuscript presents an analysis of the evolution of ozone and stratospheric water vapour from the pre-industrial to the present-day (2000 – 2014) and out to 2100 from a number of coupled chemistry climate models that were submitted to CMIP6. In addition, the present-day distribution and seasonal cycle of ozone and water vapour from these models are compared to a number of observational datasets. While the factors controlling the projected evolution of ozone and water vapour seen in these simulations are well known, the presentation of CMIP6-era chemistry climate model simulations is an important update of the literature. In particular, the future projections for the new set of CMIP6 scenarios (SSPs) is welcome.**

**The paper is well written and the presentation of the results is clear so I do not have any significant concerns about the content. I will point out that one difficult aspect of the overall presentation is the mixing of models of varying complexity in their representation of atmospheric chemistry. There are five models that include what could be considered a fully prognostic representation of chemistry, with others using specified ozone or linearized chemistry. To complicate the situation further, a couple of the models with specified ozone are closely related to models with prognostic chemistry, having derived ozone from one of the five models with prognostic chemistry in different ways. It makes interpretation of what exactly the multi-model mean represents a bit difficult to fathom. The authors have been generally clear in the description of the models and it is easy to figure out which models contain a full representation of chemistry and which do not. But I would suggest a few minor modifications to help the reader better understand the composition of the multi model ensemble and where the models with prognostic chemistry are significantly different than the other models.**

**For one, there are a number of models that specify stratospheric ozone using the CMIP6 dataset and yet, see lines 341 – 343 and lines 448-449, some of the models that use the specified CMIP6 ozone show significant differences with each other. Is it possible to include the CMIP6 ozone dataset in a few of the figures comparing the different models? Having the zonal average ozone from CMIP6 dataset shown in Figure 1 and the difference to the multi-model mean in Figure A1, would be very helpful. Perhaps Figure 5 as well?**
We have added the CMIP6 ozone dataset to Figures 1 and A1. We have also added the CMIP6 ozone dataset to Figure 3 and created a new Figure (A2) which shows the climatological (2000-2014) total column ozone differences between each model and the CMIP6 ozone dataset with respect to the CMIP6 MMM. Together, these figures give a clear indication of how different the CMIP6 ozone dataset is to the CMIP6 MMM. Overall, there are only modest differences between the CMIP6 ozone dataset and the CMIP6 MMM, consistent with the large number of models included in this analysis which prescribe the CMIP6 dataset dominating the MMM.

**Secondly, starting at line 538 there is discussion of water vapour in the models, including the behaviour of the CMIP6 multi-model mean that is shown in Figure 12. Given that a number of models do not include a chemical source of water and the fundamentally different behaviour that omitting methane oxidation produces, as seen in Figure 11, I would suggest defining the MMM in Figure 12 as only including those models that include the chemical source of stratospheric water. You really are mixing apples and oranges when you take all ten models that provided water vapour outputs and included them in the MMM. This is much less of a problem for the remaining plots that focus on lower stratospheric water (70 hPa), but for the zonal cross-sections I would suggest either a separate plot of the mean of only those models that include CH4 oxidation or redefining the MMM plot to only include those same models. I would also suggest a similar segregation of models for Figure 17 since the zonal cross-section of PI to PD changes in water vapour will be fundamentally different depending on whether or not the models account for a chemical source of water vapour. Is this**

**approach already used in Figure 18, where only five of the 10 models are used to construct the multi-model means of water vapour shown there?**

The reviewer raises an important point about the inclusion of models without a water vapour source from CH4 oxidation in the CMIP6 MMM. Excluding models from the multi-model mean is always a challenge, and highlighting that the CMIP6 MMM has lower water vapour mixing ratios in the upper stratosphere, and that this bias is arising from the fact that some models do not include CH4 oxidation, is an important conclusion, and one that climate and Earth system models can use in the future development of their models. As a result, we have left the CMIP6 MMM in Figure 11 as the mean of all the available models. However, to address this comment, we have added a second row to Figure 12 which shows the CMIP6 MMM, the mean of models which include some method of accounting for water vapour formed from the oxidation of CH4, and the differences for both of these means with respect to the SWOOSH dataset. In this way, the revised manuscript now shows both the CMIP6 MMM using all models, and the mean using only models with CH4 oxidation as the reviewer suggests. This new figure is included here and replaces the original Figure 12 in the revised manuscript.

[Figure]

As can be seen from this new Figure 12, there is a steeper vertical gradient in water vapour mixing ratios, and the upper stratospheric bias is reduced with respect to the SWOOSH climatology, but even when models with no treatment of CH4 oxidation are explicitly excluded from the mean, water vapour mixing ratios in the upper stratosphere remain smaller than those in observations. This discussion has been added to the text in the revised manuscript.

For the reviewer's final comment, only models which have run all four SSP scenarios are used for Figure 18, rather than sub-setting of the models based on processes. This was done in order to prevent differences between the future projections arising from the inclusion of different models in each projection.

**My other comments are minor and are itemized below. Lines 205-206: 'form' should be 'from' in 'instead prescribed form the CMIP6 dataset.'**

This has been corrected.

**Line 259: The text states that the SAM0-UNICON model uses specified ozone but doesn't say what the source of the data is: 'Stratospheric and tropospheric ozone is prescribed as a monthly mean 3D field with a specified annual cycle.' Is it CMIP6?**

Yes, the SAM0-UNICON model prescribes the CMIP6 ozone dataset – this has been added to the model description for SAM0-UNICON.

**Lines 341 – 343: A couple of models are found to have large differences in ozone in the upper stratosphere relative to the multi-model mean, as shown in Figure A1. In particular BCC-ESM1 and FGOALS-g3 are singled out for having much higher concentrations of ozone in the upper stratosphere, but both of these models base their ozone on the CMIP6-specified ozone dataset in some manner. Does this indicate problems with the CMIP6 ozone dataset or problems in how the models used the data?**

This issue is related to how models treat ozone in the top boundary levels. We can see from the updated Figures 1 and A1 that there are no significant problems with the top levels of the CMIP6 ozone dataset. However, what

is clear from the analysis presented in the revised manuscript is that models prescribing the CMIP6 ozone dataset are not conserving local ozone mixing ratios of the total column, despite using the same forcing file. Where these differences are coming from is a challenge for each of the modelling centres prescribing the CMIP6 ozone dataset and beyond the scope of this paper to evaluate. We have added to the conclusions of the revised manuscript a few sentences on the differences seen between models ostensibly prescribing the same ozone and highlighted that it is a challenge for the global modelling community to do this in a more robust manner. These sentences state:

'Models which prescribe stratospheric ozone from the CMIP6 ozone dataset show surprisingly large variation in TCO, particularly in the pre-industrial period, at which time there is a ~20 DU range in pre-industrial TCO values between those models prescribing the CMIP6 ozone dataset. There are also large percentage differences between zonal mean ozone fields output by the individual models and the CMIP6 ozone dataset, likely connected to the treatment of the top boundary conditions in different models. Together, this evidence suggests that TCO is not conserved after model implementation of the CMIP6 ozone dataset, and instead small differences are introduced between the models. A future challenge for modelling centres is to prescribe ozone concentrations in such a way as to preserve local mixing ratios and the total column abundance.'

**Lines 632 – 633: 'However, there is poor agreement between the individual CMIP6 models in the pre-industrial and throughout the historical period, with model TCO values spread across a range of ~60 DU.' To make this clearer I would suggest adding a few words along the lines of 'However, there is poor agreement between the individual CMIP6 models for the absolute magnitude of TCO in the...'**
This change has been made, and the sentence now reads 'However, there is poor agreement between the individual CMIP6 models for the absolute magnitude of TCO in the pre-industrial and throughout the historical period, with model TCO values spread across a range of ~60 DU.'

**Line 1268: The caption on Figure 8 should state that the average is over 90S – 90N.**
This information has been added to the Figure caption.

---

## Author Comment (AC2) · 30 Nov 2020

We thank the reviewer for their time reviewing the submitted manuscript and for their insightful comments. Please find our responses to each comment below - original reviewer comments in bold, author responses beneath. In addition to addressing the comments from the reviewer, it has also been possible to include in the analysis presented in the revised manuscript 8 additional CMIP6 models that have made available diagnostics since the original submission. Ozone and water vapour data is now available from the AWI-ESM-1-1-LR, CESM2-FV2, CESM2-WACCM-FV2, E3SM-1-1, MPI-ESM-1-2-HAM, MPI-ESM1-2-HR, MPI-ESM1-2-LR and NorESM2-MM models. We have added co-authors to the manuscript from research groups that have prepared and made available the data from these additional models. The inclusion of these models does not change the conclusions of the papers, nor does it significantly change the CMIP6 multi-model mean for the historical period or projections under different SSPs. However, including these models in the revised manuscript gives a more complete evaluation of available CMIP6 models.

**Reviewer 2:**
**The paper presents a comparison of ozone and H2O fields between different CMIP6 models. It also includes their evaluation against observations. It provides a lot of information that is necessary for the analysis of the climate simulations, notably the contributions of stratospheric ozone and H2O to radiative forcing in CMIP6 coupled climate simulations. I recommend publication after addressing several points. I think the difference between models that calculate interactively ozone and those that specify it should be even clearer in the analysis. The spread among models with specified ozone reflects how important the implementation process of a given ozone climatology into a model can be whereas the spread in the interactive models is of a different nature. I would suggest to add the ""Mean (specified)" in the Tables and add where possible "CMIP6 MMM (specified)" And 'CMIP6 MMM (interactive)" plots (e.g. Figure 4, 6). Here are below more minor comments.**

We have added a new section to the manuscript exploring the differences between models with interactive and non-interactive ozone (section 3.4). As we discuss in that section, some conclusions can be reached regarding these two subsets of models. However, of the 22 models evaluated in this study, only 6 have interactive chemistry. Comparison is further complicated by the fact that the prescribed ozone fields are taken, in part, from a forerunner to the CESM2-WACCM model (in combination with fields from the CMAM model). As a result, significant caveats exist for the conclusions drawn when comparing models with interactive and prescribed ozone fields.

When we calculate a CMIP6 MMM (interactive) mean, it is not dissimilar to the MMM presented in the original manuscript. We have updated Figure A3 in the revised manuscript to include both the CMIP6 MMM and the mean of models with interactive chemistry. We have also produced a version of Figure 4 comparing the climatological (2000-2014) mean of models with prescribed ozone fields to the climatological (2000-2014) mean of models with interactive chemistry:

[Figure]

For the zonal mean, throughout much of the stratosphere differences between these model groupings are less than 5%, although large differences are seen in the uppermost levels and the tropical troposphere. Expressed as a total column difference, the mean of models with interactive chemistry is ~10 DU higher than the mean of models with prescribed ozone in the tropics and midlatitudes. This is most likely due to the inclusion of the UKESM1-0-LL model, which has a significant high TCO bias.

However, while the means are similar, this masks a large range of TCO and zonal mean ozone mixing ratios across the models with interactive ozone chemistry. We have added the Figure above to the revised manuscript, and included a new section (section 3.4) which discusses the figure and the points made here.

**341-343: "Notable differences between the models occur in the uppermost stratosphere, and around the tropopause (Figure A1). The BCC-ESM1, CESM2, FGOALSg3 and SAM0-UNICON models all simulate much higher ozone mixing ratios in the upper stratosphere."**
This has been amended to say 'Notable differences between the models occur in the uppermost stratosphere, and around the tropopause (Figure A1). In the upper stratosphere, the BCC-ESM1, CESM2, CESM2-FV2, FGOALS-g3, NorESM2-MM and SAM0-UNICON models all simulate much higher ozone mixing ratios than the CMIP6 MMM.

**L75: "which, while consistent with the IPCC-AR5 estimate, represents an increase of ~80% compared to the CMIP5 ozone forcing dataset". Why would a difference of almost a factor 2 indicate consistency between these 2 estimates?**
The values of the radiative forcing estimation by R.Checa-Garcia et al, 2018 rely on the CMIP6 ozone concentrations dataset from input4MIPS (and the previous corresponding to CMIP5). These estimations are consistent with the multimodel estimation of IPCC-AR5 report. This is shown in the figure 1 of the paper (lines black and red of the figure on right panels).

https://agupubs.onlinelibrary.wiley.com/cms/asset/e1926f6c-7cf8-4104-b828-b5f183436083/grl57057-fig-0001-m.jpg

However, if we compare the radiative forcing of CMIP5 ozone concentrations dataset and CMIP6 ozone concentrations dataset the last one has an increase of 80%. In other words, the radiative forcing

estimated with CMIP5 ozone concentrations dataset was not consistent with the multimodel estimation of IPCC-AR5 report, meanwhile with CMIP6 we have values close to that multimodel mean.

**L77: "The relative uncertainties in radiative forcing estimates for both stratospheric ozone and water vapor are large due to the challenges in constraining the concentrations of both during the pre satellite era". That's not the only problem. If 2 modelling groups are provided with the same preindustrial and present-day fields for ozone and H2O, the radiative forcings calculated by the 2 groups would differ substantially, depending on their radiative schemes, on how the forcing is implemented (notably with respect to the tropopause adjustment), etc... (see literature about model intercomparisons including literature from some of the authors)**

We agree with the reviewer that there are a number of challenges in calculating radiative forcing associated with ozone and stratospheric water vapour changes. This sentence has been amended to say 'The relative uncertainties in radiative forcing estimates for both stratospheric ozone and water vapor are large due to the challenges in constraining the concentrations of both during the pre-satellite era. As a result, the current radiative forcing estimates rely on ozone and water vapor fields derived from simulations performed by global climate models and Earth system models. However, models use different radiation schemes and model (in the case of models with interactive chemistry scheme) or prescribe (in the case of models without interactive chemistry schemes) ozone differently, further contributing to the uncertainty estimates.'

**L80-86: I am not sure that this short history of stratospheric chemistry in the middle of the introduction is necessary.**

We include this short section on stratospheric ozone chemistry as the paper is aimed at the CMIP6/IPCC community, which covers a diverse range of academic backgrounds. The results of this study are of interest, for example, to the radiative forcing community, which may not be familiar with the details of stratospheric ozone chemistry. As such, we feel the inclusion of these 6 lines provides broader background material, but is short enough so as not to disrupt the flow of the manuscript.

**L100:" However, recent research (Polvani et al., 2018, 2019) has shown that stratospheric ozone depletion caused by increasing ODSs has accounted for around half of the acceleration of the BDC in recent decades." It is an estimation based on model simulations. However, there is no clear agreement or at least quantitative agreement between BDC changes in model simulations and observations (BDC tracer proxies). The authors should be more cautious: recent model simulations indicate. . .ODS may have accounted.**

We have added this to the manuscript. The sentence now reads 'However, recent research (Polvani et al., 2018, 2019) has shown, using model simulations, that stratospheric ozone depletion caused by increasing ODSs may have accounted for around half of the acceleration of the BDC in recent decades.'

**L110: "followed by a sudden decrease of ~10% after 2000 (e.g. Solomon et al., 2010)." Can models reproduce it? If not, again, one should be cautious with model results regarding H2O evolution.**

Solomon et al. highlight that the decrease in stratospheric water vapour after 2000 is associated with, among other things, anomalies in sea surface temperatures. As the models do not use historic sea surface temperatures, we do not expect them to have a similar decrease at the same time. However, the reviewer is correct to say that when evaluating stratospheric water vapour in Earth system models asking, 'do the models capture the observed annual to decadal variability in H2O mixing ratios?' is an important question. Here we focus on longer term (decadal to centennial) changes in water vapour, and so it is beyond the scope of this study to evaluate this fully, and we hope that other research on CMIP6 models will assess annual to decadal variability, and the drivers of this variability, in detail.

**L159-163: Some caveats should be added here. The fields are "unique, consistent" but it is just 2 models! In view of the inter-model differences and differences with observations (see Figures in the**

**paper), there are biases in those recommended fields, large uncertainties, notably in the projections. Again more caution is required.**

We have added the following text to the end of the section discussing the CMIP6 ozone dataset 'Not that, as the CMIP6 dataset uses values from model simulations, it has biases with respect to observations and uncertainties associated with the projections of stratospheric ozone beyond the period observations exist for, both from the pre-industrial to the start of the observational record, and from the present day to the end of the 21$^{st}$ century under the different SSP scenarios.'

**L446: ". . .CMIP6 models to simulation (simulating) pre-industrial TCO".**

This has been changed to 'There is poor agreement in the simulation of pre-industrial TCO across CMIP6 models'

**L446-447: Add a comment. The large differences for preindustrial are not totally surprising. Models are tuned to reproduce the observed ozone evolution (during the satellite era), not the preindustrial era when no observations are available.**

We feel it is not necessary to explicitly state that differences in the pre-industrial are not surprising – as the reviewer says models tend to better agree during periods of observations and differ outside of these periods. Whether this is an explicit tuning the models undergo or an unconscious result of the development of the chemistry scheme is harder to say. Additionally, while we may expect the models to disagree during the pre-industrial, the degree to which they disagree (spanning a range of 75 DU) perhaps is surprising.

**L449: "Surprisingly, there is a ~20 DU range in pre-industrial TCO values between those models prescribing the CMIP6 ozone dataset." Add: suggesting that the TCO is not conserved after model implementation.**

This has been added.

**L615: The conclusions do not differentiate between ozone interactive and specified models. The spread among models with specified ozone reflects how important the implementation process of a given ozone climatology into a model can be whereas the spread in the interactive models is of a different nature driven by differences between model schemes. The analysis provides important information on both approaches. I think the conclusion should also summarise the conclusions for each approach.**

We have added a paragraph to the conclusions section summarising the new section which compares models with interactive and non-interactive ozone fields.

---

## Author Comment (AC3) · 30 Nov 2020

We thank the reviewer for their time reviewing the submitted manuscript and for their insightful comments. Please find our responses to each comment below - original reviewer comments in bold, author responses beneath. In addition to addressing the comments from the reviewer, it has also been possible to include in the analysis presented in the revised manuscript 8 additional CMIP6 models that have made available diagnostics since the original submission. Ozone and water vapour data is now available from the AWI-ESM-1-1-LR, CESM2-FV2, CESM2-WACCM-FV2, E3SM-1-1, MPI-ESM-1-2-HAM, MPI-ESM1-2-HR, MPI-ESM1-2-LR and NorESM2-MM models. We have added co-authors to the manuscript from research groups that have prepared and made available the data from these additional models. The inclusion of these models does not change the conclusions of the papers, nor does it significantly change the CMIP6 multi-model mean for the historical period or projections under different SSPs. However, including these models in the revised manuscript gives a more complete evaluation of available CMIP6 models.

**Reviewer 3:**

**General Comment**

**This paper evaluated CMIP6 models in terms of their ability to simulate past to future variation in the stratospheric ozone and water vapor. The authors examined how well the CMIP6 models represent stratospheric ozone profile and past change in total column ozone (TCO) by comparing with the observation in global and regional perspective. They showed the considerably diversified future projections of stratospheric ozone with different SSP scenarios. The stratospheric water vapor in CMIP6 models was revealed to have a couple of problems for adequately simulating its observed features partly because some models ignore methane oxidation in the stratosphere. I fully acknowledge the importance of this work and it will provide a useful information to the climate science community. This paper is rightfully within the scope of ACP, however, I noticed several issues in this paper which cannot be passed over to be published. I suggested that the authors should consider the following comments: two major and several specific comments.**

**Major Comment 1:**

**The authors relatively well described the difference among CMIP6 models, pointing out some models showing over/under-estimation. However, they only provided a limited discussion and description about possible reasons for such a spread among models. As a result, the current manuscript ended up being a superficial model intercomparison. I recommend the authors to spend more words to discuss why some models differ significantly from the other models. For example, could you discuss more about why UKESM1-0-LL model greatly overestimate the TCO, GFDL models underestimate, MRI-ESM2-0 showed quite a small temporal change in TCO from 1950 onward, and the like?**

We have added a new section (section 3.4) to the manuscript comparing models with interactive and prescribed ozone fields, which further details the differences between the models. However, the largest ozone differences are seen between models with interactive chemistry and identifying the causes of these biases is a significant challenge, requiring access to the models' source code and preforming a number of tests. Further, significant differences exist between models with prescribed ozone fields, which is likely related to how those models process and implement the CMIP6 ozone dataset. As such, identifying why a model has a significant high or low bias with respect to the CMIP6 multimodel mean is a significant challenge and beyond the scope of this study. Instead, we aim to evaluate how TCO and stratospheric water vapour compare to observations in these CMIP6 models, and how they are projected to change from the pre-industrial to the present day, and into the future, so that this information may be used when comparing, for example, radiative forcing and regional climate change across the CMIP6 models evaluated here.

**Major Comment 2:**

**There is several important information which were not properly provided in the manuscript.**

**[1] Description of CMIP6 models in chapter 2.1 should provide more unified information on each model. The current descriptions differed considerably between models. At least, 1) model**

**resolution (horizontal and vertical), 2) treatment of ozone-related chemical process both in stratosphere and troposphere and 3) CH4 oxidation in the stratosphere should be provided for all models. Or it's better to include that information in Table 1.**

There is no set definition for what makes a model, for example, an Earth system model and so it hard to standardise the text across the 22 models evaluated here. Some models, for example, include some Earth system components (e.g., atmospheric chemistry, ocean biogeochemical cycles, carbon and nitrogen cycles, interactive fire schemes), but exclude others. Similarly, some models use fairly advanced chemistry schemes with 10s of chemical species and 100s of reactions, while others use simplified schemes. As such, we have worked with the individual modelling groups to write the text for the model descriptions so that the modelling centres are happy with the description of their model and point towards the full documentation for each model so that the readers can further investigate which components are included in each model and the complexity of these components. We have expanded table 1 to include more information on each of the models evaluated here to cover the reviewers 3 points.

**[2] Many statistics used in the manuscript are not well defined. The authors should carefully describe the statistics in the manuscript and figure captions. Sometimes I could not understand what kind of temporal and/or special means were used in some figures, since the descriptions in the manuscript or figure captions are so rough and blur. The lack of carefulness like this largely deteriorate readability and value of the manuscript. The author should carefully revise the manuscript and figure captions to provide sufficient description about the statistics used. I noticed several points in the following specific comments section.**

Greater care has been taken to describe better the statistics and averaging used in analysis throughout the manuscript and in the figure captions.

**Specific Comments:**

**- L80-81: Heterogeneous chemistry is also important.**

This information is included in the last line of this paragraph, which states 'Heterogeneous processes play a major role in determining ozone 85 abundances in the polar lower stratosphere (e.g. Solomon, 1999) and following large volcanic eruptions (e.g. Solomon et al., 1996; Telford et al., 2009)'

**- L99-100: Could you briefly describe how BDC control the oxidation of Cly, NOy and HOx species.**

The BDC controls the oxidation of these species by transporting source gases (such as CFC-11, CFC-12, N2O and CH4) from the lower atmosphere, where they are chemically inert/have long lifetimes, to the middle and upper stratosphere, where they are more rapidly oxidised. The faster the BDC the greater the mass flux through the region of oxidation, and so the shorter the chemical lifetime of the species. We have modified the sentence to read '…and by influencing the chemical lifetimes of $Cl_y$, $NO_y$ and $HO_x$, source gases (e.g. Revell et al., 2012; Meul et al., 2014).'

**- L104: Why you didn't use the abbreviation "SWV" here? Please uniformly use the words which you dare to define throughout the manuscript.**

We have revised the manuscript to use 'stratospheric water vapour' throughout.

**- 2.1 Models:**
**[1] The models are described as "fully coupled", "online", or "interactive" chemistry. You should give precise description what these words mean. If there are no difference among them, you should use one specific word to describe it. I suppose all these words mean that calculated chemical species concentration is used in, so "coupled" or "interactive" with, the radiation calculation. Is it correct? Are there any model who calculate the chemical species "online" but they are not used in radiation calculation? Are there any models who only calculate stratospheric or tropospheric chemistry online**

**but used prescribed concentration in the other sphere? Could you clearly describe these details of each model in this chapter or summarize in Table 1 ?(Please also see Major comment 2 [1])**
We have added more information on each model to Table 1. Please see also our response to your major comment above.

**[2] As to prescribed chemistry models, it should be clearly stated how these models treat the chemical species concentration in the model. I could not understand why these models output different ozone concentration as depicted in the Figures even if they prescribed the same CMIP6 dataset, although I know CESM2 used their original ozone data and so could output different ozone fields. This is one of the key points for the readers to correctly understand this paper. Particularly, it should be described for each model whether the model prescribed concentrations entire model domain or only prescribed at the surface and allowed to calculate the atmospheric concentrations online.**
This is a key finding for the manuscript – that models prescribing the CMIP6 ozone dataset do not agree in terms of zonal mean ozone mixing ratios and the total column. This is likely due to the numerous regridding phases the ozone fields go through. The ozone field is first regridded from the resolution of the CMIP6 ozone dataset provided by Input4MIPs (96 lats x 144 lons x 66 pressure levels) to the native model grid. The model out is then regridded to the 19 pressure levels used in the Amon output grid specified by CMIP6. Additionally, there are processing steps the models employ (redistributing the ozone dataset to match the model tropopause, prescribing only stratospheric ozone but modelling interactively tropospheric ozone) which also result in differences between the models prescribing the CMIP6 ozone dataset. However, while these processes clearly play an important role, it is beyond the scope of this study to identify where these differences arise from. Instead, we document these differences and argue that, when prescribing ozone fields, greater care should be taken to ensure that the total ozone column is conserved.

**- L152: Appendix shows the relevant "difference" among models but do not provide the "details" of each model.**
This has been changed to 'Relevant details of each model are provided below, and a summary is provided in Table 1.' This error came about as the model discriptions in section 2.1 were originally in the appendix, but were moved before submission. However, this sentence was not updated to reflect this.

**- L273: How did each model force the historical changes in short-lived species (mainly air pollutants and its precursors) and long-lived GHG? Whether were they input as emission or surface concentration?**
Many models which include interactive chemistry schemes use a mixed approach here, often using emissions for short lived species with high spatial variability (e.g. NOx, CO, VOCs) while prescribing surface concentrations of long lived (e.g. CO2, CFC-11, N2O). We do not aim here to give a detailed description of how each model is set up and the processes they include, simply because this would be a huge task and repeat information that is available elsewhere in the published literature. Instead, we sought to provide key information on how the models treat stratospheric ozone and water vapour and also provide references for each of the models evaluated in this study should more detailed information be required.

**- L287: What are "low" SSPs?**
For clarity, this has been changed to 'low numbered SSPs (i.e., SSP1 and SSP2) assume lower abundances of long-lived GHGs'

**- L298-305: Any abbreviations should be spelled out at their first appearance, NIWABS, SWOOSH and satellite sensors names.**

This has been done in the revised manuscript.

**- L322: What is (10o)?**
This should say '10° latitudinal resolution' and has been corrected in the revised manuscript.

**- L343-345: CESM2 and FGOALS-g3 models showed larger overestimation than BCCEMS1. Why did you particularly pick up these two (BCC-ESM1 and SAM0-UNICON) models here? Also, SAM0-UNICOM does not have peaks in the mid-latitude.**
These two models were singled out as having not just high biases in the upper stratosphere but also different spatial patterns, with significant biases particularly in the mid latitudes. To clarify, we have changed these sentences to 'Notable differences between the models occur in the uppermost stratosphere, and around the tropopause (Figure A1). In the upper stratosphere, the BCC-ESM1, CESM2, CESM2-FV2, FGOALS-g3, NorESM2-MM and SAM0-UNICON models all simulate much higher ozone mixing ratios than the CMIP6 MMM. Additionally, the BCC-ESM1 and SAM0-UNICON models also have a different spatial structure in the distribution of ozone at these levels, with maxima in the mid-latitudes at 1 hPa'

**- Figure2: What is the shaded region?**
The shaded region represents the standard error. As discussed for the major point above, we have gone through the manuscript and added more detail on the statistics used in the manuscript and added more detail to the figure captions.

**- L358: It's hard to distinguish each model's line, so I'm not quite sure that I could tell BCC-ESM1 correctly, this model was not low-biased, but "negatively" high-biased. SAM0-UNICON model is also negatively high-biased.**
Low biased is used here to indicate that the modelled values for the BCC-ESM1 and SAM0-UNICON models are lower than both the observations and CMIP6 MMM.

**- L362-363: Differences of the CMIP6 MMM from the observation described here in the lower stratosphere and over 1 hPa in the mid-latitudes are not mutual among all CMIP6 models. From Figure A1 it is clear that these differences are mainly owing to only a few models. So the author should describe here more carefully. Putting a figure of standard deviation among models together in Figure 4 might be an option.**
We have added more text to the discussion of Figure 4 stating that while the MMM differences are as described, they arise from different biases from each model contributing to the CMIP6 MMM.

**- Figure3: It's better to use different color pallet to make it more easy to identify the difference among models.**
The YlOrRd colour pallet is used throughout this study as it is i) colour-blind friendly, and ii) the colour pallet identified for use by IPCC, and as such has been recommended for analysis of CMIP6 models. We agree with the reviewer that many of the panels look the same, but that is a positive – most CMIP6 models accurately capture the magnitude and seasonal evolution of TCO. However, differences can be seen – SAM0-UNICON and MRI-ESM2-0 have shallower ozone holes, while the CNRM models have lower springtime arctic polar ozone. This differences are further explored in the subsequent analysis, but figure 3 is intended to demonstrate that no model so significantly misrepresents TCO as to be a clear outlier.

**- L380-381: Is the MMM TCO underestimation in SH polar region in polar winter real? The NIWA-BS data in this area in this season is mainly made by filling missing data as I correctly understand chapter 2.3 of the manuscript, so it might be artificial not real. Can you compare the MMM TCO with other data source, such as ground based TCO observation in SH polar region?**

As the reviewer states, in the region of the polar night, where large baises are seen between the CMIP6 MMM and the Bodeker dataset, the 'observations' rely on a filling routine described in the manuscript. To prevent over-interpretation of this difference we have changed Figure 4 to use the unpatched version of the Bodeker dataset, and modified the text accordingly.

**- Figure 5: Figures are a bit small and hard to recognize each symbol. Could you provide the detailed description how did you calculate statistics used in these Figures? It is not self-apparent what "spatial std dev" or "percentage bias" mean. There are several definitions to calculate those statistics. The descriptions can be in appendix or as a supplement material.**

**- L394-395: Why does a large "spatial" standard deviation for the SMA0-UNICON and MRI-ESM2-0 models indicate higher interannual, so "temporal", variability? (Please also consider my comment for Figure 5)**

**- L406-407: Why do only these two models show no interannual variability? How about other models who used prescribed ozone fields? (Please also see the comment for "2.1 Models" [2])**
We are not sure why these models show no interannual variability, and have passed this observation on to the relevant modelling groups.

**- Table2:**
**[1] The number of ensemble member for each model should be summarized in Table1.**
This information has been added to table 1.

**Moreover, it must be described somewhere in the manuscript how the ensemble member was treated in all the analysis for this paper. Did you use ensemble means for all the figure?**
Yes, we use all available ensemble members for each model to create a single ensemble mean for that model and then this mean is shown in all the figures. The CMIP6 MMM is then created from the individual model ensemble means. This process was adopted to prevent a model with many ensemble members dominating the ensemble mean – instead each model counts equally towards the MMM. This has been added to section 2.

**[2] What does "errors" exactly mean?**
It is the statistical uncertainty of the trends at 68% (1 sigma) confidence level. This has been added to the footnote of Table 2.

**[3] Could you also provide the trend of observation (NIWA-BS) for 2000-2014?**
Calculating the trends for the observations and comparing those accurately with the models is a challenge due to the issues surrounding TCO in the polar night. The models provide full lat lon domains throughout the year, and so the global mean annual mean trend is truly global. For the observations we have the option of using the patched NIWA-BS dataset (which is effectively using a model to fill in the values in the polar night), or the unpatched version, which has missing data in this region. Either way, we would not be comparing like with like, and as the recovery trends are so small (and generally not statistically significant) these differences are relatively important. A detailed evaluation of trends in the NIWA-BS dataset can be found at:
Bodeker, G. E., Nitzbon, J., Tradowsky, J. S., Kremser, S., Schwertheim, A., and Lewis, J.: A Global Total Column Ozone Climate Data Record, Earth Syst. Sci. Data Discuss., https://doi.org/10.5194/essd-2020-218, in review, 2020.

**- L411: What does "overall TCO decline" exactly mean here?**
This has been changed to 'the decrease in TCO between 1980 and 2000'

**- L430: The modelled trends in TCO for 2000-2014 are small but not mostly nonsignificant.**
We have edited this paragraph to say 'Over the period 2000-2014, generally, models show non-significant (at the 95% confidence level) positive trends in TCO. However, nine models show significant albeit weak positive trends in global TCO, of which three are INTERACTIVE models (CESM2-WACCM, MRI-ESM2-0, and UKESM1-0-LL). The significant positive trends calculated in these models show the largest positive trends in both the NH and the SH high latitudes and moderate positive trends in mid-latitudes (Table 2). The INTERACTIVE models collectively show stronger positive trends in all regions, compared to the all-model mean. Significant and the strongest positive ozone trends in the SH high latitudes occur in MRI-ESM2-0, NorESM2-MM, and UKESM1-0-LL, whereas significant and the strongest positive trends in the NH high latitudes occur in CESM2-WACCM, NorESM2-MM, and UKESM1-0-LL. Significant but weaker positive trends also occur in SAM0-UNICON and CESM2 at SH high latitudes. Here, the significance is the consequence of small variability in those models without interactive chemistry.'

**- Figure7 (and for some other figures): Why did you use "standard error" not "standard deviation" for indicating the model spread? The standard deviation is appropriate for this purpose.**
The standard error (of the mean) is shown in all the line plots to better represent the uncertainty associated with the CMIP6 MMM. In contrast, the standard deviation provides a measure of the spread about the mean, and we feel that this can be appreciated by seeing the individual models which comprise the MMM.

**- L440: I could not see the TCO increase of "20-30DU" from 1850 to 1960 in NH in Figure 7. Could you revise the number?**
We thank the reviewer for pointing this out – the correct range is 10-15 DU. This change has been made in the revised manuscript.

**- L440-441: English is too complicated for me to understand correctly what it means.**
This sentence has been modified to now read 'In the NH, TCO values increase by 10-15 DU between 1850 and 1960. This increase in TCO is larger than the TCO depletion that occurs from 1960 to 2000 in response to the emission of halogenated ODSs, resulting in higher NH mid-latitude TCO values in the late 1990s than in the pre-industrial.'

**- L446: How did you evaluate the "ability" of models to simulate pre-industrial TCO? Since we don't have TCO observation in that era, we cannot ensure the model's ability through comparing model results with observation.**
This has been changed to 'There is poor agreement in the simulation of pre-industrial TCO across CMIP6 models'

**- L448-454: Could you make more discussion about the difference among the models in simulating the past TCO changes. Discussions on why the models prescribing CMIP6 ozone data showed such a large discrepancy and those on the overestimation of ozone decline by some models are desirable.**
We have added a new section to the manuscript (section 3.4) comparing models with prescribed vs interactive ozone fields which explores in more detail the historic changes in TCO across the models.

**- Figure 8: Is the SSP370 scenario simulation result necessary for this figure. This part was never referred in the manuscript.**
We feel that inclusion of the SSP370 scenario in this figure gives important information about the future changes to stratospheric column ozone in the models explored in the figure, highlighting the fact that future TCO increases are driven by significant increases in upper stratospheric ozone, while lower stratospheric ozone increases occur much more slowly (due to increases at high latitudes being

offset by decreases in low latitudes, see our Figure 10.). We have added more text to the discussion of Figure 8 to include this point and cover the SSP370 scenario projections shown in the figure.

**- L458: Typo. "TCO seen in Figure 8" -> Figure 7**
Corrected

**- L460: Where did you describe about a large tropospheric ozone bias of UKESM1-0-LL in the manuscript? Which figure show that?**
**In the manuscript we state that '**It is also clear from Figure 8 that much of the high TCO bias for the UKESM1-0-LL model (Figure A2) comes from elevated stratospheric ozone mixing ratios, rather than a large tropospheric ozone bias.' While we do not plot the tropospheric ozone column, we see in Figure 8 that the UKESM1-0-LL model has a significant stratospheric ozone bias which is contributing to the TCO bias. If Figure 8 had shown UKESM1-0-LL stratospheric partial columns in agreement with the other models then the only location that would be left would be the troposphere. So the argument being made here is based on inference rather than being directly shown.

**- L490: Is this for SSP1-1.9 scenario not for SSP1-2.6?**
No, we mean here SSP1-2.6 as TCO values do not return to the 1960 or 1980 baseline in the northern midlatitudes under SSP1-1.9.

**- L515: Could you add the changes in the BDC simulated in the CMIP6 models to Figure 10?**
Unfortunately, this cannot be done as the models have not widely output and made available the diagnostics required to do this. However, the reviewer makes an interesting point about explicitly identifying the changes in stratospheric circulation in CMIP6 models, and it is hoped that as more diagnostics become available in the future this can be done in other studies.

**- L529-530: Figure A4 should be cited here if you want to refer to the percentage difference, since Figure 11 cannot show it.**
This has been done.

**- L535-536: How is the temperature at the tropical tropopause in the CNRM models? Is there any low temperature bias there which can cause the dry bias in the stratosphere in those models?**
The CMIP6 model data used here is only available on 19 pressure levels, and many models do not provide the tropopause pressure as a diagnostic. As a result, it is very difficult to determine the cold point tropical tropopause temperature for individual models. We agree with the reviewer that it is likely that the low $H_2O$ mixing ratios seen in the CNRM models in the lowermost stratosphere is associated with a cold tropical tropopause. However, it is not possible to state this definitively, and so we do not make this conclusion in the manuscript.

**- L538" As for ?**
This has been changes to 'As with'

**- L555: What does "CH4" exactly means in this equation? Concentration? Mixing ratio? What is its unit?**
Mixing ratio – this sentence has been amended to read 'the tropical stratosphere $H_2O$ mixing ratio will equal 7.0-2.0*$CH_4$ mixing ratio'

**- Figure 14: There are no reference to Figure 14 in the manuscript. The figure capture does not include the description of color bar. What does each point in the figure represent? Are they annual mean? Horrible lack of information for this figure.**

The discussion of Figure 14 is in the final paragraph of section 4.1 – we have now explicitly referenced the figure here. The figure caption has been significantly expanded, and now reads 'H2O vs CH4 scatter plots of the six CMIP6 models for which both H2O and CH4 mixing ratio are available from the historical simulation. The data shown here is monthly mean, zonal mean H2O and CH4 mixing ratios (in ppmv) for the years 2000-2014. The coloured shading of the points represents the altitude (in hPa). The black line gives gradient for all model points above 70 hPa, while the dashed black line gives SPARC estimate (H2O = 7-2*CH4)'.

**- Figure 15: Why you did not comment anything on the comparison with the observation. The CMIP6 models apparently underestimate the observation and the modelled increasing trend in the stratospheric water vapor can not be seen in the observation. You should discuss about those comparison in the manuscript.**
We have added more text describing Figure 15, particularly focusing on how the models compare to the SWOOSH observations, to the revised manuscript.

**- L580-582: How is the temperature change at the tropopause not at the 100 hPa? Is the increase at 100 hPa temperature and the increase in water vapor quantitatively consistent with each other?**
It is a significant challenge to calculate the temperature change at the model tropopause rather than at a given pressure level as few models have provided data on the pressure/height of their tropopauses and the vertical resolution of the data used here would make calculating a lapse rate tropopause from the temperature fields inaccurate. Generally, we can see that the 70 hPa H2O mixing ratios agree well with the 100 hPa temperatures by comparing Figures 15 and 16, but of course these annual mean temperatures at a fixed pressure level are only a very rough estimate of the tropical tropopause cold point temperatures, and so identifying a quantitative relationship between them (for example based on Clausius–Clapeyron) is a significant challenge.

**- L588-590: Could you show separately the relative contribution of 100 hPa temperature rise and CH4 concentration increase for the stratospheric water vapor increase in the CMIP6 models? Both for historical simulation and future projections.**
This is an interesting point, but very difficult to do as we lack the diagnostics to achieve this. Only a handful of models have output CH4 mixing ratios, and even from these models it is clear that there are significant differences between the amount of water vapour formed per molecule of CH4, even for models which report including CH4 oxidation, and so we cannot generalise across the models which do not provide CH4 mixing ratios.

---

## Author Response (AR2)

We thank the reviewer for these comments. Please find the original comments in bold, followed by our responses below:

**+ The description on the forcings to drive the CCMs both in the past and future simulations should be some more detailed. The information included in the authors response to my comment about L273 is desirable to be included in the manuscript.**

5    The additional information included in our responses to the reviewer's original comment has been included in this section.

**+ Caption of Fig.2 should describe about the gray shaded region as with the caption of Fig 13.**

This             is             done             –             the             caption             now             reads:
Figure 2: Climatological (2000-2014) seasonal cycle of ozone (in ppmv) at 70 hPa (15 S-15 N average) for CMIP6 models, the CMIP6 multi-model mean (MMM; solid black line) and SWOOSH combined ozone dataset (dashed black line). The light
10   grey envelope indicates the model spread about the MMM, calculated as the standard error of the mean.

**+ L513 of the revised manuscript: (typo) -0.17 should be -0.18**

corrected

**+ Table 2 of the revised manuscript: The trend value of CNRN-CM6-1 (-0.78DU/yr) is not with in the range of observation.**

15   We thank the review for spotting this, and explicit reference to the CNRM-CM6-1 model has been removed from the sentence discussing the northern mid-latitude trends shown in Figure 2, so that the sentence in the revised manuscript reads:

In the northern mid-latitudes, models considerably underestimate the observed negative trends with the exception of CNRM-ESM2-1 (-0.95 DU/year – within the range of observed trends) and UKESM1-0-LL (-1.2 DU/year – slightly stronger than the observed negative trends).

20   **+ L649 of the revised manuscript: (typo) 11 models should be 10 models**

corrected

**+ Since the figure 5 in the revised manuscript was more unclear than the previous manuscript, I could not properly check how much the author had responded to my concerns about this figure, and the author did not respond to my comments about this figure in their reply.**

25   We apologise for this – in response to the reviewer's initial comment we prepared a higher resolution version of figure 5 with larger symbols. However, this figure was poorly formatted when added to the revised manuscript. I include. Figure 5 here, and have ensured this version of the figure appears in the revised manuscript.

[Figure]

+ In the newly added section (section 3.4) in the revised manuscript, the authors discussed about the difference between "interactive" and "prescribed" chemistry models. They concluded "there is no systematic bias" between them, but in figures A3 and A5 showing the difference between the two, there are apparent differences particularly low and mid latitude region. interactive models calculate more TCO in these regions than prescribed models. So, their conclusion is not suitable to describe the situation. I think there is small but systematic bias between interactive and prescribed models.

To avoid confusion in this section we have removed reference to systematic biases. Instead, the section now highlights that TCO in the mean of models with interactive chemistry is slightly higher (by ~5 DU/2%) than the CMIP6 MMM, with the difference coming principally from the tropics. We then highlight however that this difference is likely caused by very high TCO values in the UKESM1 model, and that there is no consistent bias between the models with interactive and prescribed ozone fields, i.e., models with interactive chemistry schemes do not all have consistently higher TCO values than found in models prescribing stratospheric ozone, and instead fall either side of the prescribed multi-model mean. In this way we address

the reviewers point and make it clear that as an ensemble the interactive models do have higher TCO, but that you cannot take any one model with interactive chemistry and necessarily expect it to have larger TCO values than a model prescribing the CMIP6 ozone dataset.

**+ Fig 12 of the revised manuscript: What is the black contours in the difference figure? Should add about it in figure caption.**

The black contours were intended to show H2O mixing ratios in the CMIP6 MMM and CMIP6 MMM CH4 Oxidation cases. However, as these values are shown in the panels to the left of the figure, I had intended to remove them, and not realised that very high mixing ratios remained in the troposphere. These have been removed from the updated figure.

Please find on the following pages the manuscript with tracked changes. Note also that a co-author was omitted from the most recent manuscript and has been added.

[revised manuscript text omitted]